# Identification of transporter-dependent capsular loci associated with the invasive potential of *Escherichia coli*

Rebecca A. Gladstone [1] ✉, Maiju Pesonen[2], Anna K. Pöntinen [1], Tommi Mäklin [1,3], Neil MacAlasdair[1,4], Harry Thorpe [1], Yan Shao[4], Sudaraka Mallawaarachchi[1,5,6], Sergio Arredondo-Alonso[1], Benjamin J. Parcell[7], Jake David Turnbull[8], Gerry Tonkin-hill[1,5,6,9], Pål J. Johnsen[10], Ørjan Samuelsen [11], Nicholas R. Thomson [4,12], Trevor Lawley [4] & Jukka Corander [1,3,4] ✉

Bacterial polysaccharide capsules contribute to antigenic diversity and immune evasion. *Escherichia coli* infections, including those caused by extraintestinal pathogenic *E. coli* (ExPEC), cause substantial antimicrobial resistance-associated morbidity and mortality. However, much-needed genotypic methods for *E. coli* capsule typing to aid epidemiological analysis and therapeutic design are lacking. Here we describe the curation of an in silico typing database for group 2 and 3 ATP-binding cassette transporter-dependent capsule (K) loci from 18,185 *kps*-positive *E. coli* genomes from all continents and its application to carriage and ExPEC disease cohorts. Capsules K1, K5 and K2 were the most common types in European BSIs, and together with K100 and K52 they were responsible for 58% of multidrug resistance, with differing associations with invasiveness. Homologous recombination, insertion sequences and plasmids were associated with capsular gene exchange. These findings improve understanding of capsule epidemiology and evolution to inform future diagnostic and therapeutic strategies to combat ExPEC infections.

Capsules are major virulence determinants in bacterial pathogens. They have many critical roles, including shielding bacteria from the immune system, influencing their ability to cause invasive infections and acting as a barrier to antimicrobials. Bacterial capsular polysaccharide antigens are often well studied and used as a central component of disease surveillance and as effective vaccine targets, including for *Streptococcus pneumoniae*, *Haemophilus influenzae* and *Neisseria meningitidis*[1]. Although *Escherichia coli* is the leading cause of bacterial blood stream infections (BSIs) globally and imposes the highest burden of antimicrobial-resistant BSI-associated deaths[2], the true prevalence of different capsular types in contemporary disease has not been determined in large, unbiased extraintestinal pathogenic

*E. coli* (ExPEC) infection cohorts. Capsules are natural targets for translational research to develop new antimicrobials and vaccines. There have also been early in vivo successes for phage therapy in invasive ExPEC infections, which target the capsule and therefore require knowledge of the exact capsular (K) type causing the infection[3].

Traditional phenotypic *E. coli* K typing using a set of antisera is labour intensive and no longer in general use. Unlike for the O antigen of lipopolysaccharides and H antigen of the flagellin protein[4], no genotypic method exists for typing the K locus that encodes K antigens. *E. coli* K loci have been classified into four groups based on whether they are *wzy* dependent (groups 1 and 4) or *kps*-ATP-binding cassette (ABC) transporter dependent (groups 2 and 3) and by the

**Fig. 1 | K type prevalence across different collections.** Published genomic collections were K typed[15,17–20,27–32]. The top five K types in each collection are annotated. Untypeable isolates (that is, genomes without G2 or G3 *kps* genes) are denoted as non-G2 and -G3. The colours highlight shared K types between collections.

## Results

### Identifying K loci

We downloaded all *kps*-positive *E. coli* from a published, searchable collection of 661,000 bacterial assemblies[14] and supplemented them with published ExPEC genomic studies[15–22]. Genomes with a known phenotypic K type were sourced for 24 different K types from GenBank, the National Collection of Type Cultures (NCTC) project[23] and EnteroBase[24]. From the subsequent 18,185 *kps*-positive *E. coli* genomes, we extracted and annotated the K locus and observed 90 G2 (*n* = 68) and G3 (*n* = 22) ABC transporter-dependent K loci, based on unique K locus gene presence and absence patterns using Panaroo[25]. These unique K loci were used to create a GenBank reference database, compatible with the species-agnostic bacterial capsular locus typing tool Kaptive[26], to allow in silico G2 and G3 *E. coli* K typing from assemblies. Of the known G2 and G3 phenotypic K types, 80% are represented here by phenotype–genotype paired data, yet known phenotypes only represent 34% of the 90 K loci observed to have unique capsular gene sets. Furthermore, for 11 K loci with no known phenotype observed in Norwegian BSIs, we confirmed a K+ phenotype, with a positive precipitate on reaction

with Cetavlon, and found them to be negative for known K antigens, representing putatively novel K types. We K locus typed 6,626 assemblies from several systematic studies of infections, including: (1) large longitudinal genomic surveys of BSIs from Norway[15] and the UK[17,18,27]; (2) urinary tract infection (UTI) surveys from Norway and France[28]; (3) studies of UTIs with specific resistance profiles from the USA[29]; and (4) two collections of invasive isolates from neonates collected in low- and middle-income countries (LMICs)[30,31]. In addition, K locus typing of 1,330 assemblies from infant–mother gut metagenomic surveys in the UK[19,20] and traditional culture picks in France[32] provided information on K types in asymptomatic colonization. Throughout this manuscript, we will report the inferred phenotypic K type (for example, K1) for a given K locus (for example, KL1) genotype when available. The numbering starts at KL110 for unknown phenotypes.

### K locus epidemiology

European BSIs[15,17,18,27] in our study were predominantly caused by strains with a G2 or G3 capsule (81.3%); among them, G2 dominated, and only a minority had G3 loci (3.5%). The vast majority of BSI isolates in phylogroups B2 (94.2%), D (84.1%) and F (94.6%) had G2 K loci. The top five K types in European BSIs were K1, K5, K52, K2 and K14 (Fig. 1). These common K types accounted for over 50% of BSIs (UK[17,18,27] and Norway[15]) and UTIs (Norway[28] and France[28]). K1, K5, K52, K2 and K100 accounted for 58% of European multidrug-resistant (MDR) BSIs. Although G2 and G3 K loci were far less common in asymptomatic colonization in Europe (55.4%)[19,20,32], K1 and K5 were still among the most common colonizing G2 and G3 K types. We found no significant differences in the overall K type composition between the UK infant–mother collection and colonization in French adults. Only a single capsule type (K1) and a clonal complex (CC10) had a significantly higher prevalence in the French collection. Notably, G2 and G3 loci accounted for fewer invasive neonatal infections in LMICs[30,31] (53.9%; Fig. 1), in stark contrast with the 95% of UK BSIs in children <1 year of age that were G2 or G3.

**Estimated relative invasive potential of K types.** Using a mixed-effect model comparing UK isolates from carriage and BSIs, we estimated the association between each K locus and lineage and the relative invasive potential (Fig. 2). We determined that K52 had the highest invasive potential (odds ratio (OR) = 11.6; 95% confidence interval (CI) = 5.3–25.4; $P < 0.0001$) compared with the average untypeable isolate (genomes lacking the essential *kps* genes for G2 and G3 capsules). K14 and K100 had the second- and third-highest invasive potential, respectively (OR = 8.9 (95% CI = 3.7–21.7) and OR = 7.2 (95% CI = 2.6–20.0)). The lineage-specific odds of K100 being found in BSIs were estimated to vary within CC131 between 5.42 (CC131 clade A) and 69.80 (CC131 clade C2). There was also substantial variation in lineage random effect estimates between CC131 subclades B and C despite sharing the same O:H type. The ORs for the widely studied K types K1, K2 and K5 were ranked twelfth, fifth and sixth, respectively. These three types nevertheless had ORs significantly greater than 1, indicating greater invasive potential than untypeables. Conversely, K3 had the lowest invasive potential (OR = 0.82; 95% CI = 0.2–2.7).

We assessed the burden of *E. coli* BSI by age and sex (Extended Data Figs. 1–3) and, in an exploratory analysis, observed that K1 was over-represented in the <1 and 40–49 years age groups, but these differences were not significant after adjusting for multiple testing across all common age–K type combinations. We observed that CC95 carrying the K1 capsular polysaccharide, but not K1 overall, was enriched in females (adjusted $P = 0.013$), and K52, predominantly found in the uropathogenic CC69, was observed in females in every decade of life but only occurred in older men aged ≥50 years, making it significantly over-represented in females (adjusted $P < 0.0001$). The proportion of BSIs in the 1–59 years age group was significantly higher among females (60%) than among males (40%; $P < 0.0001$), but was similar overall (females 52%; males 48%). We quantified the odds of a patient with BSI being ≥60 years old in males compared with females for each common lineage–capsule combination. For most lineage–capsule combinations, there were no significant differences between males and females across the two age categories, except for CC131–C2 with the K5 capsule, for which older males had higher odds. Limiting the BSI model input data to only elderly adults (≥60 years; 74%) had a negligible effect on the rank order. Hard-to-treat infections could be over-represented in BSI isolates due to a greater likelihood of progression to systemic infection; however, we determined that there was no correlation between lineage random effects and the MDR or bla$_{CTX-M}$ proportion of that lineage (MDR $R^2 = 0.029$ ($P = 0.79$); bla$_{CTX-M}$ (excluding C2) $R^2 = 0.036$ ($P = 0.74$)). Including the bla$_{CTX-M}$ proportion per lineage as a covariate in the model was not significant.

**Group 2 K locus association with *E. coli* and ExPEC pathotype.** *E. coli* G2 *kps* genes are not often found in other species. In a published collection of 661,000 assemblies[14], only 11,737 were positive for the highly conserved and essential G2 *kpsF* gene (1 kilobase) at 90% *k*-mer identity. These were nearly exclusively *E. coli* (99.0%). Outside of *E. coli*, *kpsF* was most commonly observed in *Salmonella enterica* subspecies *enterica* ($n = 69$), *K. pneumoniae* ($n = 17$) and *Staphylococcus aureus* ($n = 11$). Using a published pangenome of ~7,500 *E. coli* with inferred pathotypes[33], we assessed the pathotype association for G2. ExPECs accounted for 85.8% (1,590/1,853) of the *kpsF*-positive isolates with pathotype information, and non-ExPEC pathotypes accounted for 97.5% (3,647/3,741) of the *kpsF*-negative isolates. Shiga toxin-producing *E. coli* accounted for the majority of non-ExPEC pathotypes positive for *kpsF* (90.5%; 238/263), and these Shiga toxin-producing *E. coli* were found across multiple sequence types, including ST442, ST10, ST25, ST504 and ST675.

**K loci in major MDR lineages.** The globally disseminated MDR lineage corresponding to clonal complex CC131 is known to have two dominant O:H types. However, we observed that at least 17 distinct K loci were introduced into CC131, highlighting rapid capsule diversification

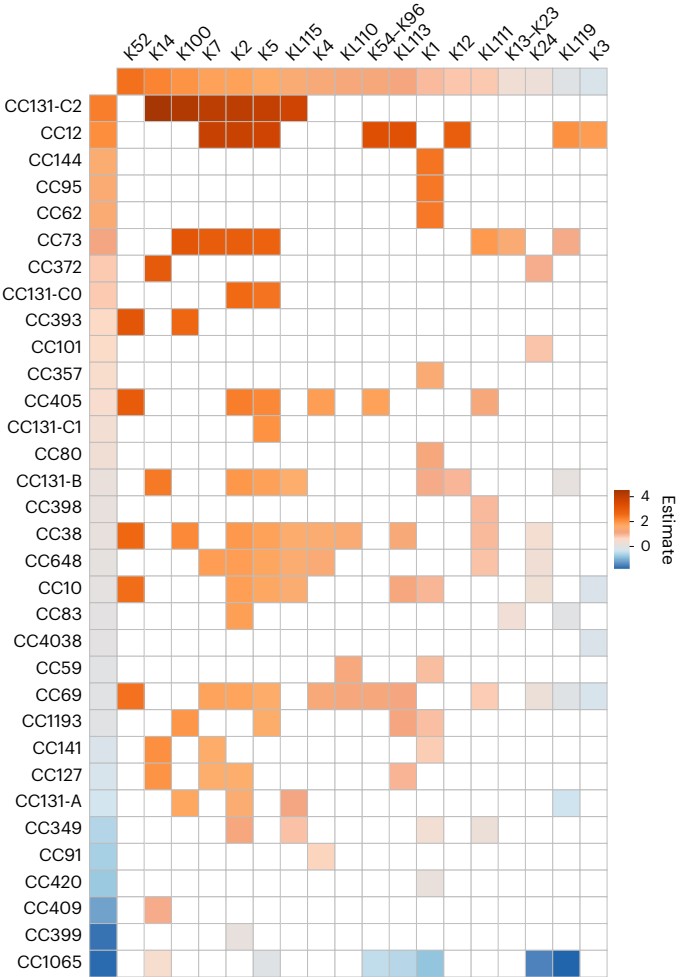

**Fig. 2 | Estimated marginal and combined invasive potentials of different K loci and lineages.** The columns show K loci and the rows show lineages. The colours represent regression coefficients on a logit scale (log odds), estimated from a generalized mixed model with clinical manifestation (infection or carriage) as the binary outcome, K loci as fixed effects and lineages as random effects. Lineages are denoted as clonal complexes derived from the representative sequence type for each PopPUNK lineage. Red tones are associated with higher estimates of invasive potential and blue tones are associated with lower estimates. The reference category for K loci was untypeable (that is, genomes without G2 and G3 *kps* genes). The rows and columns are sorted by the relative invasive potential estimates from highest to lowest. The CC131 lineage is split into its major clades (A, B, C1 and C2). The corresponding K type, if known, is presented for a given K locus.

in this lineage (Fig. 3a). The ancestral K type for clades B and C is K5 ($n = 207/453$; most recent common ancestor (MRCA) 1929 (95% credibility interval (CI):1917–1939)). For clade A, the ancestral K type is K100 ($n = 80/104$; MRCA 1974 (CI: 1968–1979)). Other K loci with notable prevalence in CC131 BSIs include K14 (MRCA 1984 (CI: 1979–1989)), KL112 (MRCA 1994 (CI: 1989–1997)) and K2, which was acquired an estimated ten times independently across the lineage. Interestingly, KL112 in CC131 ($n = 27$) is an example of a K locus with an atypical architecture, where the conserved *kps* region 1 is immediately followed by *kps* region 3, and region 2 with the capsular-determining genes is at the end of the locus. There were five subsequent fragmentation and complete G2 loss events after its acquisition.

In another globally distributed MDR ExPEC lineage, CC69, we observed 11 O types, nine H types and 23 K types. O diversity in CC69 was higher than in CC131 (Simpson's diversity index (SDI) values: 0.66 and 0.39, respectively), but still less than CC69 K locus diversity (SDI value: 0.76). K loci belonging to G2, atypical G2, and G3 are all found in this

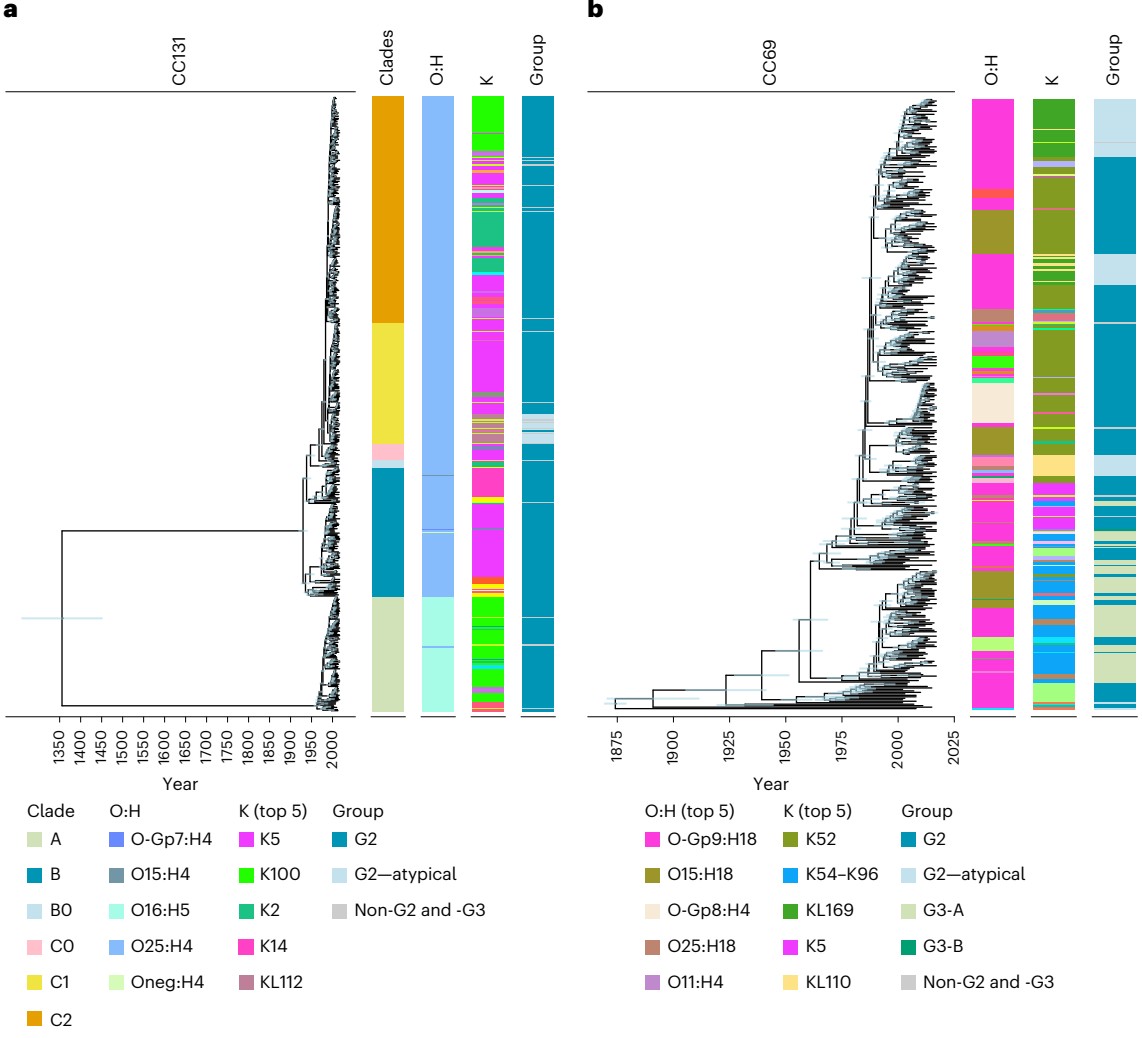

**Fig. 3 | Phylogenetic distribution of K loci across the clades of CC131 and CC69 from Norwegian and UK BSIs. a**, CC131 phylogeny, with clade, O:H type, K type and group metadata displayed. The top five K types are displayed in the legend. **b**, CC69 phylogeny, with O:H, K type and group metadata displayed. The top five O:H and K types are displayed in the legend. Where the K phenotype is unknown,

the K locus is presented. Dated lineage phylogenies with node age uncertainty (0.95 highest posterior density) are plotted as light blue. Fully annotated interactive phylogenies are available at https://microreact.org/project/cc131-bsac-norm and https://microreact.org/project/cc69-bsac-norm.

lineage (Fig. 3b). The most common K locus was K52 (39.5% (163/413); 1984 (1980–1987)), which had the highest estimated invasive potential (Fig. 2). Switches from K52 to an unrelated atypical K locus occurred at least three times after 1990.

### K locus evolution

Although K diversity measured by SDI was higher than O or H diversity in over half of the lineages observed in BSIs, lineages varied greatly in K type diversity (Fig. 4a). The number of K loci in a lineage was correlated with the recombination-to-mutation ratio ($r/m$) of that lineage, even when controlling for the lineage diversity by using the recombination-free median mutational pairwise distance (MMD; $R^2 = 0.95$; $P < 0.0001$; Fig. 4b). There was no correlation between $r/m$ and MMD ($R^2 = -0.35$; $P = 0.3$). The trend remained significant after excluding CC69 ($R^2 = 0.75$; $P = 0.02$), but not after excluding the two most recombinogenic lineages, CC69 and CC131 ($R^2 = 0.47$; $P = 0.2$). In the top four lineages in BSIs (CC73, CC95, CC69 and CC131), the $kps$ locus is a clear recombination hotspot, except for CC95, which expresses only K1.

**K locus structural variation.** Known K phenotypes accounted for only 34% of K loci, with these phenotypes sporadically distributed

across the K locus phylogeny (Fig. 5). Many K loci correspond to deep ancestral branches that probably represent antigenically distinct K types. In total, there were 238 region 2 capsular gene clusters (>70% amino acid sequence identity), of which 13.0% were annotated as a hypothetical protein, demonstrating considerable unexplored capsular-determining gene variation. The eight $kps$ genes in the G2 K loci formed a single major gene cluster each. Only two of the G2 K loci contained $kps$ genes in a divergent cluster for at least one of the $kpsM$, $kpsT$, $kpsS$ and/or $kpsC$ genes, which are important in forming the biosynthesis–export complex[34] (Fig. 5a). Nine K loci feature an atypical G2 locus structure that has not been previously reported, where region 2 is outside of regions 1 and 3 (Extended Data Fig. 4). Although atypical K locus organizations are relatively rare in BSIs (<3.5%), they were spread across the species phylogeny and found in phylogroups A, B2, D and F. There were also sizable clusters of atypical loci in major *E. coli* BSI lineages: CC69 (KL110; $n = 84/449$), CC131 (KL112; $n = 25/631$), CC73 (KL112; $n = 20/981$) and CC59 (KL110; $n = 14/93$). A Norwegian BSI isolate with the atypical K locus KL144 was confirmed to be K+ but did not react with known K antisera.

Unlike in G2, the diverse $kps$ genes in G3 form two major gene clusters delineating the G3-A and G3-B subtypes (Fig. 5b). No G3 $kps$

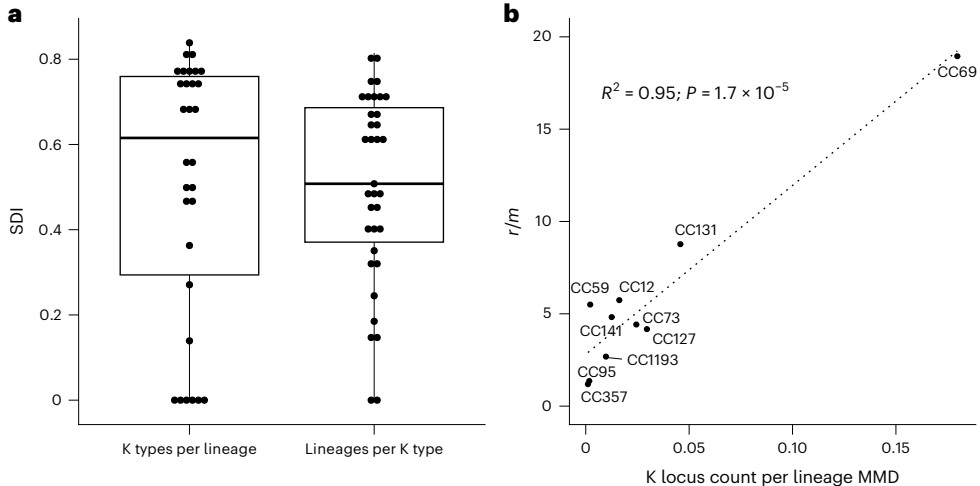

**Fig. 4 | SDI and Pearson correlation analyses, revealing associations between lineages and K types. a**, SDI 1-D for the richness and evenness of K types within lineages (left) and lineages within K types (right). For K types per lineage ($n = 30$ lineages), the values for the lower whisker (minimum), 25th percentile, median, 75th percentile and upper whisker are 0, 0.29, 0.62, 0.76 and 0.84, respectively. For lineages per K type ($n = 35$ K types), they are 0, 0.37, 0.51, 0.69 and 0.81, respectively. **b**, Two-sided Pearson correlation between the $r/m$ and the number of K loci per lineage, adjusted by the lineage diversity (MMD) and plotted for ten lineages. No adjustment for multiple comparisons was applied as only a single correlation was tested, addressing this specific, a priori hypothesis. The line of best fit is plotted as a dashed line.

genes clustered with G2 *kps* genes. Within G3-A and G3-B, the K loci share region 2 gene clusters for *wckE* and *rmlBDAC*. Meanwhile, K11, K19, KL151, KL160 and KL172 loci have divergent and unique region 2 gene sets. K19 was observed in phylogroup A and shares no *kps* gene clusters with G2; KL172 has previously been classified as G3; and G3 has been suggested to be divergent G2 K loci elsewhere[35,36]. Therefore, we denote these three loci as G3-C to G3-E.

**K locus diversification and gene mobility.** The proportion of unique K locus sequences (with a difference of at least 1 base pair (bp)) with at least one insertion sequence element was 28.2% (1,411/4,996). Whereas K1 was observed with an insertion sequence only once ($n = 1/692$), K5 was observed with at least one insertion sequence in all cases ($n = 706$). Insertion sequence 1 (IS1) and IS3 were the most common of the ten insertion sequence families observed in the K locus collection. Importantly, insertion sequences were observed overlapping with capsular coding sequences in 17 K loci, and in nine of these the insertion sequence carried a putative capsule gene in region 2 as cargo, many of which were in K loci related to K4 and K5.

The K locus KL4 (K type K4) is one of four closely related K loci based on capsular gene presence–absence patterns (Extended Data Fig. 5), and one KL4 variant (KL4-1) that differed only by non-capsular insertion sequence genes was observed where *kfoB* is split into two fragments and *kfoD* is contained within the insertion sequence element. The *kfoA*, *kfoB*, *kfoC* and *kfoD* genes are present in all five K loci, whereas *kfoE* and *kfoF* (gene cluster name *udg*) were absent in KL148 and KL156, respectively. Insertion sequence elements were all observed between *kpsS* and *kfoC* and probably contributed to the different patterns of capsular-specific genes within region 2 for this cluster of related K loci. K4 (KL4) was the most common of the K4-like loci in BSIs ($n = 54$).

There were three different K loci related to K5 (KL5) with differing capsular gene presence–absence patterns and five KL5 variants differing only by non-capsular genes, including insertion sequence elements. The KL153 locus differs from KL5 by the loss of *kfiD*; this was rare and only observed in carriage ($n = 3$). In KL127, an IS1 family element belonging to the 316 cluster contained a putative glycosyltransferase family 2 protein (group 145) and a truncated *kfiC* within its terminal inverted repeats (Extended Data Fig. 6). This was observed in BSIs ($n = 4$) and carriage ($n = 5$). An insertion sequence variant of the KL5 locus (KL5-1) seen in a single ST131 isolate had an additional transposase

and contained a fragment of the extended-spectrum β-lactamase gene $bla_{CTX-M}$ near the IS3 remnant. The $bla_{CTX-M}$ and transposase gene were observed together in multiple plasmid types within CC131 (100% identity), suggesting that both genes may have moved into this K locus from a plasmid via homologous recombination with the existing insertion sequence in these K loci. Although insertion sequence variants were rare, their existence provides insight into how capsular diversification and gene exchange may be driven by insertion sequences. No insertion sequences were observed in the atypical loci, suggesting that other mechanisms are also at play.

G3-B K loci have been observed in plasmids, which could also act as a vehicle for capsular mobility[36]. As G3-B K loci are rare in BSIs ($n = 11$), we observed just two Norwegian hybrid assemblies with G3-B K11 in multireplicon plasmids, but not in the chromosome. One of the isolates had K11 mobilized on a plasmid with multiple resistance genes: $bla_{TEM-1}$, *dfrA14*, *mph(A)*, *sul2*, *aph(3")-Ib* and *aph(6)-Id*. Additionally, a single *E. coli* hybrid assembly from Oxfordshire ($n = 1/549$) was typed as G3-B KL132 on the plasmid and not in the chromosome.

## Discussion

Extensive phenotypic and structural data have demonstrated considerable diversity in *E. coli* capsular antigens. Systematic cataloguing of capsular genetic diversity in contemporary disease is essential to further our understanding of the epidemiology of K types. In this Article, we have shown that the diversity of G2 and G3 K loci found in BSIs far exceeds the number of the currently known G2 and G3 phenotypic capsular types. Furthermore, this diversity has major implications for the development of preventive strategies, with observed differences in K type prevalence by isolation source and geographical location, as well as variation in the associated invasive potential of K types. We determined that G2 and G3 capsules are present in the majority of ExPEC infections across diverse settings. Specifically, K1 and K5 are the most common K types found in UTIs and BSIs. Although these two types are also the most common G2 and G3 K types in LMIC neonatal disease, a notable fraction of these infections are caused by non-G2 and -G3 K types. Although existing LMIC data are limited, our results clearly suggest that capsular epidemiology differs substantially from that in high-resource settings for at least vulnerable neonates, necessitating prioritizing future sampling efforts to generate region-specific, representative data similar to that for other pathogens[37–39]. In silico typing

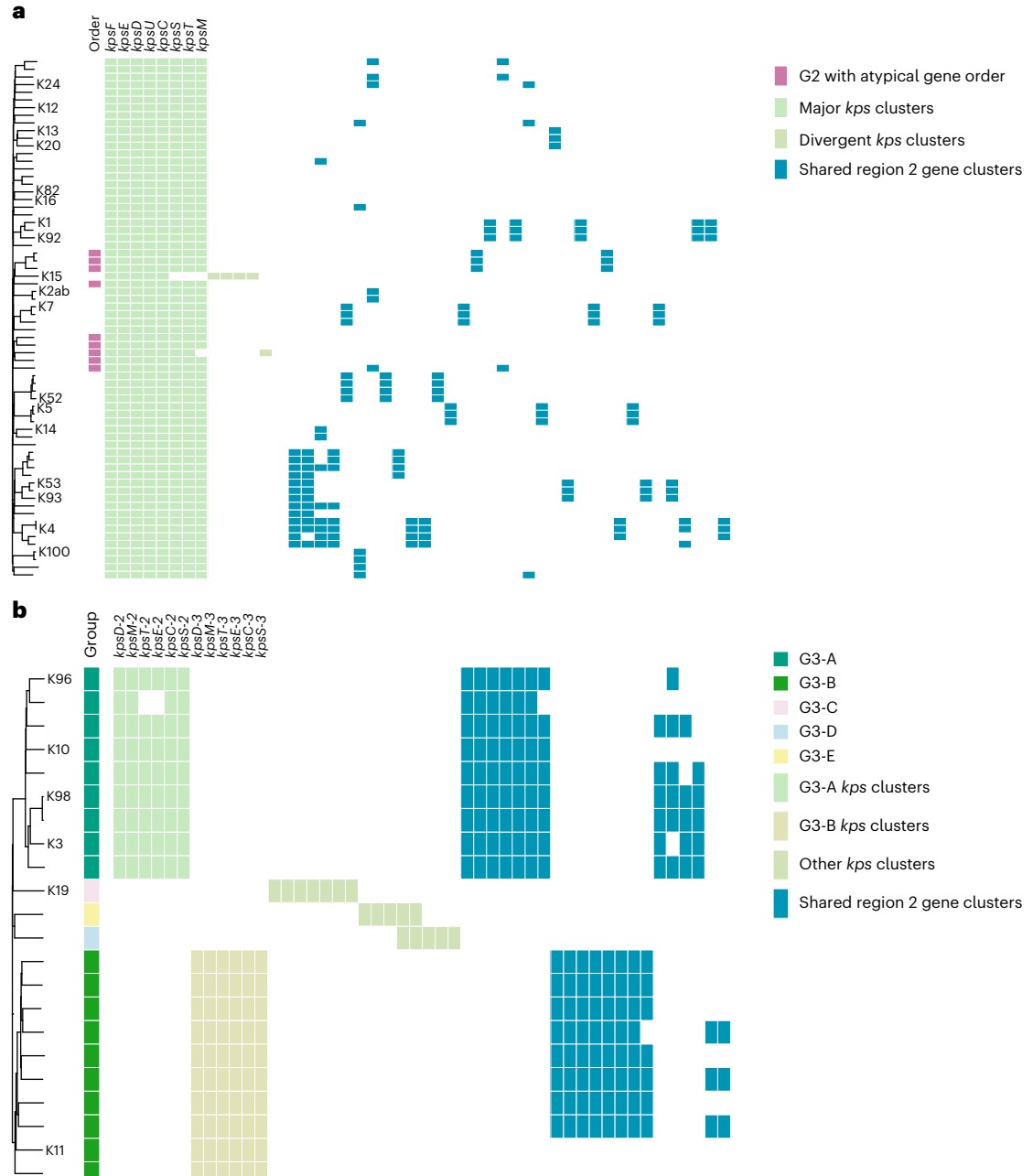

**Fig. 5 | Capsular gene presence and absence patterns of G2 and G3 K loci.**
**a,b**, Neighbour-joining trees of G2 (**a**) and G3 (**b**) K loci based on K locus core and accessory sequence presence or absence, annotated with known K phenotypes at the tips. The first metadata column in **a** denotes K loci with atypical gene organization (pink). In **b**, the first column shows subtypes of G3. To the right, the presence or absence of gene clusters is shown, first for the dominant *kps* gene clusters in the order they are most often observed in the K locus (labelled), then for divergent (**a**) or other (**b**) *kps* gene clusters, and finally for the most common region 2 gene clusters, ordered by the number of K loci in which they were observed. Only gene clusters present in more than two K loci are presented. These figures can be viewed interactively with full taxa, gene sets and annotations on phandango.net using the input files provided at https://github. com/rgladstone/EC-K-typing/tree/main/phandango.

of the *wzy*-dependent group 1 and group 4 capsules is another critical knowledge gap for understanding ExPEC disease in resource-limited settings. Nonetheless, the G2 and G3 K locus repertoire observed in LMICs was well represented by our G2 and G3 database.

The role of capsules in pathogenesis has long been appreciated across bacterial species[1,40]. Previous research has experimentally demonstrated the virulence associated with common *E. coli* capsular types (for example, K1, K2, K5, K52, K92 and K100) using either human immunological assays or murine models[41–46]. An epidemiological approach that leverages measured frequencies in systematically collected

colonization and disease isolates to calculate an OR indirectly captures the relationship between exposure and the subsequent manifestation of disease cases in a population[17,19,47]. These estimates aim to capture a fixed trait of the capsule to provide a ranking for prioritizing inclusion in vaccines and modelling optimal polyvalent vaccine formulations[48–54].

Efforts to quantify the invasive potential of different K types have been hindered by the scarcity of *E. coli* colonization data, which we addressed by leveraging microbiome data from a large-scale UK infant–mother cohort[19,20,55]. Using infant carriage to represent general colonization is one limitation of this approach. However, healthy

full-term babies acquire *E. coli* from family members and their surroundings during their first year of life, as reflected in strain sharing between mother and baby[19,56]. The similarity of French adult and UK infant–mother carriage populations suggests that there is limited age structure to human *E. coli* colonization in this geographical region, despite differences in the sampling period (2014–2017 versus 2010), sequencing approach (metagenomic versus single-colony sequencing) and geographical location.

Given the similarity between neonatal and adult carriage, we assumed that *E. coli* carriage is homogeneous across age groups. Although this carriage represents the general healthy population, the majority of BSIs (74%) occur in elderly adults (≥60 years of age) who often have comorbidities and contact with healthcare institutions that could influence colonization dynamics. The estimates reported in this manuscript therefore reflect the capsular types that are enriched in BSI, compared with the general healthy adult population. A matched cohort of colonized patients who did not progress to BSI would provide an ideal control to quantify the causal effects of capsules, accounting for potential age-based biases. We observed some age- and sex-related differences in BSI epidemiology; of note, there was an over-representation of men in the elderly adult category. We could not directly integrate age and sex as covariates into the model due to the strong correlation between age and dataset membership and therefore age and sex are possible confounders in this analysis.

Our results suggest that the high frequency of the widely studied K1 and K5 types in disease is, at least in part, a product of greater opportunistic spillover into vulnerable hosts. Our estimates also indicate that multiple K types are more strongly associated with infection when ranked by invasiveness. For example, K52 and K14, which are found in multiple lineages, including the global MDR lineages CC69 and CC131, had the highest estimates of invasiveness and subsequently ranked third and fourth in BSI prevalence despite rarely being seen in carriage samples in the UK and France (<2.5%). Although the estimated invasive potential of K1 was not top ranked, the ST95 lineage ranked fourth in the lineage random effects estimates, suggesting that the lineage is playing a key role in progression to BSIs, which is concordant with the many reported virulence factors present in this lineage[57–60]. In general, the lineage estimates are of similar magnitude to K types at least for the top-ranked lineages, again suggesting that other virulence factors within a lineage encoded outside the K locus are equally important for determining invasiveness.

The development of ExPEC vaccines is complicated because many *E. coli* are commensal, and some are considered beneficial to human health[61]. Vaccines should avoid targeting a major constituent of the gut microbiota and broad *E. coli* sterilizing immunity. This requires a greater understanding of the differences between predominantly commensal genotypes and successful ExPECs, as well as the pathogenic potential of ExPECs. K types with high estimated invasive potential are obvious vaccine targets due to both their capacity to cause disease and the likelihood that removing them from the gut flora will less often affect key *E. coli* commensals or lead to large-scale replacement in carriage and disease as other colonizers fill the vacated niche. Although ExPEC vaccines have been under development since the 1980s, only a few have been licensed. Whole cell, O, H, K and O conjugates have all undergone clinical trials[62]. Although evidence of K1 and K5 autoreactivity is limited, their mimicry of host glycobiology may explain why these anti-K antibodies are not strongly induced in vaccination or disease[62]. Nonetheless, increased genomic data and expanded O typing have contributed to the increased development of subunit-based vaccines rather than whole-cell vaccines in recent years[4,63]. Currently, there are over twice as many recognized O serogroups as K types[62]. Yet, we observed a greater diversity of K loci from G2 and G3 than O loci in most dominant ExPEC lineages.

Even greater diversity is likely to be discovered with further investigation into the genetic determinants and delineation of serogroups with shared gene content[64,65]. Such putative genetic determinants of antigenically distinct capsules need phenotypic validation to allow full genomic discrimination, which could be included as phenotype logic with Kaptive 3 (ref. 66). Alternatives to traditional K phenotyping are in development that could discriminate between subtly different K phenotypes and allow novel K types to be assessed more readily[67], and an alternative K locus typing method also identified large numbers of putatively novel K loci, and similar associations were reported between K types and invasive disease[68].

The evolution of *E. coli* G2 and G3 K loci is of considerable interest, given the fundamental biological role of the capsule and the complexity of its biosynthesis. We demonstrated that the overall tendency of homologous recombination across the core chromosome closely reflects the rate at which capsule switches occur via horizontal gene transfer, thereby influencing the K locus. Whereas earlier work detected acquisition of the K1 capsule by genetic lineages across several phylogroups over centuries[43], here we show that a similar process has influenced the spread of most of the successful and invasive capsule types. The ability to diversify the capsule locus for two major MDR lineages (CC69 and CC131) may have facilitated their rapid expansion in BSIs in the twenty-first century[15,17,18]. We further discovered that insertion sequence elements are common within the K locus and are probably contributing to diversification of the polysaccharide composition of capsules by importing genes into region 2, warranting deeper functional investigation of these evolutionary processes. This appears to be a unique feature of *E. coli* G2 and G3 capsules; whereas insertion sequence elements have been reported in other species such as *K. pneumoniae*[8,9], they could be completely removed from the typing scheme as they were less common and purely intergenic.

In summary, our results provide impetus for reinstating capsular typing for this species and renewed interest for experimental studies of virulence that could further disentangle the contributions of capsules versus lineages towards invasiveness, to allow the development of strategies to effectively control ExPEC disease.

## Methods

### Data

We used a published pangenome analysis of ~7,500 high-quality *E. coli* genomes to determine that the capsular gene *kpsF* was consistently annotated and predictive of G2 capsule presence[33]. We subsequently downloaded all *kpsF*-positive (0.9 *k*-mer identity threshold) *E. coli* from a published, searchable collection of 661,000 bacterial assemblies (*n* = 11,623)[14]. For G3, we additionally screened for all *kpsM*-positive assemblies (0.9 *k*-mer identity threshold; *n* = 853) using the G3-A and G3-B *kpsM* alleles from K96 and K11, respectively, as *kpsM* is more divergent in G3. We supplemented these genomes with data from large published *E. coli* BSI longitudinal genomic collections from Norway (2002–2017; *n* = 3,254; 60% hybrid assemblies)[15,16] and the UK (2001–2018; *n* = 2,219)[17] and a collection of infant and mother *E. coli* carriage assemblies from the UK Baby Biome study (*n* = 997; available from https://zenodo.org/records/14000489 (ref. 69))[19,20], and two One Health *E. coli* studies from the UK (*n* = 405) and USA (*n* = 2,948)[21,22]. Genomes with a known K phenotype were sourced for 24 different K types from GenBank, the NCTC project[23] and EnteroBase[24]. The NCTC reference strains (accession PRJEB6403) are maintained as preserved strains that can be obtained from the NCTC in the UK. A total of 18,185 K loci were extracted from assemblies. This was achieved using an in silico PCR (https://github.com/simonrharris/in_silico_pcr) with primers for known K locus boundaries (that is, *kpsF-kpsM* or *kpsM/D-kpsS*; Supplementary Data 1). PCR-negative K loci were manually extracted from assemblies that had been annotated with at least one *kps* gene.

### K loci

Extracted K loci with unknown bases were excluded and the K loci were filtered down to 4,996 unique sequences with at least 1 bp difference. These unique K loci were initially annotated with Prokka (version 1.14.5)[70]

and analysed with Panaroo (version 1.5.2)[25] and Bakta (version 1.11.4)[71] to consistent annotated K locus gene clusters with a conservative 70% family identity threshold. The gene cluster names were updated using the capsular-specific reference annotations to reflect the literature where possible. This resulted in an initial 225 gene absence patterns, excluding insertion sequence elements. Insertion sequence elements were identified from the annotations using ISEScan (version 1.7.2.3)[72]. These 225 sequences were pruned to remove K loci with redundant patterns due to misannotation, paralogues, non-capsular genes and incomplete K locus remnants, leaving 90 curated K loci in the final K locus database. K loci from complete genomes, long-read or hybrid assemblies and sequences containing the fewest insertion sequences were preferentially included as the database reference. The database was formatted for Kaptive 3 (version 3.0.0b5)[66], with insertion sequence annotations removed. The K locus nomenclature reflects the known paired phenotypes (for example, K1 is encoded by the KL1 locus). K loci for which K types have not yet been phenotypically identified were assigned K locus numbers starting from KL110.

Assemblies (n = 8,473) from published genomic collections[17–20,28–30,32,73] were K typed using Kaptive, and our G2 and G3 database (version 3.0.0), which is available at https://doi.org/10.5281/zenodo.18107967 (ref. 74). Untypeables with three or fewer essential *kps* genes were designated absent for G2 and G3 K loci. The assignment of chromosome or plasmid was available for hybrid assemblies from Norway and Oxfordshire[16,27]. These were used to determine whether any K loci were present in plasmids from these BSI collections.

The relatedness of K loci was assessed for G2 and G3 separately using the pairwise Mash (version 2.3)[75] hash distance of the K locus sequences converted to a proportion 1 − (n/1,000), which gives greater resolution than gene presence–absence but still captures core and accessory variation[75]. These distances were used to create neighbour-joining phylogenies in the R version 4.4.1 package ape (version 5.8) and visualized in Phandango.net (version 0.5.0)[76] using input files available at https://doi.org/10.5281/zenodo.18107967 (ref. 74).

## Phenotypic confirmation

Norwegian BSI isolates (n = 150) were phenotypically K typed; these were selected to represent imperfect matches to K loci with known phenotypes (n = 71), K loci where a phenotypic reference genome was not available (n = 40) and isolates that were *kps* negative (n = 39), to distinguish between G1–G4 and potentially acapsular isolates. K phenotyping was carried out by the Staten Serum Institut using reference capsular antisera. These data were used to confirm phenotypes assigned in the K typing database and to determine whether K loci without a known K phenotype expressed a putatively novel capsule, as evidenced by a positive precipitate reaction with Cetavlon but a negative reaction with known K antigens.

## *E. coli* carriage assemblies from metagenomic data

*E. coli* assemblies were extracted from metagenomic data from the UK Baby Biome data using a computational approach described in a previous study[77] to provide an *E. coli* gut colonization dataset[20,78]. The sequencing read data were first pseudo-aligned against a diverse reference database constructed from the 661,000 assemblies study[14] (available at https://zenodo.org/records/7736981 (ref. 79)) with Themisto (version 3.0.0-rc)[80]. The alignments were then processed with mSWEEP-mGEMS (version 2.0.0)[81,82] to assign each sequencing read to a bacterial species. Next, the reads assigned to *E. coli* were pseudo-aligned with Themisto against an *E. coli* database (https://doi.org/10.5281/zenodo.12528310 (ref. 83)) and reprocessed with mSWEEP and mGEMS to obtain assignments of the reads to *E. coli* Pop-PUNK (version 2.5.0) lineages[84]. The resulting assignments (bins) were quality controlled with demix_check (https://github.com/tmaklin/coreutils_demix_check/releases/tag/v0.3.2) and bins with scores of 1 or 2 were retained (n = 1,402). Reads belonging to the 1,402 bins were

assembled with Shovill (https://github.com/tseemann/shovill/releases/tag/v1.1.0) and small contigs <5,000 bp were discarded before K typing. The assemblies are available at https://zenodo.org/records/14000489 (ref. 69). As this collection sampled individuals within a family (mother and infant) multiple times during the first year of life, the isolates were deduplicated (n = 997/1,402), allowing only one lineage representative with the same K type per family. Mothers accounted for 67/997 of the isolates, and infants between 6 months and 1 year of age accounted for one-third of the deduplicated set. When both a typeable and an untypeable isolate were observed within a family for a particular lineage, we preferentially selected the K-typed isolate, as it was likely to be of the same strain but with a higher-quality assembly; this accounts for the fact that lower-coverage assemblies limit K type detection.

## Relative invasive potential

Here we used BSI isolates collected by the British Society for Antimicrobial Chemotherapy (BSAC) and carriage assemblies from metagenomic data. Both studies collected isolates in the UK[17–20]. Lineages were defined based on core and accessory distances using PopPUNK (version 2.6.5)[84]. Each lineage is named after the representative sequence type for that PopPUNK cluster and described as a clonal complex, as each lineage contains multiple sequence types. To avoid a temporal bias while estimating the relative invasive potential, due to differences in collection years between BSI (2001–2017) and carriage (2014–2017), we excluded BSIs from 2001–2002 from the analysis to remove a known expansion of the CC69 and CC131 lineages, which later reached a stable equilibrium population frequency by 2003. Furthermore, we tested for differences in proportions for each lineage and K type between the 2003–2013 and 2014–2017 periods using two-sided Fisher's exact tests, and adjusted for multiple testing. Only CC393 showed a significant difference between the periods. K loci, lineage information and the source of each isolate (BSI or carriage) were input into a generalized mixed model to estimate the relative invasive potential of K types and lineages. The model included the isolation source as a binary outcome variable, the K loci indicator variable as a fixed effect (constant for each isolate) and the lineage indicator variable as a random effect. Untypeable (for example, *kps*-negative genomes) was set as the reference category for K loci. Data were filtered to include only K loci found in more than 20 isolates, with at least five isolates in each infection and carriage group. The model was fitted in R using the glmer function in the lme4 package (version 1.1.35.5)[85] and is available at https://doi.org/10.5281/zenodo.18107967 (ref. 74). The age and sex of BSI patients, the likelihood that a strain was CTX-M positive and an increase in BSI prevalence of CC393 between 2002–2013 and 2014–2017 were possible confounders. The model was run five additional times: (1) with BSAC isolates restricted to those from adults only (n = 1,550/2,036); (2) with BSAC isolates from elderly adults only (n = 1,202/2,036); (3) using BSAC isolates with CC393 subsampled in 2003–2013 down to the same proportion as for 2014–2017 (that is, 0.02; n = 2,008/2,036); (4) adjusting for the CTX-M prevalence of each lineage in BSIs; and (5) with sex as a covariate. We further assessed whether the infant–mother metagenomic carriage data had significantly different K epidemiology from a healthy adult carriage whole-genome collection[32] sampled in France in 2010, using a non-parametric test of independence in the R package coin (version 1.4-3) for the K loci and lineages included in the invasive potential model, as well as a comparison of proportions for each individual K type and lineage included in our model between these two carriage collections with the Benjamini–Hochberg method for adjusting for multiple testing. We computed ORs with 95% CIs and P values for the top clonal complex–K combinations with at least 50 isolates in the BSAC BSI collection to assess the age–sex interaction within BSIs. This allowed us to compare the odds of a BSI patient being ≥60 years old in males versus females for each clonal complex–K combination. The ORs, CIs and P values were estimated with a two-sided Fisher's exact test.

## Lineage analysis

The top ten PopPUNK[84] lineages were mapped against a corresponding sequence type reference and aligned, then recombination was removed with Gubbins (version 3.4.3)[86,87]. The $r/m$ per lineage was estimated with Gubbins, and the MMD for each lineage was calculated using pairsnp (version 0.0.1)[88]. The Pearson correlation between $r/m$ and the K locus count per lineage, adjusted for lineage diversity using the MMD, was calculated in R version 4.4.1 using base::corr.test for the top ten lineages with >95% G2 and G3 capsules. For dating analysis, recombination-free phylogenies for CC69 and CC131 using Norwegian and UK BSI data[15,17] were created with Gubbins[87] using SKA2 (version 0.4.1)[89] and dated with the BactDating (version 1.1)[90] ARC model using three 10-million-iteration Markov Chain Monte Carlo (MCMC) replicates, alongside a randomized-dates replicate. The effective sampling size was >200, the Gelman statistic for convergence was ~1 and the true date models were better than the random date models. The resultant phylogenies were visualized in Microreact[91]. To characterize the diversity of K loci within a lineage, we used SDI 1-D, which measures the relationship between the number of species (here, either K loci or lineages), termed richness, and the number of individuals within each species, termed evenness. O:H typing was performed with SRST2 (version 0.2.0)[4].

## Inclusion and ethics

This publication used published genomic collections and their associated metadata.

## Reporting summary

Further information on research design is available in the Nature Portfolio Reporting Summary linked to this article.

## Data availability

All genomic data analysed here are public. The genomic accession numbers and metadata that support the findings of this study are available in Supplementary Data 1 and via Zenodo at https://doi.org/10.5281/zenodo.18154176 (ref. 92). Source data are provided with this paper. They are also available via Zenodo at https://doi.org/10.5281/zenodo.18349577 (ref. 93).

## Code availability

The R code used to estimate invasiveness is available via GitHub at https://doi.org/10.5281/zenodo.18107967 (ref. 74), along with a readme file detailing the code use.

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

## Acknowledgements

We thank our collaborators forming the Norwegian *E. coli* BSI Study Group: N. Handal (Akershus University Hospital), N. O. Hermansen (Oslo University Hospital, Ullevål), A. Kanestrøm (Østfold Hospital), H. E. Larsen (Nordland Hospital), P. C. Lindemann (Haukeland University Hospital), I. Høyland Löhr (Stavanger University Hospital), Å. Marvik (Vestfold Hospital), E. Nilsen (Molde Hospital and Ålesund Hospital), M. Pino (Oslo University Hospital, Rikshospitalet), E. Sirnes (Førde Hospital), S. Tofteland (Sørlandet Hospital) and K. Zaragkoulias (Nord-Trøndelag Hospital Trust). We also thank K. Holt and K. Wyres, who kindly advised on maximizing the database's utility with the Kaptive tool. Finally, we thank S. Jespersen (Statens Serum Institut) and M. Vilborg Jacobsen (Statens Serum Institut) for phenotypically K-typing 150 isolates, supported by the Gates Foundation. This project was funded by the Trond Mohn Foundation (grant identifier TMS2019TMT04 to A.K.P., R.A.G., Ø.S., P.J.J. and J.C.). The presented work has received funding from the European Union's Horizon 2020 research and innovation programme under the Marie Skłodowska-Curie Actions (grant 801133 to S.A.-A. and A.K.P.) and Wellcome Trust (grant 220540/Z/20/A to J.C., Y.S. and T.L.) and has also been supported by the European Research Council (grant 742158 to J.C.). This work was supported, in part, by the Bill and Melinda Gates Foundation (grant INV-025304). The conclusions and opinions expressed in this work are those of the author(s) alone and shall not be attributed to the Foundation.

## Author contributions

R.A.G. and J.C. conceived of the study idea, wrote the original draft of the manuscript, performed project administration and acquired funding. R.A.G., M.P., S.M. and G.T.-H. developed the methodology. R.A.G. and Ø.S. validated the results. R.A.G., M.P. and T.M. performed the formal analysis. R.A.G. and A.K.P. performed the investigation. R.A.G., M.P., T.M., Y.S., S.A.-A., B.J.P. and J.D.T. curated the data. R.A.G. and M.P. visualized the data. M.P. and N.M. designed the software. M.P., A.K.P., T.M., N.M., H.T., Y.S., S.A.-A., S.M., G.T.-H., B.J.P., J.D.T., P.J.J., N.R.T., T.L. and Ø.S. reviewed and edited the manuscript. B.J.P., J.D.T., P.J.J., N.R.T., T.L., Ø.S. and J.C. provided resources. J.C. supervised the study.

## Competing interests

The authors declare no competing interests.

## Additional information

**Extended data** is available for this paper at https://doi.org/10.1038/s41564-026-02283-w.

**Correspondence and requests for materials** should be addressed to Rebecca A. Gladstone or Jukka Corander.

[1]Department of Biostatistics, University of Oslo, Oslo, Norway. [2]Oslo Centre for Biostatistics and Epidemiology, Oslo University Hospital, Oslo, Norway. [3]Department of Mathematics and Statistics, University of Helsinki, Helsinki, Finland. [4]Parasites and Microbes, Wellcome Sanger Institute, Hinxton, UK. [5]Peter MacCallum Cancer Centre, Melbourne, Victoria, Australia. [6]Sir Peter MacCallum Department of Oncology, University of Melbourne, Melbourne, Victoria, Australia. [7]Division of Population Health and Genomics, School of Medicine, University of Dundee, Dundee, UK. [8]The National Collection of Type Cultures, Culture Collections, UK Health Security Agency, London, UK. [9]Department of Microbiology and Immunology, University of Melbourne, Peter Doherty Institute for Infection and Immunity, Melbourne, Victoria, Australia. [10]Department of Pharmacy, Faculty of Health Sciences, UiT The Arctic University of Norway, Tromsø, Norway. [11]Norwegian National Advisory Unit on Detection of Antimicrobial Resistance, Department of Microbiology and Infection Control, University Hospital of North Norway, Tromsø, Norway. [12]London School of Hygiene and Tropical Medicine, London, UK. ✉e-mail: r.a.gladstone@medisin.uio.no; jukka.corander@medisin.uio.no

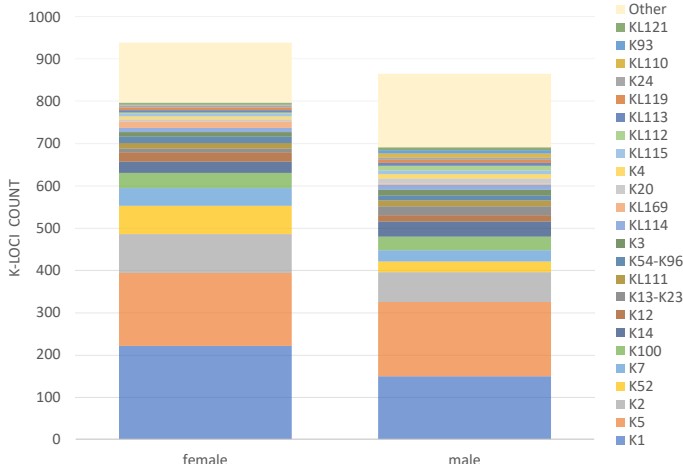

**Extended Data Fig. 1 | K type prevalence in UK BSIs with respect to sex.** UK BSI assemblies were K-typed and paired with the age and sex metadata ($n$ = 1,806). Untypeable isolates (that is, genomes without G2-G3 *kps* genes), combined with K types observed fewer than ten times, were collapsed and denoted as other.

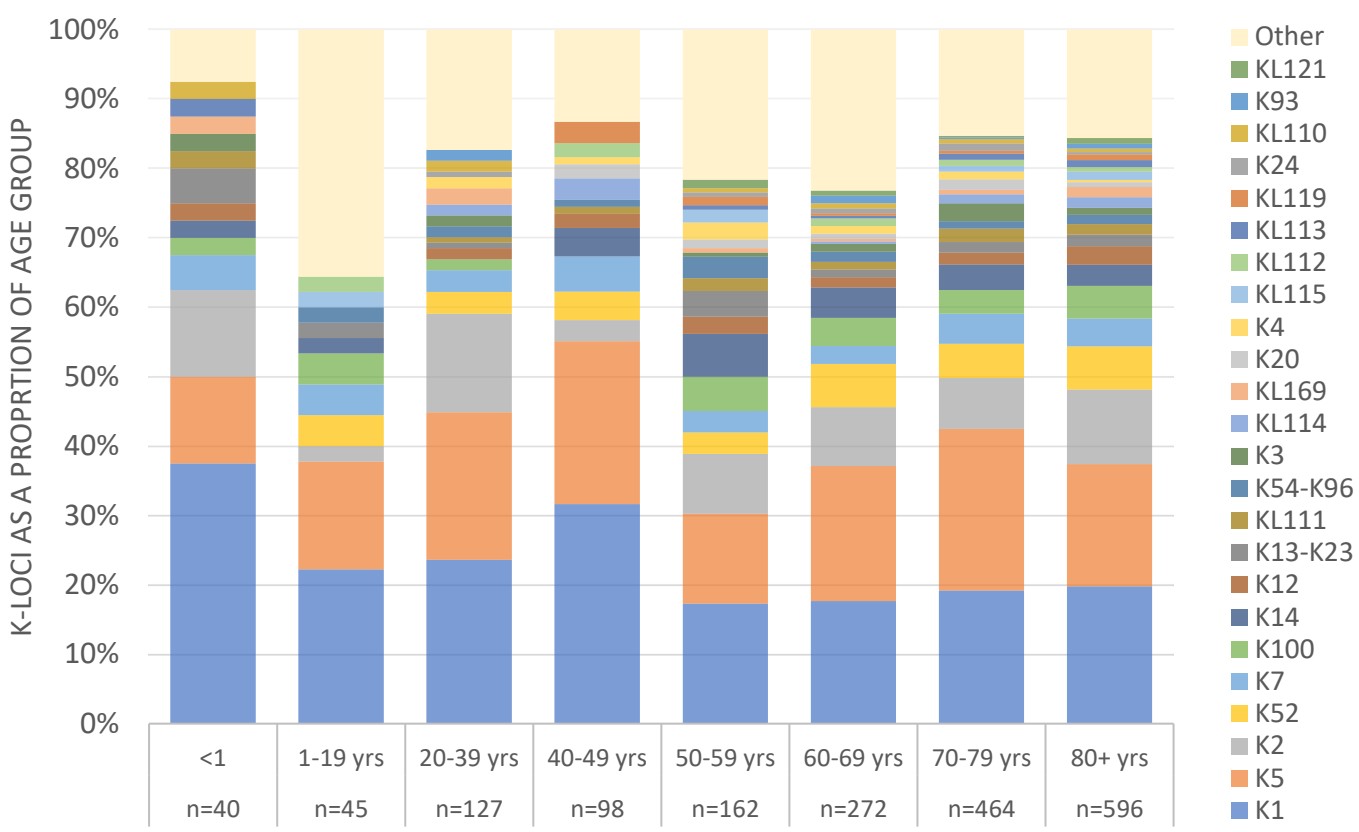

**Extended Data Fig. 2 | K type prevalence in UK BSIs with respect to age.** UK BSI assemblies were K-typed and paired with the age metadata ($n$ = 1,804). Untypeable isolates (that is, genomes without G2-G3 $kps$ genes), combined with K types observed fewer than ten times, were collapsed and denoted as other.

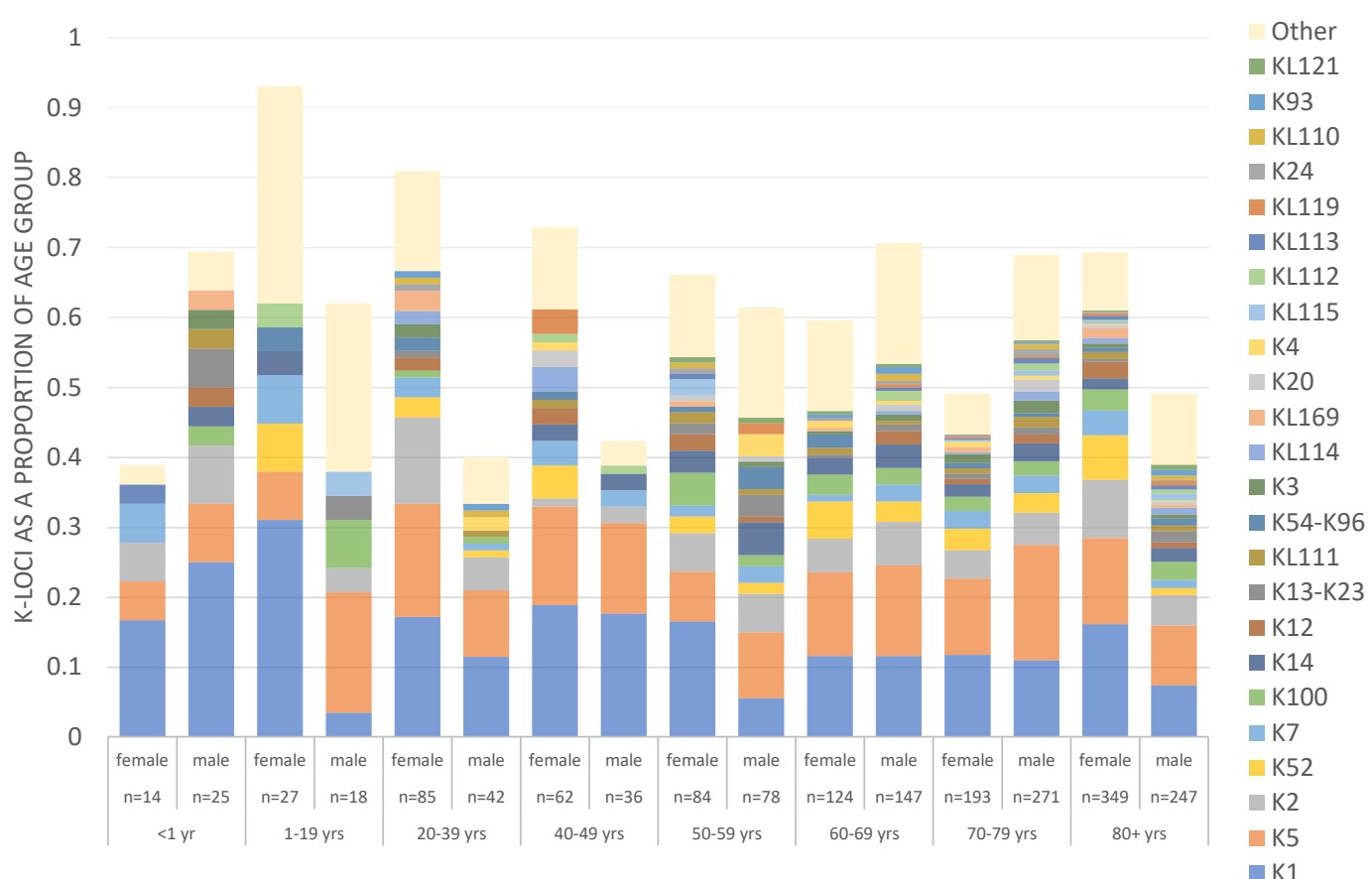

**Extended Data Fig. 3 | K type prevalence in UK BSIs with respect to sex and age.** UK BSI assemblies were K-typed and paired with the age and sex metadata ($n$ = 1,802). Untypeable isolates (that is, genomes without G2-G3 *kps* genes), combined with K types observed fewer than ten times, were collapsed and denoted as other.

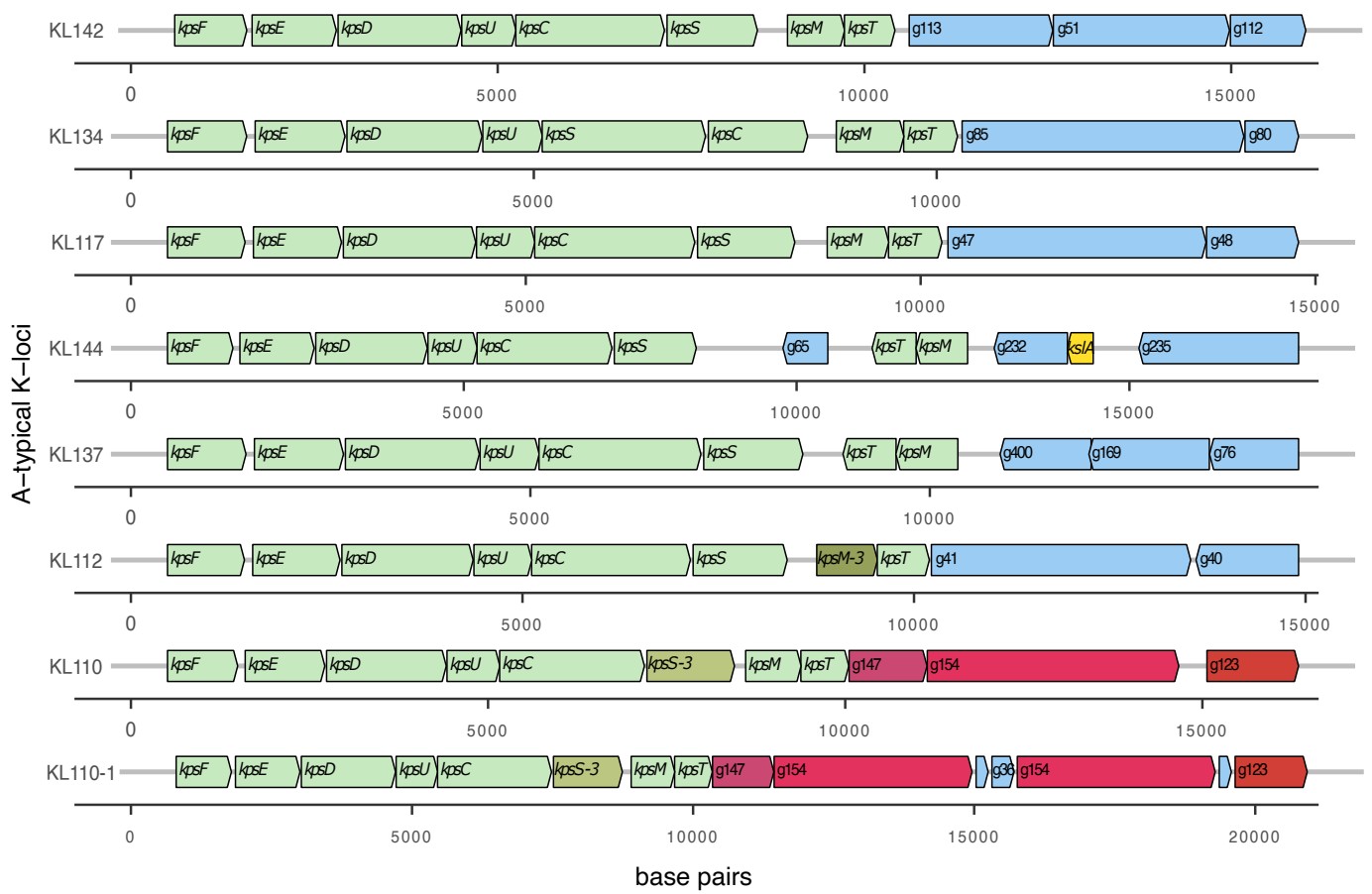

**Extended Data Fig. 4 | Genetic architecture of atypical G2 K-loci.** K-loci variants with the same capsule genes are denoted with a dash (for example, KL110-1). Green denotes *kps* gene clusters, with minor *kps* gene clusters in darker green. Yellow denotes known region 2 genes from otherwise unrelated K types. Red denotes other gene clusters that are shared between K-loci. Blue denotes other gene clusters that are not shared between K-loci. The top *E. coli* blast match is shown in brackets for unannotated gene clusters.

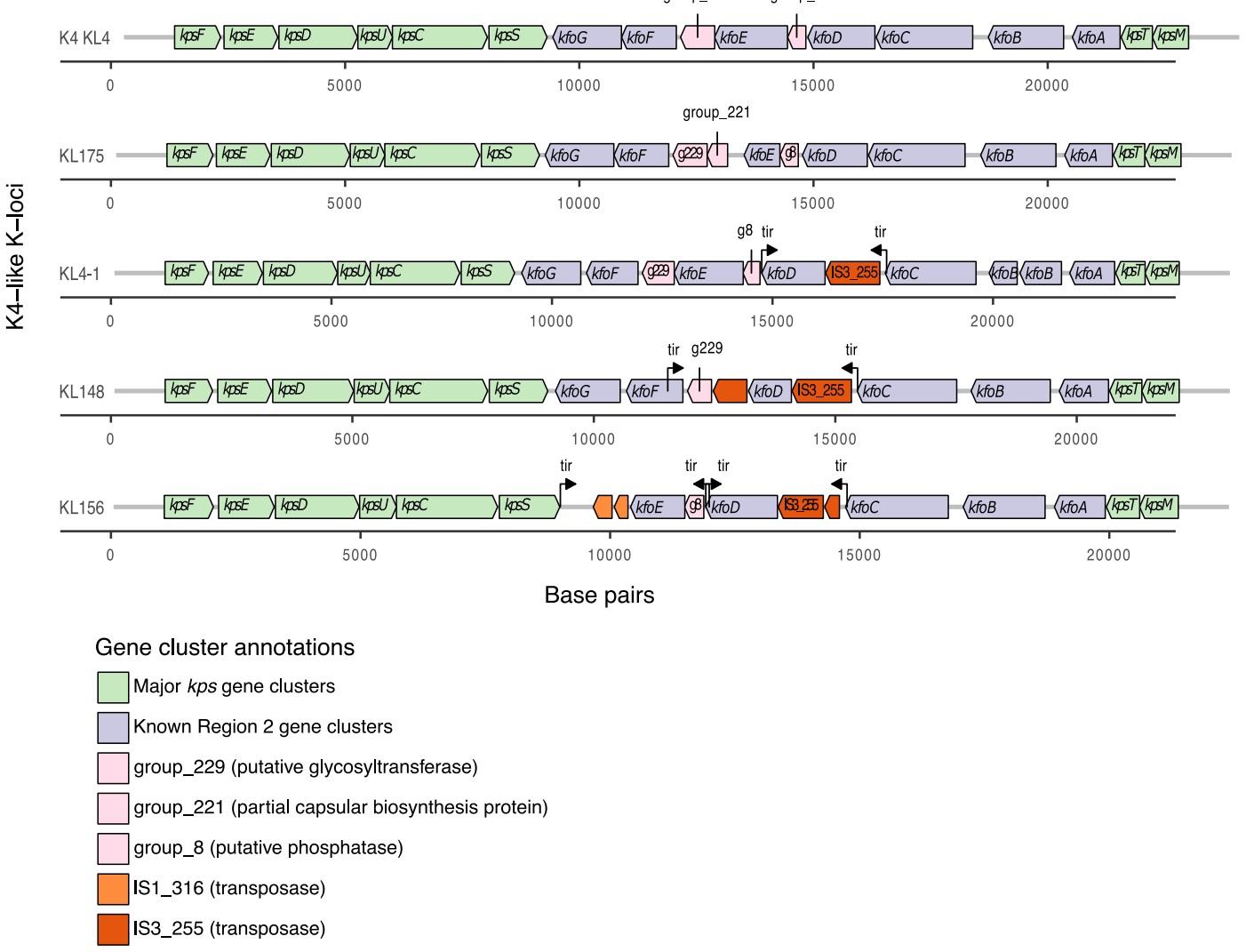

**Extended Data Fig. 5 | Insertion elements and their cargo within K4-like loci.** K-loci variants with the same capsule genes, but that vary by non-capsular genes, are denoted with a dash (for example, KL4-1). Green denotes *kps* gene clusters. Purple denotes known region 2 gene clusters. Pink denotes putative region 2 capsular genes. Red through to orange denotes insertion sequence element (IS) genes and other non-capsular genes. IS terminal inverted repeats (tir) are denoted with arrows. The top *E. coli* blast match is shown in brackets for unannotated gene clusters.

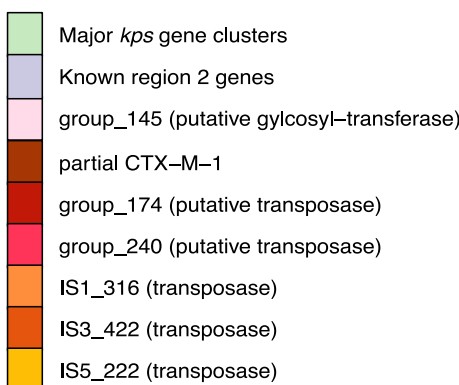

**Extended Data Fig. 6 | Insertion elements and their cargo within K5-like loci.** K-loci variants with the same capsule genes, but that vary by non-capsular genes, are denoted with a dash (for example, KL5-1). Green denotes *kps* gene clusters. Purple denotes known region 2 gene clusters. Pink denotes putative region 2 capsular genes. Red through to orange denotes insertion sequence element (IS) genes and other non-capsular genes. IS terminal inverted repeats (tir) are denoted with arrows. The top *E. coli* blast match is shown in brackets for unannotated gene clusters.

Jukka Corander

# Reporting Summary

## Statistics

For all statistical analyses, confirm that the following items are present in the figure legend, table legend, main text, or Methods section.

| n/a | Confirmed | |
|---|---|---|
| ☐ | ☒ | The exact sample size (*n*) for each experimental group/condition, given as a discrete number and unit of measurement |
| ☐ | ☒ | A statement on whether measurements were taken from distinct samples or whether the same sample was measured repeatedly |
| ☐ | ☒ | The statistical test(s) used AND whether they are one- or two-sided<br>*Only common tests should be described solely by name; describe more complex techniques in the Methods section.* |
| ☐ | ☒ | A description of all covariates tested |
| ☐ | ☒ | A description of any assumptions or corrections, such as tests of normality and adjustment for multiple comparisons |
| ☐ | ☒ | A full description of the statistical parameters including central tendency (e.g. means) or other basic estimates (e.g. regression coefficient) AND variation (e.g. standard deviation) or associated estimates of uncertainty (e.g. confidence intervals) |
| ☐ | ☒ | For null hypothesis testing, the test statistic (e.g. *F*, *t*, *r*) with confidence intervals, effect sizes, degrees of freedom and *P* value noted<br>*Give P values as exact values whenever suitable.* |
| ☒ | ☐ | For Bayesian analysis, information on the choice of priors and Markov chain Monte Carlo settings |
| ☐ | ☒ | For hierarchical and complex designs, identification of the appropriate level for tests and full reporting of outcomes |
| ☐ | ☒ | Estimates of effect sizes (e.g. Cohen's *d*, Pearson's *r*), indicating how they were calculated |

*Our web collection on statistics for biologists contains articles on many of the points above.*

## Software and code

Policy information about availability of computer code

| Data collection | We collected data using a COBS indexed database of 661k bacterial genomes (Blackwell et al 2021, https://ftp.ebi.ac.uk/pub/databases/ENA2018-bacteria-661k/ https://github.com/graceblackwell/661K_query_indexes accessed July 2023). |
|---|---|
| Data analysis | In Silico PCR https://github.com/simonrharris/in_silico_pcr<br>Bakta https://github.com/oschwengers/bakta/releases/tag/v1.11.4<br>Panaroo https://github.com/gtonkinhill/panaroo/releases/tag/v1.5.2<br>ISEscan https://github.com/xiezhq/ISEScan/releases/tag/v1.7.2.3<br>PopPUNK https://github.com/bacpop/PopPUNK/releases/tag/v2.6.5<br>Kaptive3 https://github.com/klebgenomics/Kaptive v3.0.0b5<br>Custom R v4.4.1 code: https://github.com/rgladstone/EC-K-typing/releases/tag/v3.0.0<br>Phandango https://github.com/jameshadfield/phandango/releases/tag/v0.5.0<br>Microreact https://doi.org/10.1099/mgen.0.000093<br>SRST2 https://github.com/katholt/srst2/releases/tag/v0.2.0<br>Gubbins https://github.com/nickjcroucher/gubbins/releases/tag/v3.4.3<br>pairSNP https://github.com/gtonkinhill/pairsnp/releases/tag/v0.0.1<br>BactDating https://github.com/xavierdidelot/BactDating/releases/tag/v1.1<br>SKA https://github.com/bacpop/ska.rust/releases/tag/v0.4.1<br>Mash https://github.com/marbl/Mash/releases/tag/v2.3<br>Shovill https://github.com/tseemann/shovill/releases/tag/v1.1.0<br>MSweep https://github.com/PROBIC/mSWEEP/releases/tag/v2.2.0<br>Thermisto https://github.com/algbio/themisto/releases/tag/3.0.0 |

Prokka https://github.com/tseemann/prokka/releases/tag/v1.14.5
Demix https://github.com/tmaklin/coreutils_demix_check/releases/tag/v0.3.2

For manuscripts utilizing custom algorithms or software that are central to the research but not yet described in published literature, software must be made available to editors and reviewers. We strongly encourage code deposition in a community repository (e.g. GitHub). See the Nature Portfolio guidelines for submitting code & software for further information.

## Data

Policy information about availability of data

All manuscripts must include a data availability statement. This statement should provide the following information, where applicable:
- Accession codes, unique identifiers, or web links for publicly available datasets
- A description of any restrictions on data availability
- For clinical datasets or third party data, please ensure that the statement adheres to our policy

This study used published data. Accession codes are provided for all isolates analysed in this manuscript in the Supplementary Data. The babybiome assemblies are deposited at https://zenodo.org/records/14000489).

## Research involving human participants, their data, or biological material

Policy information about studies with human participants or human data. See also policy information about sex, gender (identity/presentation), and sexual orientation and race, ethnicity and racism.

| | |
|---|---|
| Reporting on sex and gender | Sex was collected for the BSAC Bacteraemia Resistance Surveillance Programme (https://bsac.org.uk/resistance-surveillance/) from hospital records and paired with the genomic data. Sex was considered in when looking at the distribution of K-types across age groups. |
| Reporting on race, ethnicity, or other socially relevant groupings | No information on race, ethnicity or other socially relevant groupings were available. |
| Population characteristics | Isolates were collected by BSAC from all age groups. |
| Recruitment | The BSAC collection consisted of isolates submitted to a Bacteraemia Resistance Surveillance Programme (https://bsac.org.uk/resistance-surveillance/) between 2001–2017 by 11 hospitals across England. From each hospital, the first 10 isolates (when available) for each year were included into the study. |
| Ethics oversight | The BSAC collection is now housed at the University of Dundee. Advice was sought from the Senior Clinical Research Governance Manager at the Health and Clinical Services, University of Dundee who supported the use of the age and sex data. |

Note that full information on the approval of the study protocol must also be provided in the manuscript.

# Field-specific reporting

Please select the one below that is the best fit for your research. If you are not sure, read the appropriate sections before making your selection.

☒ Life sciences          ☐ Behavioural & social sciences          ☐ Ecological, evolutionary & environmental sciences

For a reference copy of the document with all sections, see nature.com/documents/nr-reporting-summary-flat.pdf

# Life sciences study design

All studies must disclose on these points even when the disclosure is negative.

| | |
|---|---|
| Sample size | Phenotyping: n=150/3254 isolates representing different K-loci were phenotyped.<br>Invasiveness: BSI n=1840 carriage n=852 |
| Data exclusions | For the invasiveness analysis: Data from disease in 2001-2002 was excluded due to large expansions in ST131 and ST69 in this time frame. Carriage data was filtered to one representative of a K-lineage within a family group to ensure independence. Data was filtered only to include K-loci found in >20 isolates, with >5 isolates of the infection and carriage groups each. |
| Replication | Phenotype concordance is reported in the supplementary Data, K-phenotyping has fallen out of use because it is laborious and subjective. Different subsamples of the invasiveness data gave the same overall findings. |
| Randomization | Invasiveness analysis used the isolation source as the experimental groups. Age and sex could not be included as a covariate so this is discussed as a limitation. |
| Blinding | There was no blinding. The study is observational. Researchers collected samples that inherently belonged to one group or the other based on the source patient's status. The "group allocation" was a fixed characteristic of the sample. |

# Reporting for specific materials, systems and methods

We require information from authors about some types of materials, experimental systems and methods used in many studies. Here, indicate whether each material, system or method listed is relevant to your study. If you are not sure if a list item applies to your research, read the appropriate section before selecting a response.

## Materials & experimental systems

| n/a | Involved in the study |
|-----|----------------------|
| ☒ ☐ | Antibodies |
| ☒ ☐ | Eukaryotic cell lines |
| ☒ ☐ | Palaeontology and archaeology |
| ☒ ☐ | Animals and other organisms |
| ☒ ☐ | Clinical data |
| ☒ ☐ | Dual use research of concern |
| ☒ ☐ | Plants |

## Methods

| n/a | Involved in the study |
|-----|----------------------|
| ☒ ☐ | ChIP-seq |
| ☒ ☐ | Flow cytometry |
| ☒ ☐ | MRI-based neuroimaging |

## Plants

| | |
|---|---|
| Seed stocks | NA |
| Novel plant genotypes | NA |
| Authentication | NA |

