## [Peer Review File · Nature Microbiology]

Identification of transporter-dependent capsular K-loci associated with invasive potential of *Escherichia coli*

Corresponding Author: Dr Rebecca Gladstone

Version 0:

Reviewer comments:

Reviewer #1

(Remarks to the Author)

This manuscript provides a clear, robust and well-conceived exploration of *E. coli* group 2 and 3 K-loci diversity, invasive potential, and evolutionary dynamics. First, the authors mined a large genome collection to create and make publicly available a curated database for in-silico typing of g2 and g3 K-loci. Second, they explored three genome collections from longitudinal sampling of BSI (Norway and UK) or carriage *E. coli* (UK only). Most importantly, this revealed that lineages, and even subclades within global MDR lineages, with specific K-types, contribute disproportionately to invasive ExPEC disease. These findings have important translational implications (Dx, Tx and alternate countermeasures, vaccines etc). I have the following suggestions/questions for the authors:

K-locus epidemiology. While K1, K5 and K52-like are consistently found in the top K-types between the two countries analyzed here, some varying K-types prevalence are observed between Norway and UK. While the manuscript briefly acknowledges this geographical limitation in the method section, briefly including contemporary BSI genomic datasets from additional regions (e.g., Asia, Africa, or South America upon availability in public databases) would allow for a more comprehensive assessment of K-loci diversity and epidemiology.

K-locus epidemiology and invasive potential. The comparison of K-loci prevalence and invasive potential between disease and carriage isolates is central to the manuscript. While age is investigated, other potential confounders (comorbidities, antibiotic exposure, local/regional outbreaks) which could influence the observed differences are not discussed. Incorporating metadata on these variables into the analysis or explicitly addressing their absence would strengthen the conclusions.

Invasive potential of K-types. The authors state that their findings "... highlight an extensive variability in the propensity of K-type and lineage to cause infection." Without experimental validation, the roles of specific K-loci in invasiveness remain an inference from a statistical model. While such analyses would significantly enhance the conclusions of this paper, simply acknowledging the need for follow up functional genomic studies would contribute to a more balanced discussion.

Role of Insertion Sequences. The findings reveal that capsule diversification is largely due to IS elements. It is also discussed that G3-B K-antigens have been observed in plasmids, including isolates of G3-B K11 from Norway (a genome collection for which hybrid assemblies are available). This search was not done for the UK genome collection with short read data available. As a possible way to expand this analysis, existing BSI plasmidome dataset from the UK (PMID: 38383544) may be used to detect K-loci plasmid carriage in clonal isolates.

Lines 129-131, 162-172 and 181-183. Should Figure 2 be cited in the text for reference?

Figure 2. It is unclear why no UK isolates were included in the phylogenies. Would including those change the inference for capsule switch events and the dating of the MRCA? Also please consider adding O and H types for both panels.

Figure 4. It is difficult to connect the text and the illustration. Please consider labelling the kps genes, providing genomic coordinates, labelling the k-loci (and not only the K-phenotypes) etc. For the region 2 gene clusters, how are genes ordered to make the presence/absence matrix?

A few typographical errors were also spotted:

Line 114 "in phylogroup in D, B2 and A."

Line 178 "homologues recombination"

Line 427 "analysis of ~75,00 high-quality *E. coli* genomes"

Reviewer #2

(Remarks to the Author)

In this manuscript, Gladstone and colleagues leverage very large *Escherichia coli* genomic datasets from several cohorts to provide a detailed catalogue of capsules belonging to group G2 and G3. More particularly, they focus on strains of *Escherichia coli* isolated from bloodstream infections. They built a curated capsule type database, with annotations and this has been formatted to be compatible with Kaptive, and easy-to-use and well-established pipeline to identify and type K-loci in other

prominent ESKAPE pathogens. This is an added value of the manuscript, and will be of use for other researchers. Given the importance of ExPEC infections, the rapid expansion of several of these lineages, acquisition of antibiotic resistance, this research is both interesting and timely.

The authors then use this 'database' to correlate the sequences with their potential for invasive disease, and provide insights into the evolution of G2 and G3 capsules. Specifically, the results suggest that capsules evolve by recombination, driven by IS.

This manuscript provides different complex analyses and, despite the large amount of work put into it, or precisely because of it, unfortunately, it is not clear what is the major question being answered or the biological advance. At times, I found the manuscript very difficult to read and follow. I hope my comments will improve the manuscript and its readability in order to render it more accessible and comprehensible. I believe that:

1. Mentioning some basic methodological procedures in the text would help with the understanding. Also, some of the methods are incomplete.
2. Another improvement would be to introduce clearly the typing scheme and to reduce the excessive capsule jargon.
3. The inclusion of numerous summary statistics in the text limits readability (maybe some can be transferred to tables?).
4. The manuscript is fairly descriptive, with no clear hypothesis from the start, which makes it difficult to follow as the reader does not know where they are being lead to. Thus, for the most of it, it reads like an exhaustive epidemiological report. It is not very clear what the ("biological") aim of this research is, nor the main message.

Major comments

1. My major concern relies on the naming scheme which makes this manuscript very complicated to follow. The authors should state clearly from the beginning what is a K type, the K-loci, and KL. Also K, KL, K-like notations are very difficult to follow. It would be appropriate to elaborate/mention/ describe the K-naming scheme from the beginning. It is very confusing at points. Most strains are identified with a K and KL numbers, which is different?. Also, the logic of naming is elusive, sometimes K is mentioned first (example L116), then KL in parenthesis. Sometimes, the opposite is true (line 175).

To complicate things further one sentence deals with Ks and the next with KL.

Are KL notations those identified by Kaptive? It is my understanding that Kaptive is a database based on *Klebsiella pneumoniae* sequences. Although it has been shown that capsules may be transferred across species, this should be introduced and commented for a broader readership to understand. That being said, is it necessary to include each KL name? maybe a supplementary table with the equivalency would be better and could alleviate the manuscript?

Also, how was the typing scheme decided/annotated? is it based on known capsules and the literature? Based on phenotypic serotype testing?

Below are some examples of some particularly confusing bits:

*Line 116, "The top five K-types causing BSIs were K1, K5, the K52-like K116 locus (KL)189, K2 and the K14-like KL137". This is very confusing, if there are 101 unique K loci (line 99), how is it that there is a KL137 and KL189?

* Also, as I understand, K1 is a G1 capsule, but is also mentioned here (line 116). This is not really clear in the text.

*it is confusing to see this K52-like KL189 (line 143). What does this mean exactly?

*Line 175: KL53 (K2-like, n=41/631), KL3 (K2, 34/631), KL137 (K14-like, n=29/631)

* Line 268: The K5 reference (KL6)

* Line 289: K4-KL29 and the K4-like KL107

*L338- "Indeed, we observe serotypes with the same capsular genes with high-sequence identities. These include K13 and K23 K-loci that are typed as KL21".

2. There are some important details in the methodology that are not sufficiently described and could bias the capsules identified. First, how was kpsF/kpsM identification performed? Was it done with nucleotide or protein sequences? Also, it is not clear how the kpsF allele to search for G3 capsules was chosen. Why particularly choose that of K54? Does changing the initial allele alter the number of G3 capsules recovered?

L442 "After extracting the kps K-loci from 21,802 assembly", how was this performed, how was the beginning and end of operon delimited?

3. Data presentation. This manuscript presents a lot of data (ratios, percentages) on the prevalence of capsules, genes, IS, etc.. These are detailed in the text which is difficult to digest. Some extra panels in figures with summary statistics and percentages would be appreciated. Also, some graphs on distributions (# of genes in different clusters, # of IS per cluster, etc..)

4. The section on K loci diversification maybe interesting for capsule and *E. coli* aficionados, but not necessarily for a broader readership as it is a catalogue of different atypical K organizations. What would be the main message here? Conceptually, capsule diversity is not exceedingly novel.

5. The study of capsule evolution is interesting, but raises some questions. Is recombination rate different across the different regions of the operon? Said otherwise, are there hotspots for recombination?

Does phylogenetic distance between variable regions correlate with distance between kps genes? Do the different kps genes share a common phylogenetic history?

6. K-loci clusters. No results are presented, yet there is a large potential in these data which goes unexplored or the text remains vague. For those clusters that have an annotated gene, it would be useful to include it in figure 4. How many genes are there in each cluster? Are most gene clusters, singletons? What is the structure of these clusters? Maybe provide a gene network would be useful. Are the different clusters linked among them, either spatially (often colocalised) in the genome or in terms of identity (ie some clusters may have members with 65% identity to others which are in another cluster)?

7. The discussion is nicely written but some parts are speculative with vague future potential implications. Indeed, sentences like “cataloguing capsular genetic diversity in contemporary disease is essential to further our understanding of K-antigens in the fight to control the increasing burden of extraintestinal pathogenic E. coli” or “diversity has major clinical implications, with large variations in the invasive potential of K-types in different genetic backgrounds”, may be true but I have difficulty in linking directly the results presented in the manuscript here these grand objectives. For example, the end of the discussion L393-415 provides specifics directly concerning the work presented. This is appreciated.

Minor comments

L92-95 “we catalogue the incredible diversity of E. coli G2 and G3 capsular loci, [...] and associations with [...] mobile genetic elements, and antimicrobial resistance (AMR)” I am not sure this has been done as stated. Were analyses of capsule types vs MGEE and AMR done for all genomes? Was this done systematically? What about phages, ICEs, etc..? No specific methods are included

Figures should be revisited as they are not color-blind friendly (specifically figure 1)

L174 is known

Figure 2 legend- Notation for K types should be explained. What would KL36_K54-K96 mean?

A more homogenous representation would help the reader (for instance same number of columns, same order, same information), for example; group membership for panel A? Keep the same K locus color in both panels for K6? Also, would a circular representation be better? Just wondering.

Figure 3B, is the correlation skewed by the two extremes (ST131, and ST69). Is the correlation still valid if these two lineages are excluded?

L332- “we observed a greater diversity of K-loci from G2 and G3 than O-loci in most dominant ExPEC”. the manuscript does not mention the number of O-loci identified in this data set, or methods to identify them.

L403 “Nevertheless, Increased”

L 457 No information is provided as to how the phylogenies were performed? On protein sequences? were these rooted? Are these trees bootstrapped? How many informative sites were taken into account? also, why neighbour-joining trees were computed as opposed to maximum likelihood?

L459-461 I don't understand

Reviewer #3

(Remarks to the Author)

Gladstone and colleagues present an analysis of previously published E. coli genomic data sets to provide insight into variation in capsule biosynthetic loci and its the associated impact on clinical infection risk. Among the major findings are: i) evidence for underappreciated diversity in capsule locus, as compared to prior serological classification, ii) evidence for capsule locus diversification in epidemic lineages and association with population expansion events and iii) evidence for the association of capsule type with risk for blood stream infection.

A major strength of this submission is the cataloging of capsule types, which has proven extremely valuable in enabling comparative genomic studies for other organisms, and a description of capsule diversity across important epidemic lineages. There are also several weaknesses. First, while intriguing, the analyses of risk associated with capsule types is severely flawed, to the point of being potentially misleading. Second, while the increased genetic diversity compared to phenotypic diversity is impressive, there is no evidence provided for functional impact associated with genetic variation. Lastly, no experimental support is provided for any claims regarding functional or clinical importance of observed variation.

Major critiques

1. A central result highlighted in the title, abstract, results and discussion is the finding that specific capsule types are associated with invasive infection. However, to say that the performed analyses are not setup to make these inferences is an understatement. Among the major issues is that the case population (i.e. bloodstream infections) is distinct from the control population (i.e. mother-infant pairs) with respect to time (cases 2003-2017, controls 2014-2017), population (i.e. healthy mothers/infants versus hospitalized patients) and clinical characteristics (i.e. likely no co-morbid conditions versus many). Note that I do not believe that these limitations can be overcome with this data, but among things that could have been done to try to allay concerns are: (i) focusing on case populations from the same time period as controls to reduce potential impact of population dynamics, (ii) focusing exclusively on mothers for controls, as host-microbe interactions and risk profiles are distinct in neonates, (iii) select cases to match the age distribution of control mothers, especially given age association with capsule/risk, (iv) limit cases to females, given the controls are all female and sex variation is reported, and (v) perform some experimental validation of key results (e.g. impact of capsule switching in lineages of interest in relevant disease models).

2. Providing the raw data underlying case-control analysis would be helpful for the reader in terms of understanding numbers of observations underlying inferences. Showing raw data in supplement (i.e. colonization versus case counts for each ST/capsule pair) would be valuable.

3. In the abstract it is stated that "paired carriage and disease cohorts" were investigated. It is misleading to say these are "paired" colonization and infection cohorts. They are in fact unrelated in time, space and population.

4. It is reported that the genotypic diversity in capsule loci is far greater than previously reported phenotypic diversity. However, this data is not clearly shown (i.e. mapping between prior serotypes and genotypes). While alluded to in several places, tabular and graphical representations of these comparisons would clarify the genotype to phenotype map (e.g. how different can capsule loci be and still be serologically equivalent, how similar can capsule loci be and be serologically distinct, etc.).

5. Several references are made to the detection of lineage expansion events associated with capsule switching. However, no statistical support is provided, nor any analyses that potential bias in data collections might have on these inferences (e.g. short-term clonal outbreaks included in collection).

Minor critiques

1. Line 117: per cent => percent

2. Line 174: know => known

3. Introduction is a bit under referenced

Decision Letter:

21st February 2025

Dear Dr Gladstone,

Thank you for your patience while your manuscript "Group 2 and 3 ABC-transporter dependant capsular K-loci contribute significantly to variation in the invasive potential of Escherichia coli" was under peer review at Nature Microbiology, and while we discussed your proposed revision plan among the team. You will find the reviewer comments at the end of this email. In light of their comments and our remaining concerns after careful consideration of the proposed revision plan, we have decided that we cannot offer to publish your manuscript in Nature Microbiology.

The referees had raised major concerns regarding the relevance of the datasets, potential confounders in the analysis looking at capsule type and invasiveness, as well as a need for further evidence to support such conclusions regarding increased invasiveness being associated with particular capsule types. Although the issue of the relevance of the datasets might be somewhat addressed by the proposed revisions, unfortunately we are not convinced that the other technical issues will be sufficiently addressed as to provide sufficient support for the key conclusions of the manuscript.

I am sorry that we cannot be more positive on this occasion, but hope that you find the referees' comments helpful when preparing your paper for resubmission elsewhere.

Although we cannot publish your paper, it may be appropriate for another journal in the Nature Portfolio. If you wish to explore the journals and transfer your manuscript please use our manuscript transfer portal. You will not have to re-supply manuscript metadata and files, unless you wish to make modifications. For more information, please see our [manuscript transfer FAQ](http://www.nature.com/authors/author_resources/transfer_manuscripts.html?WT.mc_id=EMI_NPG_1511_AUTHORTRANSF&WT.ec_id=AUTHOR) page.

Yours sincerely,

Reviewers Comments:

Reviewer #1 (Remarks to the Author):

This manuscript provides a clear, robust and well-conceived exploration of E. coli group 2 and 3 K-loci diversity, invasive potential, and evolutionary dynamics. First, the authors mined a large genome collection to create and make publicly available a curated database for in-silico typing of g2 and g3 K-loci. Second, they explored three genome collections from longitudinal sampling of BSI (Norway and UK) or carriage E. coli (UK only). Most importantly, this revealed that lineages, and even subclades within global MDR lineages, with specific K-types, contribute disproportionately to invasive ExPEC disease. These findings have important translational implications (Dx, Tx and alternate countermeasures, vaccines etc). I have the following suggestions/questions for the authors:

K-locus epidemiology. While K1, K5 and K52-like are consistently found in the top K-types between the two countries analyzed here, some varying K-types prevalence are observed between Norway and UK. While the manuscript briefly acknowledges this geographical limitation in the method section, briefly including contemporary BSI genomic datasets from additional regions (e.g., Asia, Africa, or South America upon availability in public databases) would allow for a more comprehensive assessment of K-loci diversity and epidemiology.

K-locus epidemiology and invasive potential. The comparison of K-loci prevalence and invasive potential between disease and

carriage isolates is central to the manuscript. While age is investigated, other potential confounders (comorbidities, antibiotic exposure, local/regional outbreaks) which could influence the observed differences are not discussed. Incorporating metadata on these variables into the analysis or explicitly addressing their absence would strengthen the conclusions.

Invasive potential of K-types. The authors state that their findings "... highlight an extensive variability in the propensity of K-type and lineage to cause infection." Without experimental validation, the roles of specific K-loci in invasiveness remain an inference from a statistical model. While such analyses would significantly enhance the conclusions of this paper, simply acknowledging the need for follow up functional genomic studies would contribute to a more balanced discussion.

Role of Insertion Sequences. The findings reveal that capsule diversification is largely due to IS elements. It is also discussed that G3-B K-antigens have been observed in plasmids, including isolates of G3-B K11 from Norway (a genome collection for which hybrid assemblies are available). This search was not done for the UK genome collection with short read data available. As a possible way to expand this analysis, existing BSI plasmidome dataset from the UK (PMID: 38383544) may be used to detect K-loci plasmid carriage in clonal isolates.

Lines 129-131, 162-172 and 181-183. Should Figure 2 be cited in the text for reference?

Figure 2. It is unclear why no UK isolates were included in the phylogenies. Would including those change the inference for capsule switch events and the dating of the MRCA? Also please consider adding O and H types for both panels.

Figure 4. It is difficult to connect the text and the illustration. Please consider labelling the kps genes, providing genomic coordinates, labelling the k-loci (and not only the K-phenotypes) etc. For the region 2 gene clusters, how are genes ordered to make the presence/absence matrix?

A few typographical errors were also spotted:

Line 114 "in phylogroup in D, B2 and A."

Line 178 "homologues recombination"

Line 427 "analysis of ~75,00 high-quality E. coli genomes"

Reviewer #2 (Remarks to the Author):

In this manuscript, Gladstone and colleagues leverage very large Escherichia coli genomic datasets from several cohorts to provide a detailed catalogue of capsules belonging to group G2 and G3. More particularly, they focus on strains of Escherichia coli isolated from bloodstream infections. They built a curated capsule type database, with annotations and this has been formatted to be compatible with Kaptive, and easy-to-use and well-established pipeline to identify and type K-loci in other prominent ESKAPE pathogens. This is an added value of the manuscript, and will be of use for other researchers. Given the importance of ExPEC infections, the rapid expansion of several of these lineages, acquisition of antibiotic resistance, this research is both interesting and timely.

The authors then use this 'database' to correlate the sequences with their potential for invasive disease, and provide insights into the evolution of G2 and G3 capsules. Specifically, the results suggest that capsules evolve by recombination, driven by IS.

This manuscript provides different complex analyses and, despite the large amount of work put into it, or precisely because of it, unfortunately, it is not clear what is the major question being answered or the biological advance. At times, I found the manuscript very difficult to read and follow. I hope my comments will improve the manuscript and its readability in order to render it more accessible and comprehensible. I believe that:

1. Mentioning some basic methodological procedures in the text would help with the understanding. Also, some of the methods are incomplete.
2. Another improvement would be to introduce clearly the typing scheme and to reduce the excessive capsule jargon.
3. The inclusion of numerous summary statistics in the text limits readability (maybe some can be transferred to tables?).
4. The manuscript is fairly descriptive, with no clear hypothesis from the start, which makes it difficult to follow as the reader does not know where they are being lead to. Thus, for the most of it, it reads like an exhaustive epidemiological report. It is not very clear what the ("biological") aim of this research is, nor the main message.

Major comments

1. My major concern relies on the naming scheme which makes this manuscript very complicated to follow. The authors should state clearly from the beginning what is a K type, the K-loci, and KL. Also K, KL, K-like notations are very difficult to follow. It would be appropriate to elaborate/mention/ describe the K-naming scheme from the beginning. It is very confusing at points. Most strains are identified with a K and KL numbers, which is different?. Also, the logic of naming is elusive, sometimes K is mentioned first (example L116), then KL in parenthesis. Sometimes, the opposite is true (line 175).

To complicate things further one sentence deals with Ks and the next with KL.

Are KL notations those identified by Kaptive? It is my understanding that Kaptive is a database based on Klebsiella pneumoniae sequences. Although it has been shown that capsules may be transferred across species, this should be introduced and commented for a broader readership to understand. That being said, is it necessary to include each KL name? maybe a supplementary table with the equivalency would be better and could alleviate the manuscript?

Also, how was the typing scheme decided/annotated? is it based on known capsules and the literature? Based on phenotypic serotype testing?

Below are some examples of some particularly confusing bits:

*Line 116, "The top five K-types causing BSIs were K1, K5, the K52-like K116 locus (KL)189, K2 and the K14-like KL137". This is very confusing, if there are 101 unique K loci (line 99), how is it that there is a KL137 and KL189?

* Also, as I understand, K1 is a G1 capsule, but is also mentioned here (line 116). This is not really clear in the text.

*it is confusing to see this K52-like KL189 (line 143). What does this mean exactly?

*Line 175: KL53 (K2-like, n=41/631), KL3 (K2, 34/631), KL137 (K14-like, n=29/631)

* Line 268: The K5 reference (KL6)

* Line 289: K4-KL29 and the K4-like KL107

*L338- "Indeed, we observe serotypes with the same capsular genes with high-sequence identities. These include K13 and K23 K-loci that are typed as KL21".

2. There are some important details in the methodology that are not sufficiently described and could bias the capsules identified. First, how was kpsF/kpsM identification performed? Was it done with nucleotide or protein sequences? Also, it is not clear how the kpsF allele to search for G3 capsules was chosen. Why particularly choose that of K54? Does changing the initial allele alter the number of G3 capsules recovered?

L442 "After extracting the kps K-loci from 21,802 assembly", how was this performed, how was the beginning and end of operon delimited?

3.Data presentation. This manuscript presents a lot of data (ratios, percentages) on the prevalence of capsules, genes, IS, etc.. These are detailed in the text which is difficult to digest. Some extra panels in figures with summary statistics and percentages would be appreciated. Also, some graphs on distributions (# of genes in different clusters, # of IS per cluster, etc..)

4.The section on K loci diversification maybe interesting for capsule and E. coli aficionados, but not necessarily for a broader readership as it is a catalogue of different atypical K organizations. What would be the main message here? Conceptually, capsule diversity is not exceedingly novel.

5. The study of capsule evolution is interesting, but raises some questions. Is recombination rate different across the different regions of the operon? Said otherwise, are there hotspots for recombination? Does phylogenetic distance between variable regions correlate with distance between kps genes? Do the different kps genes share a common phylogenetic history?

6.K-loci clusters. No results are presented, yet there is a large potential in these data which goes unexplored or the text remains vague. For those clusters that have an annotated gene, it would be useful to include it in figure 4. How many genes are there in each cluster? Are most gene clusters, singletons? What is the structure of these clusters? Maybe provide a gene network would be useful. Are the different clusters linked among them, either spatially (often colocalised) in the genome or in terms of identity (ie some clusters may have members with 65% identity to others which are in another cluster)?

7. The discussion is nicely written but some parts are speculative with vague future potential implications. Indeed, sentences like "cataloguing capsular genetic diversity in contemporary disease is essential to further our understanding of K-antigens in the fight to control the increasing burden of extraintestinal pathogenic E. coli" or "diversity has major clinical implications, with large variations in the invasive potential of K-types in different genetic backgrounds", may be true but I have difficulty in linking directly the results presented in the manuscript here these grand objectives. For example, the end of the discussion L393-415 provides specifics directly concerning the work presented. This is appreciated.

Minor comments

L92-95 "we catalogue the incredible diversity of E. coli G2 and G3 capsular loci, [...] and associations with [...] mobile genetic elements, and antimicrobial resistance (AMR)" I am not sure this has been done as stated. Were analyses of capsule types vs MGEE and AMR done for all genomes? Was this done systematically? What about phages,ICEs, etc..? No specific methods are included

Figures should be revisited as they are not color-blind friendly (specifically figure 1)

L174 is known

Figure 2 legend- Notation for K types should be explained. What would KL36_K54-K96 mean?

A more homogenous representation would help the reader (for instance same number of columns, same order, same information), for example; group membership for panel A? Keep the same K locus color in both panels for K6?

Also, would a circular representation be better? Just wondering.

Figure 3B, is the correlation skewed by the two extremes (ST131, and ST69). Is the correlation still valid if these two lineages are excluded?

L332- "we observed a greater diversity of K-loci from G2 and G3 than O-loci in most dominant ExPEC". the manuscript does not mention the number of O-loci identified in this data set, or methods to identify them.

L403 "Nevertheless, Increased"

L 457 No information is provided as to how the phylogenies were performed? On protein sequences? were these rooted? Are these trees bootstrapped? How many informative sites were taken into account? also, why neighbour-joining trees were computed as opposed to maximum likelihood?

L459-461 I don't understand

Reviewer #3 (Remarks to the Author):

Gladstone and colleagues present an analysis of previously published *E. coli* genomic data sets to provide insight into variation in capsule biosynthetic loci and its the associated impact on clinical infection risk. Among the major findings are: i) evidence for underappreciated diversity in capsule locus, as compared to prior serological classification, ii) evidence for capsule locus diversification in epidemic lineages and association with population expansion events and iii) evidence for the association of capsule type with risk for blood stream infection.

A major strength of this submission is the cataloging of capsule types, which has proven extremely valuable in enabling comparative genomic studies for other organisms, and a description of capsule diversity across important epidemic lineages. There are also several weaknesses. First, while intriguing, the analyses of risk associated with capsule types is severely flawed, to the point of being potentially misleading. Second, while the increased genetic diversity compared to phenotypic diversity is impressive, there is no evidence provided for functional impact associated with genetic variation. Lastly, no experimental support is provided for any claims regarding functional or clinical importance of observed variation.

Major critiques

1. A central result highlighted in the title, abstract, results and discussion is the finding that specific capsule types are associated with invasive infection. However, to say that the performed analyses are not setup to make these inferences is an understatement. Among the major issues is that the case population (i.e. bloodstream infections) is distinct from the control population (i.e. mother-infant pairs) with respect to time (cases 2003-2017, controls 2014-2017), population (i.e. healthy mothers/infants versus hospitalized patients) and clinical characteristics (i.e. likely no co-morbid conditions versus many). Note that I do not believe that these limitations can be overcome with this data, but among things that could have been done to try to allay concerns are: (i) focusing on case populations from the same time period as controls to reduce potential impact of population dynamics, (ii) focusing exclusively on mothers for controls, as host-microbe interactions and risk profiles are distinct in neonates, (iii) select cases to match the age distribution of control mothers, especially given age association with capsule/risk, (iv) limit cases to females, given the controls are all female and sex variation is reported, and (v) perform some experimental validation of key results (e.g. impact of capsule switching in lineages of interest in relevant disease models).

2. Providing the raw data underlying case-control analysis would be helpful for the reader in terms of understanding numbers of observations underlying inferences. Showing raw data in supplement (i.e. colonization versus case counts for each ST/capsule pair) would be valuable.

3. In the abstract it is stated that "paired carriage and disease cohorts" were investigated. It is misleading to say these are "paired" colonization and infection cohorts. They are in fact unrelated in time, space and population.

4. It is reported that the genotypic diversity in capsule loci is far greater than previously reported phenotypic diversity. However, this data is not clearly shown (i.e. mapping between prior serotypes and genotypes). While alluded to in several places, tabular and graphical representations of these comparisons would clarify the genotype to phenotype map (e.g. how different can capsule loci be and still be serologically equivalent, how similar can capsule loci be and be serologically distinct, etc.).

5. Several references are made to the detection of lineage expansion events associated with capsule switching. However, no statistical support is provided, nor any analyses that potential bias in data collections might have on these inferences (e.g. short-term clonal outbreaks included in collection).

Minor critiques

1. Line 117: per cent => percent

2. Line 174: know => known

3. Introduction is a bit under referenced

Version 1:

Decision Letter:

10th April 2025

Dear Dr Gladstone,

Thank you for your letter asking us to reconsider our decision on your Article entitled "Group 2 and 3 ABC-transporter dependant capsular K-loci contribute significantly to variation in the invasive potential of *Escherichia coli*".

I have discussed the matter carefully with my colleagues, although we cannot reverse our decision at this stage without having seen the new data, we would encourage you to submit a revised paper with accompanying point-by-point response to all of the reviewers' comments. We are not overruling a need for functional work at this point, as it was made by several referees who both had expertise in pathogenomics and genomic epidemiology for bacterial pathogens. In addition, we would not recruit a new referee, as the expertise was covered already across several of the referees. However, we would be willing to potentially reconsider our decision upon seeing the revised paper and send it back to the original reviewers, if the rebuttal addresses all of

the reviewer concerns.

Best wishes,

Version 2:

Reviewer comments:

Reviewer #1

(Remarks to the Author)

The authors have thoroughly replied to my suggestions and questions.

The manuscript has been revised accordingly, including: i) a new Fig. 1 with a broader geographical sampling, ii) a clarification that the in silico approach only provides an estimated invasive potential of E. coli, iii) a reference to the existing literature for the experimental verification of the virulence of many E. coli capsule types, iv) an analysis of K-loci prevalence accounting for patient age and resistance type (MDR/ESBL vs non), v) a revised Figure 3 (previously #2) to include all relevant isolates, and vi) a revised Figure 5 with more labels to guide the reader.

I find the revised manuscript to be clear, the results to be novel and the conclusions well supported by the data. I have no further questions/suggestions for the authors.

(Remarks on code availability)

Code already publicly available.

README file available and useful.

Successful install and test run.

Reviewer #2

(Remarks to the Author)

In this revised version, Gladstone and colleagues have substantially strengthened their study. The inclusion of larger datasets confirm their previously observed results and reduces potential sampling biases. The new analyses provide additional insights. Furthermore, the authors have undertaken an extensive reformulation of several sections in direct response to reviewer feedback. This improves the clarity of the work and allows the impact of the results to be more fully appreciated, particularly in light of the expanded discussion on study limitations.

The standardization of the nomenclature has also been improved and is now much clearer overall. However, it remains somewhat counterintuitive that the K and KL designations do not always correspond to one another, as these labels are determined by algorithmic assignment rather than human curation, particularly in cases where a K-phenotype is already associated.

Overall, the manuscript is now considerably easier to read and interpret, and the revisions have enhanced both the scientific rigor and the accessibility of the study.

(Remarks on code availability)

I did not properly review the code, but I had a look. It is organized and sufficiently commented. Someone with R knowledge should be able to easily read through it.

There is a README file. I did not install anything (mostly R scripts do not require installations and commonly used packages). I did not run the code

Reviewer #3

(Remarks to the Author)

I thank the authors for addressing many of the key reviewer comments. However, I still have concerns regarding the analyses that are being used to infer that the invasive potential of E. coli strains can be directly attributed to specific K-types. The authors do a reasonable job justifying their use of the infant/mother cohort as a proxy for colonization among healthy adults through comparison of strain distributions to other cohorts. However, I am not convinced that a population of healthy adults is a proper control population to compare to hospitalized patients with BSIs. The ideal control population would be a propensity-score matched cohort of colonized patients who did not proceed to BSI. While I appreciate the author's point that this is a logistically challenging cohort to assemble given the relative rarity of the infection, this does not mitigate concerns over not controlling for relevant factors (i.e. age, sex, antibiotic use, comorbid conditions, etc.), while claiming a direct impact of the variable of interest. I would ask at a minimum for the following:

1) Properly controlling for age and sex, which seem to be variables that are available for both cases and controls, by adjusting for these variables in their models. I believe the authors have performed analyses to attempt to control for age, but in the methods it seems they did this by restricting to older adults in the BSI cohort. However, the BSI cohort is already enriched in older adults, so I was unclear on how this controlled for the age-bias in cases.

2) Tempering language to reflect the associative nature of these findings, given that this was an observational study design with poorly matched populations. Here are some examples, where I find that the analysis does not support the conclusions:

(i) "Estimating the relative invasive potential allowed us to put these experimental findings into a wider context by ranking the capsular types in terms of their propensity to be found in BSIs, given the exposure frequency in asymptomatic colonisation."

In fact, with this study design you do not know the exposure frequencies in the BSI population, and whether they match the healthy child/adult cohorts.

(ii) "However, 74% of BSIs are in elderly patients of greater than 59 years, and so the estimates are likely to better represent this important high-risk age group."

The direct attributable risk associated with capsule cannot be inferred in elderly cohorts, as you have not presented data on colonization frequencies in this cohort, nor controlled for other relevant variables. In fact, you may be overestimating the risk of some capsule types in the elderly cohort if they are present at higher frequencies in asymptomatic patients that are properly matched on age, sex and other relevant factors.

What you can say is that certain capsular types are enriched in BSI, as compared to the general healthy adult population, which may have its own important implications. However, this is different than inferring a causal role of specific K-types in the risk of colonized patients developing BSIs, which is what is being inferred/implied/stated. Importantly, it may be that with a properly controlled study that the findings would remain the same, but it does not seem to me that this is a justifiable conclusion to draw from the data presented.

(Remarks on code availability)

Decision Letter:

21st October 2025

Dear Dr Gladstone,

Thank you for your patience while your manuscript "Group 2 and 3 ABC-transporter-dependent capsular K-loci contribute significantly to variation in the estimated invasive potential of *Escherichia coli*" was under peer-review at Nature Microbiology. It has now been seen by the original 3 referees, whose expertise and comments you will find at the of this email. You will see from their comments below that while they find the study has improved and many of the previous concerns have been addressed by the revisions, there are a few remaining points to address. We are very interested in the possibility of publishing your study in Nature Microbiology, but would like to consider your response to these concerns in the form of a revised manuscript before we make a final decision on publication.

In particular, you will see that Referee #3 still has concerns about use of the mother-infant colonisation control, but also notes the difficulty in obtaining the more appropriate control group (colonised patients who don't progress to BSI). This reviewer suggests some further adjustments to the analyses to deal with potential confounding effects due to age and sex. We would also suggest discussing this as a limitation. In addition, the referee makes some suggestions for a few minor text edits in line with these limitations. A revised manuscript needs to address all of these points.

If you have not done so already please begin to revise your manuscript so that it conforms to our Article format instructions at <http://www.nature.com/nmicrobiol/info/final-submission/>

The usual length limit for a Nature Microbiology Article is six display items (figures or tables) and 3,500 words. We have some flexibility, upto 4,500 words.

We strongly support public availability of data. If you have not already done so, please place the data used in your paper into a public data repository, if one exists, or alternatively, present the data as Source Data or Supplementary Information. If data can only be shared on request, please explain why in your Data Availability Statement, and also in the correspondence with your editor. For some data types, deposition in a public repository is mandatory - more information on our data deposition policies and available repositories can be found at <https://www.nature.com/nature-research/editorial-policies/reporting-standards#availability-of-data>.

Please include a data availability statement as a separate section after Methods but before references, under the heading "Data Availability". This section should inform readers about the availability of the data used to support the conclusions of your study. This information includes accession codes to public repositories (data banks for protein, DNA or RNA sequences, microarray, proteomics data etc...), references to source data published alongside the paper, unique identifiers such as URLs to data repository entries, or data set DOIs, and any other statement about data availability. At a minimum, you should include the following statement: "The data that support the findings of this study are available from the corresponding author upon request", mentioning any restrictions on availability. If DOIs are provided, we also strongly encourage including these in the Reference list

(authors, title, publisher (repository name), identifier, year). For more guidance on how to write this section please see: <http://www.nature.com/authors/policies/data/data-availability-statements-data-citations.pdf>

To improve the accessibility of your paper to readers from other research areas, please pay particular attention to the wording of the paper's opening bold paragraph, which serves both as an introduction and as a brief, non-technical summary in about 150 words. If, however, you require one or two extra sentences to explain your work clearly, please include them even if the paragraph is over-length as a result. The opening paragraph should not contain references. Because scientists from other sub-disciplines will be interested in your results and their implications, it is important to explain essential but specialised terms concisely. We suggest you show your summary paragraph to colleagues in other fields to uncover any problematic concepts.

If your paper is accepted for publication, we will edit your display items electronically so they conform to our house style and will reproduce clearly in print. If necessary, we will re-size figures to fit single or double column width. If your figures contain several parts, the parts should form a neat rectangle when assembled. Choosing the right electronic format at this stage will speed up the processing of your paper and give the best possible results in print. We would like the figures to be supplied as vector files - EPS, PDF, AI or postscript (PS) file formats (not raster or bitmap files), preferably generated with vector-graphics software (Adobe Illustrator for example). Please try to ensure that all figures are non-flattened and fully editable. All images should be at least 300 dpi resolution (when figures are scaled to approximately the size that they are to be printed at) and in RGB colour format. Please do not submit Jpeg or flattened TIFF files. Please see also 'Guidelines for Electronic Submission of Figures' at the end of this letter for further detail.

Figure legends must provide a brief description of the figure and the symbols used, within 350 words, including definitions of any error bars employed in the figures.

When submitting the revised version of your manuscript, please pay close attention to our [href="https://www.nature.com/nature-research/editorial-policies/image-integrity">Digital Image Integrity Guidelines. and to the following points below:](https://www.nature.com/nature-research/editorial-policies/image-integrity)

EXTENDED DATA FIGURES

Please include a statement before the acknowledgements naming the author to whom correspondence and requests for materials should be addressed.

Finally, we require authors to include a statement of their individual contributions to the paper -- such as experimental work, project planning, data analysis, etc. -- immediately after the acknowledgements. The statement should be short, and refer to authors by their initials. For details please see the Authorship section of our joint Editorial policies at http://www.nature.com/authors/editorial_policies/authorship.html

* include a point-by-point response to any editorial suggestions and to our referees. Please include your response to the editorial suggestions in your cover letter, and please upload your response to the referees as a separate document.

* ensure it complies with our format requirements for Letters as set out in our guide to authors at www.nature.com/nmicrobiol/info/gta/

* state in a cover note the length of the text, methods and legends; the number of references; number and estimated final size of figures and tables

* resubmit electronically if possible using the link below to access your home page:

Link Redacted

*This url links to your confidential homepage and associated information about manuscripts you may have submitted or be reviewing for us. If you wish to forward this e-mail to co-authors, please delete this link to your homepage first.

Please ensure that all correspondence is marked with your Nature Microbiology reference number in the subject line.

Nature Microbiology is committed to improving transparency in authorship. As part of our efforts in this direction, we are now requesting that all authors identified as 'corresponding author' on published papers create and link their Open Researcher and Contributor Identifier (ORCID) with their account on the Manuscript Tracking System (MTS), prior to acceptance. This applies to primary research papers only. ORCID helps the scientific community achieve unambiguous attribution of all scholarly contributions. You can create and link your ORCID from the home page of the MTS by clicking on 'Modify my Springer Nature account'. For more information please visit www.springernature.com/orcid.

We hope to receive your revised paper within three weeks. If you cannot send it within this time, please let us know.

Yours sincerely,

Reviewer Expertise:

Referee #1: genetics, bioinformatics, epidemiology

Referee #2: capsule

Referee #3: microbial pathogenomics, evolution, AMR, infectious disease

Reviewers Comments:

Reviewer #1 (Remarks to the Author):

The authors have thoroughly replied to my suggestions and questions.

The manuscript has been revised accordingly, including: i) a new Fig. 1 with a broader geographical sampling, ii) a clarification that the in silico approach only provides an estimated invasive potential of E. coli, iii) a reference to the existing literature for the experimental verification of the virulence of many E. coli capsule types, iv) an analysis of K-loci prevalence accounting for patient age and resistance type (MDR/ESBL vs non), v) a revised Figure 3 (previously #2) to include all relevant isolates, and vi) a revised Figure 5 with more labels to guide the reader.

I find the revised manuscript to be clear, the results to be novel and the conclusions well supported by the data. I have no further questions/suggestions for the authors.

Reviewer #1 (Remarks on code availability):

Code already publicly available.

README file available and useful.

Successful install and test run.

Reviewer #2 (Remarks to the Author):

In this revised version, Gladstone and colleagues have substantially strengthened their study. The inclusion of larger datasets confirm their previously observed results and reduces potential sampling biases. The new analyses provide additional insights. Furthermore, the authors have undertaken an extensive reformulation of several sections in direct response to reviewer feedback. This improves the clarity of the work and allows the impact of the results to be more fully appreciated, particularly in light of the expanded discussion on study limitations.

The standardization of the nomenclature has also been improved and is now much clearer overall. However, it remains somewhat counterintuitive that the K and KL designations do not always correspond to one another, as these labels are determined by algorithmic assignment rather than human curation, particularly in cases where a K-phenotype is already associated.

Overall, the manuscript is now considerably easier to read and interpret, and the revisions have enhanced both the scientific rigor and the accessibility of the study.

Reviewer #2 (Remarks on code availability):

I did not properly review the code, but I had a look. It is organized and sufficiently commented. Someone with R knowledge should be able to easily read through it.

There is a README file. I did not install anything (mostly R scripts do not require installations and commonly used packages). I did not run the code

Reviewer #3 (Remarks to the Author):

I thank the authors for addressing many of the key reviewer comments. However, I still have concerns regarding the analyses that are being used to infer that the invasive potential of E. coli strains can be directly attributed to specific K-types. The authors do a reasonable job justifying their use of the infant/mother cohort as a proxy for colonization among healthy adults through comparison of strain distributions to other cohorts. However, I am not convinced that a population of healthy adults is a proper control population to compare to hospitalized patients with BSIs. The ideal control population would be a propensity-score matched cohort of colonized patients who did not proceed to BSI. While I appreciate the author's point that this is a logistically challenging cohort to assemble given the relative rarity of the infection, this does not mitigate concerns over not controlling for relevant factors (i.e. age, sex, antibiotic use, comorbid conditions, etc.), while claiming a direct impact of the variable of interest. I would ask at a minimum for the following:

1) Properly controlling for age and sex, which seem to be variables that are available for both cases and controls, by adjusting for these variables in their models. I believe the authors have performed analyses to attempt to control for age, but in the methods it seems they did this by restricting to older adults in the BSI cohort. However, the BSI cohort is already enriched in older adults, so I was unclear on how this controlled for the age-bias in cases.

2) Tempering language to reflect the associative nature of these findings, given that this was an observational study design with poorly matched populations. Here are some examples, where I find that the analysis does not support the conclusions:

(i) "Estimating the relative invasive potential allowed us to put these experimental findings into a wider context by ranking the capsular types in terms of their propensity to be found in BSIs, given the exposure frequency in asymptomatic colonisation."

In fact, with this study design you do not know the exposure frequencies in the BSI population, and whether they match the healthy child/adult cohorts.

(ii) "However, 74% of BSIs are in elderly patients of greater than 59 years, and so the estimates are likely to better represent this important high-risk age group."

The direct attributable risk associated with capsule cannot be inferred in elderly cohorts, as you have not presented data on colonization frequencies in this cohort, nor controlled for other relevant variables. In fact, you may be overestimating the risk of some capsule types in the elderly cohort if they are present at higher frequencies in asymptomatic patients that are properly matched on age, sex and other relevant factors.

What you can say is that certain capsular types are enriched in BSI, as compared to the general healthy adult population, which may have its own important implications. However, this is different than inferring a causal role of specific K-types in the risk of colonized patients developing BSIs, which is what is being inferred/implied/stated. Importantly, it may be that with a properly controlled study that the findings would remain the same, but it does not seem to me that this is a justifiable conclusion to draw from the data presented.

Version 3:

Reviewer comments:

Reviewer #3

(Remarks to the Author)

I appreciate the authors substantially addressing my concerns by adding significantly to the discussion. I think the paragraph added adequately addresses my concerns. However, I have two remaining concerns:

1) I may just be misunderstanding, but I was not able to follow how the sub-analyses to assess the issue of age mismatch between cases and controls alleviate the concern. In particular:

"first, we found that the odds of a BSI patient being ≥ 60 in males compared to females were not significant for most lineage-capsule combinations, except for CC131-C2 with K5."

=> This mitigates sex as a confounder, but I don't see how this mitigates age.

"Secondly, as we could not explicitly include age as a covariate, we highlight that the elderly adult high-risk age group (≥ 60 years) is overrepresented in the data (74% of BSIs) and, therefore, that the estimates are most relevant for this important clinical population."

=> This is quite circular. The issue is that the over-representation of elderly individuals in the BSI groups makes it impossible to determine the direct attributable risk of capsule type to BSI. The over-representation doesn't address the concern, it is the concern.

"Furthermore, we ran additional iterations of the model, excluding the other age groups, to remove the assumption that the invasive potential is homogeneous across age groups and limit it to the simpler assumption for which we have some evidence: that carriage is homogeneous in the healthy population."

=> If I understand correctly, this analysis just limited the BSIs to >60 , not the controls? I don't follow the logic for how this mitigates concern regarding age as a confounder. The concern is that carriage is not homogeneous between healthy and BSI populations, potentially due to age itself (e.g. altered gut microbiota, altered immune status) or due to altered exposure patterns

(e.g. frequent healthcare exposure).

To me, it would be preferable to just acknowledge the limitation, instead of trying to work/talk around structural issues in the data.

2) While the discussion softens language and acknowledges that the study design can only yield associations, the title and the abstract don't have this nuance. Given the key roles of the title and the abstract in conveying the central message to non-domain experts, especially in a high-impact journal, I think more effort should go into softening the language/implications to accurately reflect the data/analysis performed.

(Remarks on code availability)

Decision Letter:

Our ref: NMICROBIOL-24123762C

12th December 2025

Dear Dr. Gladstone,

Thank you for submitting your revised manuscript "Group 2 and 3 ABC-transporter-dependent capsular K-loci contribute significantly to variation in the estimated invasive potential of *Escherichia coli*" (NMICROBIOL-24123762C). It has now been seen by the original referees and their comments are below. The reviewers find that the paper has improved in revision, and therefore we'll be happy in principle to publish it in *Nature Microbiology*, pending minor revisions to satisfy the referees' final requests and to comply with our editorial and formatting guidelines.

Thank you again for your interest in *Nature Microbiology*. Please do not hesitate to contact me if you have any questions.

Sincerely,

Reviewer #3 (Remarks to the Author):

I appreciate the authors substantially addressing my concerns by adding significantly to the discussion. I think the paragraph added adequately addresses my concerns. However, I have two remaining concerns:

1) I may just be misunderstanding, but I was not able to follow how the sub-analyses to assess the issue of age mismatch between cases and controls alleviate the concern. In particular:

"first, we found that the odds of a BSI patient being ≥ 60 in males compared to females were not significant for most lineage-capsule combinations, except for CC131-C2 with K5."

=> This mitigates sex as a confounder, but I don't see how this mitigates age.

"Secondly, as we could not explicitly include age as a covariate, we highlight that the elderly adult high-risk age group (≥ 60 years) is overrepresented in the data (74% of BSIs) and, therefore, that the estimates are most relevant for this important clinical population."

=> This is quite circular. The issue is that the over-representation of elderly individuals in the BSI groups makes it impossible to determine the direct attributable risk of capsule type to BSI. The over-representation doesn't address the concern, it is the concern.

"Furthermore, we ran additional iterations of the model, excluding the other age groups, to remove the assumption that the invasive potential is homogeneous across age groups and limit it to the simpler assumption for which we have some evidence: that carriage is homogeneous in the healthy population."

=> If I understand correctly, this analysis just limited the BSIs to >60 , not the controls? I don't follow the logic for how this mitigates concern regarding age as a confounder. The concern is that carriage is not homogeneous between healthy and BSI populations, potentially due to age itself (e.g. altered gut microbiota, altered immune status) or due to altered exposure patterns (e.g. frequent healthcare exposure).

To me, it would be preferable to just acknowledge the limitation, instead of trying to work/talk around structural issues in the data.

2) While the discussion softens language and acknowledges that the study design can only yield associations, the title and the abstract don't have this nuance. Given the key roles of the title and the abstract in conveying the central message to non-domain experts, especially in a high-impact journal, I think more effort should go into softening the language/implications to accurately reflect the data/analysis performed.

Version 4:

Decision Letter:

29th January 2026

Dear Dr Gladstone,

I am pleased to accept your Article "Identification of transporter-dependent capsular K-loci associated with invasive potential of *Escherichia coli*" for publication in Nature Microbiology. Thank you for having chosen to submit your work to us and many congratulations.

Authors may need to take specific actions to achieve compliance with funder and institutional open access mandates. If your research is supported by a funder that requires immediate open access (e.g. according to [Plan S principles](https://www.springernature.com/gp/open-science/plan-s-compliance) or the [NIH public access policy](https://www.springernature.com/gp/open-science/us-federal-agency-compliance)) then you should select the gold OA route, and we will direct you to the compliant route where possible. Because authors warrant under our subscription licensing terms that they haven't committed to licensing any version of their article under a licence inconsistent with the terms of our agreement – including the applicable embargo period – publication under the subscription model isn't suitable for authors whose funders require no embargo.

With kind regards,

P.S. Click on the following link if you would like to recommend Nature Microbiology to your librarian
<http://www.nature.com/subscriptions/recommend.html#forms>

** Visit the Springer Nature Editorial and Publishing website at http://editorial-jobs.springernature.com?utm_source=ejP_NMicro_email&utm_medium=ejP_NMicro_email&utm_campaign=ejp_NMicro for more information about our career opportunities. If you have any questions please click [here](mailto:editorial.publishing.jobs@springernature.com).**

Point-by-point response to reviewer comments

Please find our point-by-point response to all feedback below, we have endeavored to incorporate all suggested changes.

Editor summary

Reviewer #1 has concerns about the data being geographically limited and asks whether more global datasets can be incorporated.

We thank the editor for highlighting these important issues. We have made every effort to ensure that the K-typing database is applicable across geographical locations by screening for *kps*-positive *E. coli* strains indexed in a published database of 661,000 bacterial assemblies, which utilised all deposited bacterial short-read genome data in the ENA as of November 26, 2018 (Blackwell et al. 2021). The reviewer rightly points out that we previously missed the opportunity to include other published longitudinal datasets to inform more broadly on K-type epidemiology. We have now addressed this limitation and summarise the K-typing results in Figure 1 for a much wider collection of published genomic ExPEC collections from France (carriage, UTIs), Norway (UTIs, BSIs), the UK (carriage, BSIs), the USA (resistant UTIs), and LMICs (neonatal infections). In addition, we discuss the known geographical sequencing and publication biases, and provide suggestions on how to prioritise future genomic surveillance.

They also point out that it isn't possible to draw strong conclusions regarding the invasiveness of strains associated with given K-loci without functional evidence to support this point.

This is a key point that we now clarify in the manuscript. As we are not directly measuring the virulence of a capsule in a laboratory setting, we have changed the language used in the title and throughout the manuscript to make clear that we are presenting estimates of relative invasive potential and that our findings could reinvigorate an entire area of neglected research and strategically invest resources in laboratory experiments. We also realised that the previous version of the paper did not properly communicate the fact that the virulence of the capsule locus *per se* has already been experimentally demonstrated for the majority of common capsule types (including K1, K2, K5, K52, K92, K100) using immunological assays and murine models. We now properly cite the relevant literature to inform the reader about this. Of note, using the population-based approach, we can quantify and then crucially rank the relative invasive potential of different capsules across multiple contemporary circulating genetic backgrounds to capture the population-level link between the exposure frequencies in healthy carriage and clinical isolates from disease. While experimental studies of small numbers of laboratory and clinical strains would struggle to rank human invasiveness, we have generated clear testable hypotheses and prioritisation of certain capsular types for comprehensive study of their interaction with the immune system and role in pathogenesis. Further, by extending our analyses according to the comments from

reviewers, we were able to demonstrate that the ranking of capsules and lineages is sufficiently robust with respect to several assumptions.

This powerful epidemiological approach we use was first established in pneumococcal capsular research over 20 years ago and has also been applied to *E. coli* in recent years to quantify relative invasiveness of lineages. It has highlighted the importance of considering capsules with high relative invasive potential for pneumococcal vaccine formulations and has been used to predict the impact of future vaccine formulations on disease and AMR when only a subset of a species is targeted by vaccination against particular capsular types. (Corander et al. 2017; Colijn, Corander, and Croucher 2020; Gladstone et al. 2019; Angela B. Brueggemann et al. 2003; A. B. Brueggemann et al. 2004; Lo et al. 2019; Løchen, Truscott, and Croucher 2022; Mäklin et al. 2022; Pöntinen et al. 2024; Ojala et al. 2024)

We have expanded our introduction by describing in detail the population-based epidemiological approach. As suggested by reviewer 1, we also fully discuss the limitations and the subsequent opportunities to renew experimental investigation of K-antigens. We additionally include in the discussion our application of the model to pneumococci, which has a more established body of literature on the relative invasive potential of capsular types and show that our model reproducibly ranks the capsular types accepted to be the most virulent capsular types as those with the most invasive potential.

Reviewer #3 raises concerns about potential bias in the datasets between BSI and control sample populations and whether this precludes the drawing of conclusions regarding invasiveness from these data.

The reviewer correctly points out that it is necessary to carefully pair both the geographical area and the time period of sampling to prevent differences in strain circulation biasing the results. We also agree that our wording concerning matched carriage and disease data was misleading and have now carefully rephrased the text to avoid this. In an ideal world, the carriage and disease data would both be either local or national. Still, the rarity of *E. coli* BSI cases limits the sample size for local studies, and national carriage studies remain currently infeasible. However, even in a species with a much stronger geographical population structure than *E. coli*, local carriage and national disease for pneumococci have yielded robust and reproducible estimates (Løchen, Truscott, and Croucher 2022). To ensure sufficient numbers of disease cases, we extended the time period from 2014-2017 for carriage to 2003-2017 for disease. We have previously shown that the population structure for *E. coli* reached equilibrium quickly after the expansion of ST69 and ST131 lineages between 2001 and 2003 (we excluded data from years 2001 and 2002 for this reason). (Kallonen et al. 2017; Pöntinen et al. 2024).

However, we have now formally tested for any significant changes in proportion between the 2003-2013 and 2014-2017 periods for all lineages with more than five counts and all K-types for the BSAC BSI collection to further assess potential bias. Only one K-type differed significantly in proportion; however, it was not included in our assessment of invasive potential due to small numbers and does not affect our conclusions. Four out of the 37 lineages with >5 isolates changed proportionately between the 2003-2013 and 2014-2017 periods. The previous manuscript Figure 1 included only one of these (ST393). In the revision, we have now demonstrated using multiple additional analyses that the ranking of the estimated relative invasiveness is robust with respect to the assumptions and included a discussion of the study limitations. In particular, we identified a culture-based adult carriage survey from a similar time period (2010 vs 2014-2017) and geographical location (France vs UK). Given that these data are not an exact match regarding geographical location, year or method, it is reassuring that we found a highly similar population structure of colonising *E. coli* in French adults compared to the UK infant-mother cohort, strongly suggesting that metagenomic data from this cohort is an appropriate proxy for adult carriage exposure frequencies.

Similar to Reviewer #1, they also note that there is no experimental evidence for the functional or clinical importance of the K-loci variation.

Please see the above responses regarding experimental evidence regarding invasiveness. In addition, we have included the results of phenotypic K-typing for a selection of K-loci that represent putatively novel K-types present in Norwegian BSIs isolates. Eleven K-loci were found to have a K+ precipitate reaction with Cetavlon, but are negative for the currently known K-type antisera. This further validates the gene-based K-locus typing approach first introduced for *Klebsiella* and *Acinetobacter* (Wyres et al. 2016, 2020) in response to an initial lack of phenotypic capsular data, which subsequently stimulated considerable future study of capsules as vaccine targets for these species.

Reviewer comments

Reviewer #1

Summary

This manuscript provides a clear, robust and well-conceived exploration of *E. coli* group 2 and 3 K-loci diversity, invasive potential, and evolutionary dynamics. First, the authors mined a large genome collection to create and make publicly available a curated database for in-silico typing of g2 and g3 K-loci. Second, they explored three genome collections from longitudinal sampling of BSI (Norway and UK) or carriage *E. coli* (UK only). Most importantly, this revealed that lineages, and even subclades within global MDR lineages, with specific K-types, contribute disproportionately to invasive ExPEC disease. These findings have important translational implications (Dx, Tx and alternate countermeasures, vaccines etc). I have the following suggestions/questions for the authors

Suggestions/concerns

K-locus epidemiology. While K1, K5 and K52-like are consistently found in the top K-types between the two countries analyzed here, some varying K-types prevalence are observed between Norway and the UK. While the manuscript briefly acknowledges this geographical limitation in the method section, briefly including contemporary BSI genomic datasets from additional regions (e.g., Asia, Africa, or South America upon availability in public databases) would allow for a more comprehensive assessment of K-loci diversity and epidemiology.

We thank the reviewer for highlighting the desirability of broader geographical analysis. We endeavoured to capture as much geographical K-locus diversity for our typing database so the community could use it to type *E. coli* K-loci sampled in different global regions. We achieved this by including all *kpsF* (group 2) and *kpsM* (group 3) positive *E. coli* strains indexed in a database of 661,000 bacterial assemblies, which utilised all deposited bacterial short-read genome data in the ENA as of November 26, 2018, regardless of the sampling origin (Blackwell et al. 2021). The *kps*-positive isolates represented 69 countries with >90 isolates from all continents. This has been added to the supplementary information.

“However, by screening a published database of assemblies created from all data deposited in the sequence read archive (SRA) pre-2018(Blackwell et al. 2021), we were able to represent K-loci in the database from 69 countries, including 31 countries from Africa, Asia and Oceania. The G2-G3 K-loci repertoire observed in LMICs was well represented by the G2-G3 database even if the exact prevalence of G2-G3 K-types vary.”

We have now additionally selected published genomic studies detailing *E. coli* representative of BSIs or UTIs, primarily without selection for particular phenotypes. This enabled us to extend the geographical coverage of the analysis, including LMIC data from Africa. The amended data are now presented in the new Fig.1 to highlight both similarities and differences across regions.

“We K-locus typed assemblies from several systematic studies of infections, including: 1) large longitudinal genomic surveys of bloodstream infections (BSIs) from Norway(Gladstone et al. 2021) (2002-2017, n=3,254) and the UK(Kallonen et al. 2017; Pöntinen et al. 2024; Lipworth et al. 2024) (2001-2018, n=2,611) 2, two urinary tract infection (UTI) surveys from Norway and France(Handal et al. 2025), 3) UTIs with specific resistance profiles from the USA(Thänert et al. 2022), and 4) two collections of invasive isolates from neonates collected in low and middle-income countries (LMICs).(Sands et al. 2021; Pearse et al. 2025) In addition, K-locus typing of assemblies from infant-mother gut metagenomic surveys in the UK(Mäklin et al. 2022; Shao et al. 2024) and from traditional culture picks (France(Marin et al. 2022)) provided information on K-types in asymptomatic colonisation.”

Notably, to estimate invasive potential, comparing carriage and disease collections from the same country is necessary due to differences in strain circulation and prevalence that would affect the estimations. We are, hence, limited to data from the UK. However, it has been shown that invasive potential has temporal and geographical stability for other species, as this is considered a fixed trait of the capsule structure independent of when and where it was isolated (Angela B. Brueggemann et al. 2003) although host factors could also influence the estimates. This means that for the K-loci observed in sufficient numbers to estimate invasive potential in the UK datasets, we can expect the relative estimated invasiveness metric to be robust more widely, but have amended the discussion to acknowledge study limitations in this regard, particularly regarding age (see below). We highlight that as more countries generate both carriage and disease datasets, such data can be utilised in our model by adding country as a fixed effect, as we have performed for pneumococci to reduce uncertainty in the estimates.

“Whilst here we are currently limited to UK data for which representative snapshots of both carriage and disease have been sampled and sequenced, it has been shown that estimates of invasive potential has temporal and geographical stability for other species, as this is considered a fixed trait of the capsule structure independent of when and where matched datasets were isolated. (Angela B. Brueggemann et al. 2003)”

“As further colonisation data is generated alongside additional BSI metadata, there is potential for future meta-analyses across countries that would increase the power to estimate the relative invasive potential of K-types in different groups and reduce uncertainty around the point estimates.”

K-locus epidemiology and invasive potential. The comparison of K-loci prevalence and invasive potential between disease and carriage isolates is central to the manuscript. While age is investigated, other potential confounders (comorbidities, antibiotic exposure, local/regional outbreaks) which could influence the observed differences are not discussed. Incorporating metadata on these variables into the analysis or explicitly addressing their absence would strengthen the conclusions.

The reviewer rightly points out that host factors affect whether an individual is more susceptible to invasive disease. We have now extended the analyses to take into account both age and antibiotic treatment aspects. Previous analyses have demonstrated a lack of phylogeographical signal at the regional/national level due to rapid circulation and endemicity of *E. coli* lineages (Gladstone et al. 2021), and hence, we expect that pooling of case data on a national level will not bias the results but does significantly increase the BSI sample size. The discussion section has been further extended to allow for a more balanced treatment of these important issues.

“Given that we observed some age and sex differences in epidemiology, the invasive potential may vary in specific patient groups. The estimates reported in this manuscript

represent the average estimated relative invasive potential in BSIs across all age groups. However, 74% of BSIs are in elderly patients of greater than 59 years, and so the estimates are likely to better represent this important high-risk age group. Limiting the data to only elderly adults aged 59 years or older had a negligible effect on the rank order but, as expected, increased uncertainty in the confidence intervals of all the K-type estimates. There was insufficient data from elderly adults alone to estimate the relative invasive potential for four capsular types (KL30, K4, KL13, KL70). The only other notable difference was that K12 moved up four places in the rank order for the estimates using only elderly adults. It may thus represent a K-type that either requires a more vulnerable population or circulates more commonly in older demographics, which were not captured by the infant-mother carriage cohort. This further highlights how systematic population-based K-type screening paired with high-quality metadata can advance understanding of any potential age and sex-specific epidemiology of ExPEC infections. As further colonisation data is generated alongside additional BSI metadata, there is potential for future meta-analyses across countries that would increase the power to estimate the relative invasive potential of K-types in different groups and reduce uncertainty around the point estimates”

Hard-to-treat infections could also conceivably be overrepresented in BSI isolates due to an increased opportunity to progress to a systemic infection and influence estimates of invasive potential. To account for this, we tested the association between the lineage random effects and the proportion of UK BSI isolates per lineage that were either phenotypically MDR (EUCAST v15) or *bla*_{CTX-M} positive. There was no correlation with the MDR proportion ($R^2 = 0.029$, $p = 0.79$), and when the CC131 C2 outlier was excluded, there was also no correlation with the *bla*_{CTX-M} proportion ($R^2 = 0.036$, $p = 0.74$). Furthermore, when including the *bla*_{CTX-M} proportion per lineage in an alternative mixed model, this covariate was not significant, and the only resulting difference was that the CC131 C2 lineage ranking fell one position (behind CC12). Finally, to further assess its robustness, we applied our model to a pneumococcal collection that had previously been used to assess invasive potential (Gladstone et al. 2019) and reproduced the general ranking order of invasiveness by others, which includes capsules since long established to be the most virulent for this bacterium (Gladstone et al. 2019; Løchen, Truscott, and Croucher 2022; Angela B. Brueggemann et al. 2003).”

Invasive potential of K-types. The authors state that their findings “... highlight an extensive variability in the propensity of K-type and lineage to cause infection.” Without experimental validation, the roles of specific K-loci in invasiveness remain an inference from a statistical model. While such analyses would significantly enhance the conclusions of this paper, simply acknowledging the need for follow-up functional genomic studies would contribute to a more balanced discussion.

Indeed, the experimental assessment of invasiveness would naturally complement our epidemiological population-based approach. Whilst the complexities and impact of genetic background are infeasible to be fully accounted for in experimental studies, our

results offer the opportunity for targeted experimental work in the future to understand the relative invasive potential further. We argue that using natural colonisation and clinical samples to capture the link between exposure frequencies and disease overcomes the limitations of experimental models. This approach is epidemiologically sound and, as discussed above, has provided the basis for establishing capsule virulence and vaccine composition in another major bacterial pathogen (the pneumococcus), which is now more fully introduced and discussed, in addition to a rewording of the title and references to relative invasive potential. However, as also noted in our above responses to the editor, there is already an extensive literature on experimental verification of the virulence of many *E. coli* capsule types, which was missing from the previous version and is now appropriately cited in the introduction and discussion.

“The role of capsules in pathogenesis has long been appreciated across bacterial species (Mba et al. 2023; Moxon and Kroll 1990). Whilst the study of *E. coli* capsules has stalled more recently, previous research efforts have already experimentally demonstrated the virulence associated with most of the common *E. coli* capsular types (e.g. K1, K2, K5, K52, K92 and K100) using either human immunological assays or murine models (Buckles et al. 2009; Cross et al. 1986; Arredondo-Alonso et al. 2023; Merino et al. 2020; Mostafavi et al. 2019; Suerbaum et al. 1994). Estimating the relative invasive potential allowed us to put these experimental findings into a wider context by ranking the capsular types in terms of their propensity to be found in BSIs, given the exposure frequency in asymptomatic colonisation.”

“In summary, our results provide in this regard impetus for reinstating phenotypic capsular typing for this species and renewed interest for experimental studies of virulence that could further disentangle the contributions of capsules versus lineages towards invasiveness.”

Role of Insertion Sequences. The findings reveal that capsule diversification is largely due to IS elements. It is also discussed that G3-B K-antigens have been observed in plasmids, including isolates of G3-B K11 from Norway (a genome collection for which hybrid assemblies are available). This search was not done for the UK genome collection with short read data available. As a possible way to expand this analysis, existing BSI plasmidome dataset from the UK (PMID: 38383544) may be used to detect K-loci plasmid carriage in clonal isolates.

In this manuscript, we had the opportunity to assess the disease prevalence of plasmid-borne K-loci that had been reported in the recent literature. It is an excellent suggestion that typing additional datasets could enable further investigation of this phenomenon. We found only one additional isolate in UK hybrid assemblies with a K-locus in a plasmid. It may be more important in other disease types or populations. Our typing database enables others to more easily determine this in additional datasets in the future.

“Additionally, a single *E. coli* isolate from Oxfordshire (n=1/549) was typed as G3-B KL139 on the plasmid and not in the chromosome.”

Lines 129-131, 162-172 and 181-183. Should Figure 2 be cited in the text for reference?

Figure 3A and 3B (previously figure 2) are now mentioned in the respective paragraphs.

Figure 2. It is unclear why no UK isolates were included in the phylogenies. Would including those change the inference for capsule switch events and the dating of the MRCA? Also please consider adding O and H types for both panels.

This is an excellent comment. We have now included the UK isolates in this analysis and show that the conclusions remain unchanged. O and H types are now included for both lineages. To avoid overloading the phylogenetic analyses with detail, we have now omitted the CaveDive expansion analyses as these had only limited biological and epidemiological relevance beyond our demonstration of the capsule diversity and acquisition timelines. We summarise the main metadata in the figure and provide a link to an interactive version with complete metadata.

“Figure 3. Diversity of K-loci across the clades of CC131 and CC69 from Norwegian and UK BSIs. Major capsule switch events are indicated with a light grey circle and dashed line from the relevant nodes with the time to the most recent common ancestor and confidence intervals. A) The four major CC131 clades are labelled in column one. The major O-H types are displayed in column 2. The K-loci are colour-coded in column 3, with the adjacent key displaying the known K-type if phenotypic data were available for the corresponding locus. B). The major CC69 K-loci are colour-coded in column 1, and the K-group in column 2. Full annotated phylogenies available at <https://tinyurl.com/CC131-NORM-BSAC> <https://tinyurl.com/CC69-NORM-BSAC>”

Figure 4. It is difficult to connect the text and the illustration. Please consider labelling the *kps* genes, providing genomic coordinates, labelling the k-loci (and not only the K-phenotypes) etc. For the region 2 gene clusters, how are genes ordered to make the presence/absence matrix?

We thank the reviewer for their suggestions for improving this figure. The dominant *kps* gene clusters are ordered according to the K-locus, and the remaining blocks representing region 2 accessory genes were ordered by the number of KL they were found in, from largest to smallest. These details are now included in the figure legend. We felt that including all tip labels and gene annotations would be difficult to read, but we have included the major *kps* labels and K-phenotypes to show how little diversity the known phenotypes represent. The underlying data input files are now available on GitHub to allow them to be viewed in an interactive figure format on Phandango with all KL and all annotated genes for Fig. 4A and 4B. We have added an additional description of the figure and pointed readers to the interactive version for finer scale interrogation.

Figure 5. Neighbour-joining trees of G2 (5A) and G3 (5B) K-loci based on core and accessory sequence presence/absence, annotated with known K-phenotypes at the tips. The first metadata column denotes the K-loci with atypical gene organisation (5A) and subtypes of G3 (5B). The gene cluster presence (blue) and absence (white) used a 70% AA identity threshold. Three highlighted blocks show the dominant *kps* gene clusters in the order they are most often observed in the K-locus, divergent *kps* gene clusters, and the most common region 2 gene clusters ordered by the number of KL they were observed. *only gene-clusters present in >2 K-loci are presented. These figures can be viewed interactively with full taxa and gene sets and annotation on phandango.net using the input files provided on <https://github.com/rgladstone/EC-K-typing/tree/main/phandango>

A few typographical errors were also spotted:

Line 114 "in phylogroup in D, B2 and A."

Line 178 "homologues recombination"

Line 427 "analysis of ~75,00 high-quality E. coli genomes"

These are corrected.

Reviewer #2

In this manuscript, Gladstone and colleagues leverage very large Escherichia coli genomic datasets from several cohorts to provide a detailed catalogue of capsules belonging to group G2 and G3. More particularly, they focus on strains of Escherichia coli isolated from bloodstream infections. They built a curated capsule type database, with annotations and this has been formatted to be compatible with Kaptive, and easy-to-use and well-established pipeline to identify and type K-loci in other prominent ESKAPE pathogens. This is an added value of the manuscript, and will be of use for other researchers. Given the importance of ExPEC infections, the rapid expansion of several of these lineages, acquisition of antibiotic resistance, this research is both interesting and timely.

The authors then use this 'database' to correlate the sequences with their potential for invasive disease, and provide insights into the evolution of G2 and G3 capsules. Specifically, the results suggest that capsules evolve by recombination, driven by IS.

This manuscript provides different complex analyses and, despite the large amount of work put into it, or precisely because of it, unfortunately, it is not clear what is the major question being answered or the biological advance. At times, I found the manuscript very difficult to read and follow. I hope my comments will improve the manuscript and its readability in order to render it more accessible and comprehensible. I believe that:

1. Mentioning some basic methodological procedures in the text would help with the understanding. Also, some of the methods are incomplete.
2. Another improvement would be to introduce clearly the typing scheme and to reduce the excessive capsule jargon.
3. The inclusion of numerous summary statistics in the text limits readability (maybe some can be transferred to tables?).
4. The manuscript is fairly descriptive, with no clear hypothesis from the start, which makes it difficult to follow as the reader does not know where they are being lead to. Thus, for the most of it, it reads like an exhaustive epidemiological report. It is not very clear what the ("biological") aim of this research is, nor the main message.

Major comments

1. My major concern relies on the naming scheme which makes this manuscript very complicated to follow. The authors should state clearly from the beginning what is a K type, the K-loci, and KL. Also K, KL, K-like notations are very difficult to follow. It would be appropriate to elaborate/mention/ describe the K-naming scheme from the beginning. It is very confusing at points. Most strains are identified with a K and KL numbers, which is different?. Also, the logic of naming is elusive, sometimes K is mentioned first (example L116), then KL in parenthesis. Sometimes, the opposite is true (line 175).

To complicate things further one sentence deals with Ks and the next with KL.

Are KL notations those identified by Kaptive? It is my understanding that Kaptive is a database based on *Klebsiella pneumoniae* sequences. Although it has been shown that capsules may be transferred across species, this should be introduced and commented for a broader readership to understand. That being said, is it necessary to include each KL name? maybe a supplementary table with the equivalency would be better and could alleviate the manuscript?

Kaptive is a species-agnostic system for surface polysaccharide typing from bacterial genome sequences. This was first used for *Klebsiella*, followed by *Acinetobacter baumannii* (Wyres et al. 2016, 2020). It has two main components: A curated reference database of capsular polysaccharide gene clusters (loci) for a species, which we have created for *E. coli*, and a command-line interface that searches user-provided assemblies for matches to the curated species-specific reference database. We followed the suggested Kaptive nomenclature, which allows for the differentiation between the known capsular phenotype, e.g. K1 and the capsular K-locus (KL) that encodes it, here denoted as KL8. The phenotype is not yet known for multiple K-loci with putative novel K-types reported here, and so we refer to the K-locus (KL). We thank the reviewer for pointing out that we were not consistent in the way we used the nomenclature and where we introduced unnecessary complexity.

We have now standardised the references to K-types and their corresponding loci and added a clarification to the reader early in the Results section to avoid confusion. We will refer to the K phenotype directly when it is known for a K-locus, and when it is not, instead refer to the KL-type. Where both are required, it will be presented as K (KL), e.g., K1 (KL8). Given the feedback, we have also decided to remove the “like” designation used in the literature to reduce the complexity and instead direct readers to the representation of their phylogenetic relationships between K-loci and in the supplementary summary of the database. We have also expanded the methods so that it is clear that the K-loci number is an arbitrary designation resulting from the clustered gene presence-absence profile of each locus.

“Extracted K-loci with unknown bases were excluded, and the K-loci were filtered down to 4,996 unique sequences with at least 1bp difference. These unique K-loci were annotated with Prokka(Seemann 2014) and analysed with Panaroo(Tonkin-Hill et al. 2020) to consistently annotate K-loci gene clusters with a conservative 70% family identity threshold. The gene cluster names were updated using the capsular-specific reference annotations to reflect the literature where possible. This resulted in an initial

225 gene absence patterns, excluding insertion elements (IS), which are arbitrarily numbered KL1-225. IS elements were identified from the annotations using ISEscan(Xie and Tang 2017). These 225 sequences were pruned to remove K-loci with redundant patterns due to misannotation, paralogs, non-capsular genes, and incomplete K-locus remnants, leaving 90 curated K-loci in the final K-loci database”

“As numerous K-loci were not observed among the limited reference genomes with a known phenotypic K-type, the phenotypes can be added retrospectively in the future when more data become available. The KL designations reported here are determined algorithmically and, consequently, do not mirror a K-type. Throughout this manuscript, we will report the inferred phenotypic K-type (e.g, K1) for a given K-locus (KL) where possible. Where no phenotypic K-type currently exists for a K-locus, an ordering-based KL designation (e.g. KL5) will be used instead.”

Also, how was the typing scheme decided/annotated? is it based on known capsules and the literature? Based on phenotypic serotype testing?

Kaptive requires a reference database containing complete K-loci, each representing a unique gene presence and absence pattern observed in the population for the capsule locus. The K-loci were first extracted from assemblies where at least one *kps* gene was detected (annotation or kmer screen) using an *in-silico* PCR for known K-loci terminal genes, and PCR-negative K-loci were then manually extracted. The extracted loci were annotated with Prokka. Then we used Panaroo, which clusters genes into families, allowing us to update the initial annotations so that they were consistent across the database. When we had a genome for which the K phenotype was known, we could additionally assign the K-phenotype to a K-locus. All unique K-loci were included in the database regardless of whether the phenotype was available. We now include additional phenotypic data provided by the Statens Serum Institute in Copenhagen, showing that isolates with K-loci where the phenotype is not known do produce capsular polysaccharide and represent multiple putatively novel K-types.

We have added the necessary details to the methods and the phenotype-genotype evidence to the supplementary.

“Genomes with a known K phenotype were sourced for 24 different K-types from GenBank, the NCTC project(Dicks et al. 2023) and Enterobase(Zhou, Charlesworth, and Achtman 2021). The NCTC reference strains (accession PRJEB6403) are maintained as preserved strains that can be obtained from the National Collection of Type Cultures, UK. K-loci were initially extracted from 21,802 assemblies using an *in-silico* PCR (https://github.com/simonrharris/in_silico_pcr) for known K-loci boundaries i.e. *kpsF-kpsM* or *kpsM/D-kpsS*. PCR-negative K-loci were then manually extracted from assemblies that had been annotated with one or more *kps* genes.”

“Extracted K-loci with unknown bases were excluded, and the K-loci filtered down to 4,996 unique sequences with at least 1bp difference. These unique K-loci were annotated with Prokka(Seemann 2014) and analysed with Panaroo(Tonkin-Hill et al. 2020) to consistently annotate K-loci gene clusters with a conservative 70% family identity threshold. The gene cluster names were updated using the capsular-specific reference annotations to reflect the literature where possible. This resulted in an initial 225 gene absence patterns, excluding insertion elements (IS), which are arbitrarily numbered KL1-225. IS elements were identified from the annotations and using ISEscan(Xie and Tang 2017). These 225 sequences were pruned to remove K-loci with redundant patterns due to misannotation, paralogs, non-capsular genes, and incomplete K-locus remnants, leaving 90 curated K-loci in the final K-loci database”

Below are some examples of some particularly confusing bits:

*Line 116, “The top five K-types causing BSIs were K1, K5, the K52-like K116 locus (KL)189, K2 and the K14-like KL137”. This is very confusing, if there are 101 unique K loci (line 99), how is it that there is a KL137and KL189?

The K-loci (KL) are arbitrary numerical assignments to differentiate them from each other. Initially, there were over 200 unique gene presence-absence patterns identified. Over 100 K-loci have been pruned to remove those with redundant patterns due to misannotation, paralogs, non-capsular genes, and incomplete K-locus remnants, resulting in 90 curated K-loci. This is also true for K-phenotypes that are numbered K1-K103 in the existing literature, but over 30 of which were later removed as they turned out to be O-antigens. The naming is complex and will evolve as more phenotypes and K-loci are added or removed as new knowledge comes to light. Critically, the assignment of KL to new data is robust and reproducible, and the existing K-locus names will be maintained for continuity.

* Also, as I understand, K1 is a G1 capsule, but is also mentioned here (line 116). This is not really clear in the text.

Although both species use the same K nomenclature, here we are talking about the *E. coli* K1 capsule, which belongs to group 2.(Arredondo-Alonso et al. 2023)

*it is confusing to see this K52-like KL189 (line 143). What does this mean exactly?

This has now been phenotypically confirmed as K52 and will be referred to as such. All “-like” designations have been removed to improve clarity.

*Line 175: KL53 (K2-like, n=41/631), KL3 (K2, 34/631), KL137 (K14-like, n=29/631)

From the phylogeny, we can deduce that KL53 is related to another K-locus with the known phenotype K2 (KL3). KL137 is related to the K-locus with a known K14

phenotype, but as this introduces unnecessary complexity, we now exclude these details from the manuscript and describe them in the supplementary.

* Line 268: The K5 reference (KL6)

The reference genome known to have the K5 phenotype has the K-locus KL6. This has now been better introduced at the beginning of the results.

* Line 289: K4-KL29 and the K4-like KL107

The K-like designation has been removed from the manuscript.

*L338- “Indeed, we observe serotypes with the same capsular genes with high-sequence identities. These include K13 and K23 K-loci that are typed as KL21”.

K13 and K23 are phenotypes; all isolates that phenotype as K13 and K23 have a K-locus denoted KL21. These capsular types are in a serogroup and share the same genes with sequence differences that determine the exact K-antigen.

We have rephrased to make this clearer.

“These include the K13 and K23 reference phenotypes that are both typed as KL21. They have previously been reported to belong to a serogroup along with K20, which has replaced one gene relative to K13 and K23 (Vann et al. 1983). When comparing K13 (NCTC9022, ERR999921) and K23 (NCTC10430, ERR968281), there were only two non-synonymous changes in a region 2 gene, *vatD*, that encodes an acetyltransferase. This putative determinant needs an in-depth phenotypic validation to allow full genomic discrimination between them.”

2. There are some important details in the methodology that are not sufficiently described and could bias the capsules identified.

First, how was *kpsF*/*kpsM* identification performed? Was it done with nucleotide or protein sequences? Also, it is not clear how the *kpsF* allele to search for G3 capsules was chosen. Why particularly choose that of K54? Does changing the initial allele alter the number of G3 capsules recovered?

All G2 *kpsF* sequences cluster together and are highly conserved, whilst G3 *kpsM* sequences cluster in G3-A and G3-B, so we searched for representative sequences using a 0.90 threshold for kmer identity in a searchable indexed database of assemblies. We do not expect that reducing the kmer threshold will identify the gene in significantly more assemblies without losing specificity. In our typed collections, the number of *kps*-positive isolates that are untypeable is very low, <0.3%. However, as novel K-loci are identified in further collections, they can be added to the database. We are currently

working with the Kaptive researchers as they are interested in finding a solution to co-host the database, which is publicly available and will be maintained and updated in the future.

“We used a published pangenome analysis of ~7,500 high-quality *E. coli* genomes to determine that the capsular gene *kpsF* was consistently annotated and predictive of G2 capsule presence (Horesh et al. 2021). We subsequently downloaded all *kpsF*-positive (0.9 kmer-ID threshold) *E. coli* in a published searchable collection of 661-thousand bacterial assemblies (n=11,623)(Blackwell et al. 2021). For G3, we additionally screened the 661K database for all *kpsM*-positive assemblies (0.9 kmer-ID threshold, n=853) using the G3-A and G3-B *kpsM* alleles from K96 and K11, respectively, as *kpsM* is more divergent in G3”

L442 “After extracting the *kps* K-loci from 21,802 assembly”, how was this performed, how was the beginning and end of operon delimited?

Apologies for this oversight; this detail has been added to the methods.

“K-loci were initially extracted from 21,802 assemblies using an *in-silico* PCR (https://github.com/simonrharris/in_silico_pcr) for known K-loci boundaries i.e. *kpsF-kpsM* or *kpsM/D-kpsS*. PCR-negative K-loci were then manually extracted from assemblies that had been annotated with one or more *kps* genes.”

3.Data presentation. This manuscript presents a lot of data (ratios, percentages) on the prevalence of capsules, genes, IS, etc.. These are detailed in the text which is difficult to digest. Some extra panels in figures with summary statistics and percentages would be appreciated. Also, some graphs on distributions (# of genes in different clusters, # of IS per cluster, etc..)

We have simplified this section for clarity and moved the figures detailing IS variants to the supplementary as it distracts from the main story of the paper.

“The genomic survey of G2 and G3 K-loci allowed us to interrogate the mechanisms that may have generated this considerable diversity. The proportion of unique K-loci sequences (at least 1bp difference) with one or more insertion sequences (IS) was 28.2% (1411/4996). While K1 was only observed with an IS once (n=1/692), K5 always had at least one IS (n=706). IS1 and IS3 were the most common of the ten IS families observed in the K-loci collection. Importantly, IS were observed overlapping with capsular coding sequences (CDS) in 17 K-loci and in nine of these, the IS carried a putative capsule gene in region 2 as cargo.”

4.The section on K loci diversification maybe interesting for capsule and *E. coli* aficionados, but not necessarily for a broader readership as it is a catalogue of different atypical K organizations. What would be the main message here? Conceptually, capsule diversity is not exceedingly novel.

We now draw a parallel between O and K-types capsular diversity, which makes it unlikely that any vaccine could target all invasive K-types, and there is then potential for disease replacement as seen in pneumococci. This section is intended to provide understanding about the evolutionary mechanisms underlying capsular diversification, which has implications for the development of diagnostic targets, phage therapies and vaccine design.

5. The study of capsule evolution is interesting, but raises some questions. Is recombination rate different across the different regions of the operon? Said otherwise, are there hotspots for recombination?

Using the multi-capsular and recombinogenic lineage ST131 as an example, we can determine that the *kps* locus is found at the peak of the fourth highest recombination hotspot across the genome, recombinations at this peak span a 300 kb region, with the longest detected recombination event being 100kb encompassing the 15 kb *kps* locus, the number of recombinations detected across the *kps* locus was stable at around 40 but variation over this region is likely hard to quantify exactly due to the large number of recombination events happening here. We have added that the *kps*-locus is a recombination hotspot to the results.

“In the top four most common lineages in BSIs (CC73, CC95, CC69 and CC131) the *kps*-locus is a clear recombination hotspot with the exception of CC95 which only expresses K1.”

Does phylogenetic distance between variable regions correlate with distance between *kps* genes? Do the different *kps* genes share a common phylogenetic history?

The conserved organisation of the core *kps* genes in the *kps* locus, even between G2 and G3, suggests that this type of capsule locus shares an ancient origin. The different clusters of *kpsM* (<70 AA %ID) within G3 and between G2 and G3 suggest multiple historical importation events of the *kps* locus from more diverse strains or species, though attempting to assess the origins is out of the scope of this manuscript and an interesting area for future research. The gene presence-absence patterns in Fig. 5A and 5B for the region 2 genes show that there is much more variation in this accessory region, with few gene clusters shared between K-loci.

6.K-loci clusters. No results are presented, yet there is a large potential in these data which goes unexplored or the text remains vague. For those clusters that have an annotated gene, it would be useful to include it in figure 4. How many genes are there in each cluster? Are most gene clusters, singletons? What is the structure of these clusters? Maybe provide a gene network would be useful. Are the different clusters linked among them, either spatially (often colocalised) in the genome or in terms of identity (ie some clusters may have members with 65% identity to others which are in another cluster)?

We thank the reviewer for this suggestion. We have added to the figure legend that the cluster threshold is 70% AA ID. We have separated the core *kps* clusters and the divergent *kps* clusters in the figure and added the major *kps* gene names in the order they are found in the locus. We agree that these details are interesting to explore, but we felt that they would overwhelm the figure or be too small to read. We now point readers to the interactive figure viewer Phandango.net and the required input files on <https://github.com/rqladstone/EC-K-typing/tree/main/phandango>, where all the gene annotations are included for all gene clusters.

7. The discussion is nicely written but some parts are speculative with vague future potential implications. Indeed, sentences like “cataloguing capsular genetic diversity in contemporary disease is essential to further our understanding of K-antigens in the fight to control the increasing burden of extraintestinal pathogenic *E. coli*” or “diversity has major clinical implications, with large variations in the invasive potential of K-types in different genetic backgrounds”, may be true but I have difficulty in linking directly the results presented in the manuscript here these grand objectives. For example, the end of the discussion L393-415 provides specifics directly concerning the work presented. This is appreciated.

We thank the reviewer for constructive remarks and have improved the Discussion to remove elements of ambiguity. The Discussion section has also been amended based on comments from the other reviewers.

Minor comments

L92-95 “we catalogue the incredible diversity of *E. coli* G2 and G3 capsular loci, [...] and associations with [...] mobile genetic elements, and antimicrobial resistance (AMR)” I am not sure this has been done as stated. Were analyses of capsule types vs MGEE and AMR done for all genomes? Was this done systematically? What about phages, ICEs, etc..? No specific methods are included

We have updated this sentence to more specifically represent the findings “and quantify their relative invasiveness and associations with DNA exchange in homologous recombination, insertion sequence elements and plasmids”

Figures should be revisited as they are not color-blind friendly (specifically figure 1)

We have changed the colours to improve them in this regard.

L174 is known

Corrected.

Figure 2 legend- Notation for K types should be explained. What would KL36_K54-K96 mean?

A more homogenous representation would help the reader (for instance same number of columns, same order, same information), for example; group membership for panel A? Keep the same K locus color in both panels for K6?

Also, would a circular representation be better? Just wondering.

Thank you for pointing out these shortcomings. The figure has been revised for clarity accordingly, and an interactive version has been provided that can be viewed and searched in different ways.

“Full annotated phylogenies available at <https://tinyurl.com/CC131-NORM-BSAC>
<https://tinyurl.com/CC69-NORM-BSAC>”

Figure 3B, is the correlation skewed by the two extremes (ST131, and ST69). Is the correlation still valid if these two lineages are excluded?

The positive trend still exists with the exclusion of these two outliers, although no longer significant. We have added this to the manuscript.

“Indeed, the number of K-loci in a lineage was correlated with the r/m (recombination to mutation ratio) of that lineage, even when controlling for the lineage diversity by using the recombination-free median mutational pairwise distance (MMD, $R^2=0.95$, $p<0.0001$, Fig. 4B). There was no correlation between r/m and MMD ($R^2=-0.35$, $p=0.3$). The trend remained significant with the exclusion of CC69 ($R^2=0.75$, $p=0.02$), but not after excluding the two most recombinogenic lineages CC69 and CC131, though the association remained positive ($R^2=0.47$, $p=0.2$). In the top four most common lineages in BSIs (CC73, CC95, CC69 and CC131) the *kps*-locus is a clear recombination hotspot with the exception of CC95 which only expresses K1. Highly recombinogenic lineages could contribute to the discovered changes in the K-locus genetic architecture, including the observed atypical K-loci.”

L332- “we observed a greater diversity of K-loci from G2 and G3 than O-loci in most dominant ExPEC”. the manuscript does not mention the number of O-loci identified in this data set, or methods to identify them.

O: H-typing was performed with SRST2 (Ingle et al. 2016). This reference has been added to the methods. The diversity of O, H and K types was measured using the Simpson's diversity index 1-D. The OHK call and Simpsons diversity indexes per lineage have been added to the supplementary metadata.

L403 “Nevertheless, Increased”

Corrected.

L 457 No information is provided as to how the phylogenies were performed? On protein sequences? were these rooted? Are these trees bootstrapped? How many informative sites were taken into account? also, why neighbour-joining trees were computed as opposed to maximum likelihood?

As the capsule locus is made of both core (regions 1 and 3) and accessory components (region 2 that determines the capsular sugars), traditional sequence alignment-based phylogenies are inappropriate. Instead, the relatedness of the K-loci was assessed using their pairwise mash distances. We chose to use a matrix of mash hash distance, which is the number of shared kmers out of 1000. We converted this to a proportion $1 - (n/1000)$ to represent a distance of 0 (1000/1000 shared kmers) to 1 (0/1000 shared kmers). This gives greater resolution than a gene presence-absence distance matrix as each K-locus is around 15-20 genes, but still captures both core and accessory variation (Ondov et al. 2016). The distance matrix was then used to create an unrooted Neighbour-joining phylogeny in the R v4.4.1 package ape 5.8, which is a standard approach for distance matrix data. The figure legend and methods have been expanded with these details.

“The relatedness of K-loci was assessed for G2 and G3 separately using the pairwise mash hash distance converted to a proportion $1 - (n/1000)$, which gives greater resolution than gene presence-absence but still captures core and accessory variation (Ondov et al. 2016). These distances were used to create Neighbour-joining phylogenies in the R v4.4.1 package ape 5.8 and visualised in [Phandango.net](https://github.com/rghadfield/Phandango.net) (Hadfield et al. 2018) using the following input files <https://github.com/rghadfield/Phandango.net>.”

L459-461 I don't understand

This line has been removed from the discussion, and we have replaced it with a sentence at the start of the results that clarifies the difference between the phenotype designation (K) and the genotype designation (K-locus KL).

“Throughout this manuscript, we will report the inferred phenotypic K-type (e.g, K1) for a given K-locus (KL) genotype when available. Where no phenotypic K-type currently exists for a K-locus, an ordering-based KL designation (e.g. KL5) will be used instead.”

Reviewer #3

Gladstone and colleagues present an analysis of previously published E. coli genomic data sets to provide insight into variation in capsule biosynthetic loci and its the associated impact on clinical infection risk. Among the major findings are: i) evidence for underappreciated diversity in capsule locus, as compared to prior serological classification, ii) evidence for capsule locus diversification in epidemic lineages and association with population expansion events and iii) evidence for the association of capsule type with risk for blood stream infection.

A major strength of this submission is the cataloging of capsule types, which has proven extremely valuable in enabling comparative genomic studies for other organisms, and a description of capsule diversity across important epidemic lineages. There are also several weaknesses. First, while intriguing, the analyses of risk associated with capsule types is severely flawed, to the point of being potentially misleading. Second, while the increased genetic diversity compared to phenotypic diversity is impressive, there is no evidence provided for functional impact associated with genetic variation. Lastly, no experimental support is provided for any claims regarding functional or clinical importance of observed variation.

Many of the reported K phenotypes for the 28 reference genomes included in the database have established clinical relevance due to their differing associations with particular disease types, for example, K1, K2 and K5 (Arredondo-Alonso et al. 2023; Buckles et al. 2009; Cross et al. 1986). Experimental verification of the virulence of the capsule type using immunological assays and murine models has also been provided for many other types. We apologise that the previous version omitted the existence of such data and have now cited the relevant literature in our revision. In addition to the paired genotype-phenotypes for 21/31 known G2-G3 K-types from reference genomes, we now include K phenotype data generated by the Statens Serum Institute in Copenhagen. These data provided known phenotypes for four additional K-loci and further demonstrated that 11 G2 K-loci with unknown phenotypes are K-positive, as they have a positive precipitate reaction with Cetavlon, but are negative for the currently known K-type antisera. These represent putative novel K-antigens. However, we acknowledge the need for future comparative functional studies to further examine the mechanistic contributions of the K types in the invasiveness, as has been done for the K1 capsule, for instance (Arredondo-Alonso et al. 2023).

The absence of phenotypic data was one rationale for performing in-silico K-typing, where it is biologically plausible that a K-locus that has a different set of region 2 capsular-determining genes could result in a novel phenotype. This is the approach previously taken by others for Klebsiella K-loci, and we are now actively working with the Kaptive team to ensure the longevity of the *E. coli* database, which will be regularly updated when new phenotypes are determined, including for any K-loci that turn out to be acapsular or degenerate capsules. As we have observed several K-loci for which the phenotype is not yet known in major BSIs lineages and over-represented in disease, this again suggests that these likely express a capsule to survive in the bloodstream and warrant further investigation.

The Discussion has been amended to address this aspect, as suggested by the reviewers, and the phenotypic evidence for each K-type is provided in the supplementary. We also now highlight in the title that the study focuses on variation in the *estimated invasive potential*, to avoid a misleading interpretation.

Major critiques

1. A central result highlighted in the title, abstract, results and discussion is the finding that specific capsule types are associated with invasive infection. However, to say that the performed analyses are not setup to make these inferences is an understatement. Among the major issues is that the case population (i.e. bloodstream infections) is distinct from the control population (i.e. mother-infant pairs) with respect to time (cases 2003-2017, controls 2014-2017), population (i.e. healthy mothers/infants versus hospitalized patients) and clinical characteristics (i.e. likely no co-morbid conditions versus many). Note that I do not believe that these limitations can be overcome with this data, but among things that could have been done to try to allay concerns are: (i) focusing on case populations from the same time period as controls to reduce potential impact of population dynamics, (ii) focusing exclusively on mothers for controls, as host-microbe interactions and risk profiles are distinct in neonates, (iii) select cases to match the age distribution of control mothers, especially given age association with capsule/risk, (iv) limit cases to females, given the controls are all female and sex variation is reported, and (v) perform some experimental validation of key results (e.g. impact of capsule switching in lineages of interest in relevant disease models).

It has been established in the microbiome research that *E. coli* (ExPEC in particular) are among the first bacteria to colonise newborns and remain typically in high abundance during the early life after birth, which further facilitates detection assembly and subsequent high-resolution genomic typing of the colonising bacterial strains, as shown by the data from (Mäklin et al. 2022; Shao et al. 2024, 2019). Despite *E. coli* being a normal constituent of the adult gut, its low abundance in adults meant that only 67/997 assembled strains were from the mothers. Infants, however, are colonised by bacteria transmitted from the mothers and other family members, as shown by previous research. Thus, this provides an excellent opportunity to establish the distribution of lineages and genetic elements, such as the capsule locus, in a baseline healthy colonisation population. To further support this, we found a published healthy adult colonisation dataset from France from 2010 based on whole-genome sequencing of colony picks without phenotypic bias (such as particular antibiotic resistance) and used this to show that the distribution of top capsule and sequence types is highly similar to that found in the UK infant-mothers, supporting that the latter are a suitable proxy for adult colonisation frequencies. We have also amended the invasive potential analysis with respect to the effect of BSI age groups and antibiotic resistance, which was also suggested by another reviewer, showing that the conclusions remain sufficiently robust. We have now expanded the Discussion to better acknowledge study limitations, which were not properly addressed previously.

“Furthermore, we found no significant differences in the overall K-type composition between the UK infant-mother collection and French adults. K1 and CC10 were the only K-type and lineage with a significantly higher prevalence in the French collection. If these two differences in carriage were due to the different age-groups sampled, the relative invasive potential could be over-estimated for K1 and CC10 in our analysis, this in fact would further support our conclusion that high K1 prevalence in carriage influences the high prevalence in disease and that other lesser studied capsular types

top the invasiveness rankings. Therefore, despite differences in the sampling time period (2014-2017 vs. 2010), sequencing approach (metagenomic vs. single colony sequencing) and geographical location, the two carriage populations are similar suggesting that there is limited age structure to human *E. coli* colonisation in this geographical region. Whilst here we are currently limited to UK data for which representative snapshots of both carriage and disease have been sampled and sequenced, it has been shown that estimates of invasive potential has temporal and geographical stability for other species, as this is considered a fixed trait of the capsule structure independent of when and where matched datasets were isolated.(Angela B. Brueggemann et al. 2003)”

2. Providing the raw data underlying case-control analysis would be helpful for the reader in terms of understanding numbers of observations underlying inferences. Showing raw data in supplement (i.e. colonization versus case counts for each ST/capsule pair) would be valuable.

We have clarified which isolates are included in the analysis in the supplementary information and provided all the model input and output on GitHub <https://github.com/rqladstone/EC-K-typing/tree/main/invasiveness>

3. In the abstract it is stated that “paired carriage and disease cohorts” were investigated. It is misleading to say these are “paired” colonization and infection cohorts. They are in fact unrelated in time, space and population.

We agree that the wording “paired” was highly misleading and have edited the text accordingly. The reviewer rightly points out that it is necessary to carefully match both the geographical area and the time period of sampling to prevent differences in strain circulation biasing the results. In an ideal world, the carriage and disease data would both be either local or national. However, the rarity of *E. coli* BSI disease cases limits the sample size for a local analysis, and national carriage studies are infeasible. However, even in a species with a much stronger geographical population structure than *E. coli*, local carriage and national disease for pneumococci have yielded robust and reproducible estimates (Løchen, Truscott, and Croucher 2022). Again, to ensure sufficient numbers of disease cases, we extended the time period that the sampling was performed from 2014-2017 for carriage to 2003-2017 for disease. We have previously shown that the population structure for *E. coli* reached equilibrium quickly after the expansion of ST69 and ST131 lineages between 2001 and 2003 (years 2001 and 2002 were excluded for this reason). (Kallonen et al. 2017; Pöntinen et al. 2024).

We have now additionally tested for significant changes in proportion between the 2003-2013 and 2014-2017 periods for all lineages with more than five counts and all K-types in the BSAC BSI collection to further assess potential bias. Only one K-type differed significantly in proportion; however, it was not included in our assessment of invasive potential due to small numbers and does not affect our conclusions. Four out of the 37

lineages with >5 isolates changed proportionately between the 2003-2013 and 2014-2017 periods. The previous manuscript Figure 1 (now Figure 2) included only one of these (ST393), for which our conclusions could potentially have been affected. The additional analyses subsampling ST393 and the amended discussion about study limitations hopefully address the concerns raised in this comment.

“The age of BSI patients, the likelihood that a strain was CTX-M positive, and an increase in BSI prevalence of CC393 between 2002-2013 and 2014-2017 were possible confounders, the model was run four additional times: 1) BSAC restricted to adults only n=1550/2036, 2) BSAC elderly adults only n=1202/2036, 3) BSAC with CC393 subsampled in 2003-2013 down to the same proportion as 2014-2017 of 0.02 n=2008/2036, 4) adjusting for the CTX-M prevalence of each lineage in BSIs. We further assessed whether the neonatal metagenomic carriage data had significantly different K-epidemiology to a healthy adult carriage whole genome collection (Marin et al. 2022) sampled in France in 2010, using a nonparametric test of independence in R package coin v1.4-3 for the K-loci and lineages included in the invasive potential model, and a comparison of proportions for each individual K-type and lineage included in our model between these two carriage collections with the “BH” method for adjusting for multiple testing.”

“Given that we observed some age and sex differences in epidemiology, the invasive potential may vary in specific patient groups. The estimates reported in this manuscript represent the average estimated relative invasive potential in BSIs across all age groups. However, 74% of BSIs are in elderly patients of greater than 59 years, and so the estimates are likely to better represent this important high-risk age group. Limiting the data to only elderly adults aged 59 years or older had a negligible effect on the rank order but, as expected, increased uncertainty in the confidence intervals of all the K-type estimates. There was insufficient data from elderly adults alone to estimate the relative invasive potential for four capsular types (KL30, K4, KL13, KL70). The only other notable difference was that K12 moved up four places in the rank order for the estimates using only elderly adults. It may thus represent a K-type that either requires a more vulnerable population or circulates more commonly in older demographics, which were not captured by the infant-mother carriage cohort. This further highlights how systematic population-based K-type screening paired with high-quality metadata can advance understanding of any potential age and sex-specific epidemiology of ExPEC infections. As further colonisation data is generated alongside additional BSI metadata, there is potential for future meta-analyses across countries that would increase the power to estimate the relative invasive potential of K-types in different groups and reduce uncertainty around the point estimates.”

4. It is reported that the genotypic diversity in capsule loci is far greater than previously reported phenotypic diversity. However, this data is not clearly shown (i.e. mapping between prior serotypes and genotypes). While alluded to in several places, tabular and graphical

representations of these comparisons would clarify the genotype to phenotype map (e.g. how different can capsule loci be and still be serologically equivalent, how similar can capsule loci be and be serologically distinct, etc.).

Thank you for pointing this out. Figure 5 shows the genetic relatedness of all observed K-loci for G2 and G3; all of the known phenotypes are labelled on the tree tips, all other tips have an unknown phenotype and represent most of the observed K-loci. We now draw attention to this fact. The supplementary tabular representation of the database now includes the number of genomes with a reported K-type, supporting the K phenotype assignment to a K-type. We specifically state that this gene-based approach, which for the most part, results in a one K to each KL and clearly describes the exception where some capsules within a serogroup share gene sets (i.e. a KL-type) but have sequence variation in the alleles (23/25 of the KL with a reported phenotype). This is a strong starting point for *in silico* K-typing with the potential for sequence-based differentiating criteria to follow, as has been proposed in the documentation for Kaptive V3.

“Known K-phenotypes only account for 34% of the numerous K-loci identified in this study with the known phenotypes sporadically distributed across the K-loci phylogeny (Fig. 5), and many K-loci corresponded to deep ancestral branches that likely represent distinct K-types.”

5. Several references are made to the detection of lineage expansion events associated with capsule switching. However, no statistical support is provided, nor any analyses that potential bias in data collections might have on these inferences (e.g. short-term clonal outbreaks included in collection).

We used the published method CaveDive for Bayesian inference of clonal expansions in a dated phylogeny to detect probable expansion events (Helekal et al. 2021). However, we do realise that the expansion results provide limited biological and epidemiological insight beyond the analyses of antigenic diversity and dating of acquisition events, and have accordingly omitted the expansion analysis from the revision.

Minor critiques

1. Line 117: per cent => percent

Corrected.

2. Line 174: know => known

Corrected.

3. Introduction is a bit under referenced

We have added further citations for the summarised concepts in the introduction.

References

- Arredondo-Alonso, Sergio, George Blundell-Hunter, Zuyi Fu, Rebecca A. Gladstone, Alfred Fillol-Salom, Jessica Loraine, Elaine Cloutman-Green, et al. 2023. "Evolutionary and Functional History of the Escherichia Coli K1 Capsule." *Nature Communications* 14 (1): 3294.
- Blackwell, Grace A., Martin Hunt, Kerri M. Malone, Leandro Lima, Gal Horesh, Blaise T. F. Alako, Nicholas R. Thomson, and Zamin Iqbal. 2021. "Exploring Bacterial Diversity via a Curated and Searchable Snapshot of Archived DNA Sequences." *PLoS Biology* 19 (11): e3001421.
- Brueggemann, A. B., T. E. Peto, D. W. Crook, J. C. Butler, K. G. Kristinsson, and B. G. Spratt. 2004. "Temporal and Geographic Stability of the Serogroup-Specific Invasive Disease Potential of Streptococcus Pneumoniae in Children." *The Journal of Infectious Diseases* 190 (7): 1203–11.
- Brueggemann, Angela B., David T. Griffiths, Emma Meats, Timothy Peto, Derrick W. Crook, and Brian G. Spratt. 2003. "Clonal Relationships between Invasive and Carriage Streptococcus Pneumoniae and Serotype- and Clone-Specific Differences in Invasive Disease Potential." *The Journal of Infectious Diseases* 187 (9): 1424–32.
- Buckles, Eric L., Xiaolin Wang, M. Chelsea Lane, C. Virginia Lockett, David E. Johnson, David A. Rasko, Harry L. T. Mobley, and Michael S. Donnenberg. 2009. "Role of the K2 Capsule in Escherichia Coli Urinary Tract Infection and Serum Resistance." *The Journal of Infectious Diseases* 199 (11): 1689–97.
- Colijn, Caroline, Jukka Corander, and Nicholas J. Croucher. 2020. "Designing Ecologically Optimized Pneumococcal Vaccines Using Population Genomics." *Nature Microbiology* 5 (3): 473–85.
- Corander, Jukka, Christophe Fraser, Michael U. Gutmann, Brian Arnold, William P. Hanage, Stephen D. Bentley, Marc Lipsitch, and Nicholas J. Croucher. 2017. "Frequency-Dependent Selection in Vaccine-Associated Pneumococcal Population Dynamics." *Nature Ecology & Evolution* 1 (12): 1950–60.
- Cross, A. S., K. S. Kim, D. C. Wright, J. C. Sadoff, and P. Gemski. 1986. "Role of Lipopolysaccharide and Capsule in the Serum Resistance of Bacteremic Strains of Escherichia Coli." *The Journal of Infectious Diseases* 154 (3): 497–503.
- Dicks, Jo, Mohammed-Abbas Fazal, Karen Oliver, Nicholas E. Grayson, Jake D. Turnbull, Evangeline Bane, Edward Burnett, et al. 2023. "NCTC3000: A Century of Bacterial Strain Collecting Leads to a Rich Genomic Data Resource." *Microbial Genomics* 9 (5): mgen000976.
- Gladstone, Rebecca A., Stephanie W. Lo, John A. Lees, Nicholas J. Croucher, Andries J. van Tonder, Jukka Corander, Andrew J. Page, et al. 2019. "International Genomic Definition of Pneumococcal Lineages, to Contextualise Disease, Antibiotic Resistance and Vaccine Impact." *EBioMedicine* 43 (May):338–46.
- Gladstone, Rebecca A., Alan McNally, Anna K. Pöntinen, Gerry Tonkin-Hill, John A. Lees, Kusti Skytén, François Cléon, et al. 2021. "Emergence and Dissemination of Antimicrobial Resistance in Escherichia Coli Causing Bloodstream Infections in Norway in 2002–17: A

- Nationwide, Longitudinal, Microbial Population Genomic Study." *The Lancet Microbe* 2 (7): e331–41.
- Hadfield, James, Nicholas J. Croucher, Richard J. Goater, Khalil Abudahab, David M. Aanensen, and Simon R. Harris. 2018. "Phandango: An Interactive Viewer for Bacterial Population Genomics." *Bioinformatics* 34 (2): 292–93.
- Handal, Nina, Håkon Kaspersen, Solveig Sølvørød Mo, Nicolas Cabanel, Silje Bakken Jørgensen, Nicolas Fortineau, Saoussen Oueslati, Thierry Naas, Philippe Glaser, and Marianne Sunde. 2025. "A Comparative Study of the Molecular Characteristics of Human Uropathogenic Escherichia Coli Collected from Two Hospitals in Norway and France in 2019." *The Journal of Antimicrobial Chemotherapy*, April, dkaf130.
- Horesh, Gal, Grace A. Blackwell, Gerry Tonkin-Hill, Jukka Corander, Eva Heinz, and Nicholas R. Thomson. 2021. "A Comprehensive and High-Quality Collection of Escherichia Coli Genomes and Their Genes." *Microbial Genomics* 7 (2).
<https://doi.org/10.1099/mgen.0.000499>.
- Ingle, Danielle J., Mary Valcanis, Alex Kuzevski, Marija Tauschek, Michael Inouye, Tim Stinear, Myron M. Levine, Roy M. Robins-Browne, and Kathryn E. Holt. 2016. "In Silico Serotyping of E. Coli from Short Read Data Identifies Limited Novel O-Loci but Extensive Diversity of O:H Serotype Combinations within and between Pathogenic Lineages." *Microbial Genomics* 2 (7): e000064.
- Kallonen, Teemu, Hayley J. Brodrick, Simon R. Harris, Jukka Corander, Nicholas M. Brown, Veronique Martin, Sharon J. Peacock, and Julian Parkhill. 2017. "Systematic Longitudinal Survey of Invasive Escherichia Coli in England Demonstrates a Stable Population Structure Only Transiently Disturbed by the Emergence of ST131." *Genome Research*, July.
<https://doi.org/10.1101/gr.216606.116>.
- Lipworth, Samuel, William Matlock, Liam Shaw, Karina-Doris Vihta, Gillian Rodger, Kevin Chau, Leanne Barker, et al. 2024. "The Plasmidome Associated with Gram-Negative Bloodstream Infections: A Large-Scale Observational Study Using Complete Plasmid Assemblies." *Nature Communications* 15 (1): 1612.
- Løchen, Alessandra, James E. Truscott, and Nicholas J. Croucher. 2022. "Analysing Pneumococcal Invasiveness Using Bayesian Models of Pathogen Progression Rates." *PLoS Computational Biology* 18 (2): e1009389.
- Lo, Stephanie W., Rebecca A. Gladstone, Andries J. van Tonder, John A. Lees, Mignon du Plessis, Rachel Benisty, Noga Givon-Lavi, et al. 2019. "Pneumococcal Lineages Associated with Serotype Replacement and Antibiotic Resistance in Childhood Invasive Pneumococcal Disease in the Post-PCV13 Era: An International Whole-Genome Sequencing Study." *The Lancet Infectious Diseases* 19 (7): 759–69.
- Mäklin, Tommi, Harry A. Thorpe, Anna K. Pöntinen, Rebecca A. Gladstone, Yan Shao, Maiju Pesonen, Alan McNally, et al. 2022. "Strong Pathogen Competition in Neonatal Gut Colonisation." *Nature Communications* 13 (1): 7417.
- Marin, Julie, Olivier Clermont, Guilhem Royer, Mélanie Mercier-Darty, Jean Winoc Decousser, Olivier Tenaillon, Erick Denamur, and François Blanquart. 2022. "The Population Genomics of Increased Virulence and Antibiotic Resistance in Human Commensal Escherichia Coli over 30 Years in France." *Applied and Environmental Microbiology* 88 (15): e0066422.
- Mba, Ifeanyi Elibe, Hyelnya Cletus Sharndama, Zikora Kizito Glory Anyaegbunam, Chijioko

- Chinedu Anekpo, Ben Chibuzo Amadi, Daji Morumda, Yandev Doowuese, Uchechi Justina Ihezuo, Joseph Ukomadu Chukwukelu, and Onyekachi Philomena Okeke. 2023. "Vaccine Development for Bacterial Pathogens: Advances, Challenges and Prospects." *Tropical Medicine & International Health: TM & IH* 28 (4): 275–99.
- Merino, Irene, Stephen B. Porter, Brian Johnston, Connie Clabots, Paul Thuras, Patricia Ruiz-Garbajosa, Rafael Cantón, and James R. Johnson. 2020. "Molecularly Defined Extraintestinal Pathogenic Escherichia Coli Status Predicts Virulence in a Murine Sepsis Model Better than Does Virotype, Individual Virulence Genes, or Clonal Subset among E. Coli ST131 Isolates." *Virulence* 11 (1): 327–36.
- Mostafavi, Seyyed Khalil Shokouhi, Shahin Najjar-Peerayeh, Ashraf Mohabbati Mobarez, and Mehdi Kardoust Parizi. 2019. "Characterization of Uropathogenic E. Coli O25b-B2-ST131, O15:K52:H1, and CGA: Neutrophils Apoptosis, Serum Bactericidal Assay, Biofilm Formation, and Virulence Typing." *Journal of Cellular Physiology* 234 (10): 18272–82.
- Moxon, E. R., and J. S. Kroll. 1990. "The Role of Bacterial Polysaccharide Capsules as Virulence Factors." *Current Topics in Microbiology and Immunology* 150:65–85.
- Ojala, Fanni, Henri Pesonen, Rebecca A. Gladstone, Tommi Mäklin, Gerry Tonkin-Hill, Pekka Marttinen, and Jukka Corander. 2024. "Basic Reproduction Number for Pandemic *Escherichia Coli* Clones Varies Markedly and Can Be Comparable to Pandemic Influenza Viruses." *bioRxiv*. <https://doi.org/10.1101/2024.05.08.593267>.
- Ondov, Brian D., Todd J. Treangen, Páll Melsted, Adam B. Mallonee, Nicholas H. Bergman, Sergey Koren, and Adam M. Phillippy. 2016. "Mash: Fast Genome and Metagenome Distance Estimation Using MinHash." *Genome Biology* 17 (1): 132.
- Pearse, Oliver, Allan Zuza, Edith Tewesa, Patricia Siyabu, Alice J. Fraser, Jennifer Cornick, Kondwani Kawaza, et al. 2025. "High Diversity of Escherichia Coli Causing Invasive Disease in Neonates in Malawi Poses Challenges for O-Antigen Based Vaccine Approach." *Communications Medicine* 5 (1): 298.
- Pöntinen, Anna K., Rebecca A. Gladstone, Henri Pesonen, Maiju Pesonen, François Cléon, Benjamin J. Parcell, Teemu Kallonen, et al. 2024. "Modulation of Multidrug-Resistant Clone Success in Escherichia Coli Populations: A Longitudinal, Multi-Country, Genomic and Antibiotic Usage Cohort Study." *The Lancet. Microbe* 5 (2): e142–50.
- Sands, Kirsty, Maria J. Carvalho, Edward Portal, Kathryn Thomson, Calie Dyer, Chinenye Akpulu, Robert Andrews, et al. 2021. "Characterization of Antimicrobial-Resistant Gram-Negative Bacteria That Cause Neonatal Sepsis in Seven Low- and Middle-Income Countries." *Nature Microbiology* 6 (4): 512–23.
- Seemann, Torsten. 2014. "Prokka: Rapid Prokaryotic Genome Annotation." *Bioinformatics*, March.
- Shao, Yan, Samuel C. Forster, Evdokia Tsaliki, Kevin Vervier, Angela Strang, Nandi Simpson, Nitin Kumar, et al. 2019. "Stunted Microbiota and Opportunistic Pathogen Colonization in Caesarean-Section Birth." *Nature* 574 (7776): 117–21.
- Shao, Yan, Cristina Garcia-Mauriño, Simon Clare, Nicholas J. R. Dawson, Andre Mu, Anne Adoum, Katherine Harcourt, et al. 2024. "Primary Succession of Bifidobacteria Drives Pathogen Resistance in Neonatal Microbiota Assembly." *Nature Microbiology* 9 (10): 2570–82.
- Suerbaum, S., S. Friedrich, H. Leying, and W. Opferkuch. 1994. "Expression of Capsular

- Polysaccharide Determines Serum Resistance in Escherichia Coli K92." *Zentralblatt Für Bakteriologie: International Journal of Medical Microbiology* 281 (2): 146–57.
- Thänert, Robert, Joohee Choi, Kimberly A. Reske, Tiffany Hink, Anna Thänert, Meghan A. Wallace, Bin Wang, et al. 2022. "Persisting Uropathogenic Escherichia Coli Lineages Show Signatures of Niche-Specific within-Host Adaptation Mediated by Mobile Genetic Elements." *Cell Host & Microbe* 0 (0). <https://doi.org/10.1016/j.chom.2022.04.008>.
- Tonkin-Hill, Gerry, Neil MacAlasdair, Christopher Ruis, Aaron Weimann, Gal Horesh, John A. Lees, Rebecca A. Gladstone, et al. 2020. "Producing Polished Prokaryotic Pangenomes with the Panaroo Pipeline." *Genome Biology* 21 (1): 180.
- Vann, W. F., T. Soderstrom, W. Egan, F. P. Tsui, R. Schneerson, I. Orskov, and F. Orskov. 1983. "Serological, Chemical, and Structural Analyses of the Escherichia Coli Cross-Reactive Capsular Polysaccharides K13, K20, and K23." *Infection and Immunity* 39 (2): 623–29.
- Wyres, Kelly L., Sarah M. Cahill, Kathryn E. Holt, Ruth M. Hall, and Johanna J. Kenyon. 2020. "Identification of Acinetobacter Baumannii Loci for Capsular Polysaccharide (KL) and Lipooligosaccharide Outer Core (OCL) Synthesis in Genome Assemblies Using Curated Reference Databases Compatible with Kaptive." *Microbial Genomics* 6 (3). <https://doi.org/10.1099/mgen.0.000339>.
- Wyres, Kelly L., Ryan R. Wick, Claire Gorrie, Adam Jenney, Rainer Follador, Nicholas R. Thomson, and Kathryn E. Holt. 2016. "Identification of Klebsiella Capsule Synthesis Loci from Whole Genome Data." *Microbial Genomics* 2 (12): e000102.
- Xie, Zhiqun, and Haixu Tang. 2017. "ISEScan: Automated Identification of Insertion Sequence Elements in Prokaryotic Genomes." *Bioinformatics (Oxford, England)* 33 (21): 3340–47.
- Zhou, Zhemin, Jane Charlesworth, and Mark Achtman. 2021. "HierCC: A Multi-Level Clustering Scheme for Population Assignments Based on Core Genome MLST." *Bioinformatics (Oxford, England)* 37 (20): 3645–46.

Reviewers Comments:

Reviewer #1 (Remarks to the Author):

I have no further questions/suggestions for the authors.

Reviewer #2 (Remarks to the Author):

In this revised version, Gladstone and colleagues have substantially strengthened their study. The inclusion of larger datasets confirm their previously observed results and reduces potential sampling biases. The new analyses provide additional insights. Furthermore, the authors have undertaken an extensive reformulation of several sections in direct response to reviewer feedback. This improves the clarity of the work and allows the impact of the results to be more fully appreciated, particularly in light of the expanded discussion on study limitations. The standardization of the nomenclature has also been improved and is now much clearer overall.

However, it remains somewhat counterintuitive that the K and KL designations do not always correspond to one another, as these labels are determined by algorithmic assignment rather than human curation, particularly in cases where a K-phenotype is already associated.

We thank the reviewer for their patience regarding this issue. We acknowledge that the KL designations were somewhat counterintuitive. KL loci nomenclature now reflects the known paired phenotypes, e.g. K1 is encoded by the KL1 locus. K-loci for which K-types have not yet been phenotypically identified were assigned KL numbers starting from KL110. As there are 6 remaining known phenotypes for which no paired genotype (KL) and phenotype are yet available, we will update the KL to the corresponding K number as this data arises, and this will be clearly logged in the GitHub database in tandem with a version update for posterity.

Reviewer #3 (Remarks to the Author):

I thank the authors for addressing many of the key reviewer comments. However, I still have concerns regarding the analyses that are being used to infer that the invasive potential of E. coli strains can be directly attributed to specific K-types. The authors do a reasonable job justifying their use of the infant/mother cohort as a proxy for colonization among healthy adults through comparison of strain distributions to other cohorts.

However, I am not convinced that a population of healthy adults is a proper control population to compare to hospitalized patients with BSI's. The ideal control population would be a propensity-score matched cohort of colonized patients who did not proceed to BSI. While I appreciate the author's point that this is a logistically challenging cohort to assemble given the relative rarity of the infection, this does not mitigate concerns over not controlling for relevant factors (i.e. age, sex, antibiotic use, comorbid conditions, etc.), while claiming a direct impact of the variable of interest. I would ask at a minimum for the following:

1) Properly controlling for age and sex, which seem to be variables that are available for both cases and controls, by adjusting for these variables in their models. I believe the authors have performed analyses to attempt to control for age, but in the methods it seems they did this by restricting to older adults in the BSI cohort. However, the BSI cohort is already enriched in older

adults, so I was unclear on how this controlled for the age-bias in cases.

We thank the reviewer for their constructive comments, which have helped us better highlight the limitations and adjust our conclusions to reflect the data. We have sex data for 80% of the BSI and 94% of the carriage data used in the model. However, given the data structure, we could not directly include age and sex as covariates in the model due to strong correlations between age and dataset membership and the observed age-sex interaction. Instead, we were able to quantify the odds of a BSI patient being ≥ 60 in males compared to females for each common combination of lineage and capsule. For most lineage and capsule combinations, we confirmed that the proportion of older patients is higher in males than in females. However, these associations were not significant except for the CC131-C2 with the K5 capsule.

As we could not explicitly include age as a covariate, we aimed to highlight that the high-risk age group (>59 years) is overrepresented in the data and, therefore, that the estimates are more relevant for this important clinical population. Furthermore, by excluding all the other age groups, we removed the assumption that the invasive potential is homogeneous across age groups and limited it to the simpler assumption that carriage is homogeneous across age groups, which we have evidence for from the comparison between the UK and French carriage data. We have now amended the Discussion, as suggested, regarding the need for a propensity-score-matched cohort of colonised patients who did not progress to BSI to determine causality, and we have noted the potential that colonisation in elderly adults differs from that in the general population. Even if the assumption of homogeneous colonisation were later to be found incorrect in future research, we have taken the reviewer's suggestion below and explicitly stated that our estimates can be interpreted as indicating that certain capsular types are enriched in BSI compared to the general healthy adult population, highlighting capsules of concern without implying causality.

2) Tempering language to reflect the associative nature of these findings, given that this was an observational study design with poorly matched populations. Here are some examples, where I find that the analysis does not support the conclusions:

(i) "Estimating the relative invasive potential allowed us to put these experimental findings into a wider context by ranking the capsular types in terms of their propensity to be found in BSIs, given the exposure frequency in asymptomatic colonisation."

In fact, with this study design you do not know the exposure frequencies in the BSI population, and whether they match the healthy child/adult cohorts.

(ii) "However, 74% of BSIs are in elderly patients of greater than 59 years, and so the estimates are likely to better represent this important high-risk age group."

The direct attributable risk associated with capsule cannot be inferred in elderly cohorts, as you have not presented data on colonization frequencies in this cohort, nor controlled for other relevant variables. In fact, you may be overestimating the risk of some capsule types in the elderly cohort if they are present at higher frequencies in asymptomatic patients that are properly matched on age, sex and other relevant factors.

What you can say is that certain capsular types are enriched in BSI, as compared to the general healthy adult population, which may have its own important implications. However, this is

different than inferring a causal role of specific K-types in the risk of colonized patients developing BSIs, which is what is being inferred/implied/stated.

Importantly, it may be that with a properly controlled study that the findings would remain the same, but it does not seem to me that this is a justifiable conclusion to draw from the data presented.

Several amendments were made to the Discussion to temper the language and avoid the impression of quantified causal effects for the different K types, as suggested by the reviewer. We edited all instances where a K-type was previously stated to 'cause' an infection, in order to avoid misleading interpretation. Please find the relevant section in the discussion at lines 289-315.

Please look into and address this final point from the reviewer to make sure the limitations are clearly stated and that the title and abstract reflect the message of the main text (we have some suggested edits, details provided below). Reviewer comments: I appreciate the authors substantially addressing my concerns by adding significantly to the discussion. I think the paragraph added adequately addresses my concerns. However, I have two remaining concerns:

1) I may just be misunderstanding, but I was not able to follow how the sub-analyses to assess the issue of age mismatch between cases and controls alleviate the concern. In particular:

"first, we found that the odds of a BSI patient being ≥ 60 in males compared to females were not significant for most lineage-capsule combinations, except for CC131-C2 with K5."

-- This mitigates sex as a confounder, but I don't see how this mitigates age.

"Secondly, as we could not explicitly include age as a covariate, we highlight that the elderly adult high-risk age group (≥ 60 years) is overrepresented in the data (74% of BSIs) and, therefore, that the estimates are most relevant for this important clinical population."

--This is quite circular. The issue is that the over-representation of elderly individuals in the BSI groups makes it impossible to determine the direct attributable risk of capsule type to BSI. The over-representation doesn't address the concern, it is the concern.

"Furthermore, we ran additional iterations of the model, excluding the other age groups, to remove the assumption that the invasive potential is homogeneous across age groups and limit it to the simpler assumption for which we have some evidence: that carriage is homogeneous in the healthy population."

-- If I understand correctly, this analysis just limited the BSIs to >60 , not the controls? I don't follow the logic for how this mitigates concern regarding age as a confounder. The concern is that carriage is not homogeneous between healthy and BSI populations, potentially due to age itself (e.g. altered gut microbiota, altered immune status) or due to altered exposure patterns (e.g. frequent healthcare exposure).

To me, it would be preferable to just acknowledge the limitation, instead of trying to work/talk around structural issues in the data.

The reviewer makes good points with useful suggestions. Following on the comments by the reviewer and editor suggestions, we have removed the discussion of additional analyses and instead simply acknowledged the limitations related to carriage data not being currently available for the elderly population.

2) While the discussion softens language and acknowledges that the study design can only yield associations, the title and the abstract don't have this nuance. Given the key roles of the title and the abstract in conveying the central message to non-domain experts, especially in a high-impact journal, I think more effort should go into softening the language/implications to accurately reflect the data/analysis performed.

We have edited the title, and abstract as suggested by the reviewer and editor.